# Disorder-free spin glass transitions and jamming in exactly solvable mean-field models

**Hajime Yoshino[1,2]\***

**1** Cybermedia Center, Osaka University, Toyonaka, Osaka 560-0043, Japan
**2** Graduate School of Science, Osaka University, Toyonaka, Osaka 560-0043, Japan

\* yoshino@cmc.osaka-u.ac.jp

## Abstract

We construct and analyze a family of $M$-component vectorial spin systems which exhibit glass transitions and jamming within supercooled paramagnetic states without quenched disorder. Our system is defined on lattices with connectivity $c = \alpha M$ and becomes exactly solvable in the limit of large number of components $M \to \infty$. We consider generic $p$-body interactions between the vectorial Ising/continuous spins with linear/non-linear potentials. The existence of self-generated randomness is demonstrated by showing that the random energy model is recovered from a $M$-component ferromagnetic $p$-spin Ising model in $M \to \infty$ and $p \to \infty$ limit. In our systems the quenched disorder, if present, and the self-generated disorder act additively. Our theory provides a unified mean-field theoretical framework for glass transitions of rotational degree of freedoms such as orientation of molecules in glass forming liquids, color angles in continuous coloring of graphs and vector spins of geometrically frustrated magnets. The rotational glass transitions accompany various types of replica symmetry breaking. In the case of repulsive hardcore interactions in the spin space, the criticality of the jamming or SAT/UNSTAT transition becomes the same as that of hardspheres.



# 1  Introduction

Simple spin models often provide useful grounds to develop statistical mechanical approaches for various kinds of phase transitions. For the glass transition [1–3], which is one of the most important open problem in physics, a family of mean-field spinglass models called as the random energy model [4] and $p$-spin spinglass models [4–11] have played important roles.

The concepts and techniques used in the spinglass theory have promoted substantial progress of the first principle theory for the glass transitions of supercooled liquids [12, 13]. Most notably exact mean-field theory in the large dimensional limit was constructed recently for the hardspheres [14–20] using the replica approach on the supercooled liquids [12, 13].

There remains, however, a conceptual problem regarding the origin of the randomness. The spinglass models [21], which have been developed originally by Edwards and Anderson to model a class of disordered and frustrated magnetic materials [22], have quenched disorder which is apparently absent in glass forming liquids. It is often emphasized in the studies of spinglass materials that both the quenched disorder and frustration are important. However it is believed that somehow the disorder is self-generated in structural glasses which are born out of supercooled liquid and thus the quenched disorder is not necessarily. Early seminal works [23–27] have suggested that self-generated randomness are actually realized in some spin models without quenched disorder. However a comprehensive understanding of the mechanism of the putative self-generated randomness and its possible relation to the quenched randomness in spinglass models is still lacking.

In order to shed a light on this issue, we explicitly develop and analyze a family of mean-field vectorial spin models. We show that they exhibit glass transitions within their supercooled paramagnetic phases without quenched disorder. Our model consists of $M$-component vectorial spins, which can take either the Ising $\pm 1$ or continuous values, put on tree-like lattices with connectivity $c = \alpha M$, which becomes exactly solvable in the limit of large number of components $M \to \infty$. We perform a unified study of the crystalline phase (e.g. ferromagnetic phase), supercooled paramagnetic phases and glassy phases of the same model. We clarify the condition needed to ensure local stability of supercooled liquids and glasses against crystallization. We demonstrate in particular that the theoretical results of the random energy model [4] and the $p$-spin spinglass models [4, 5, 9] can be fully recovered from a $M$-component $p$-spin models with purely ferromagnetic interactions within their supercooled paramagnetic phases. This proves the existence of the self-generated randomness in our models. In a sense this observation strengthen the view that the $p$-spin spinglass models are good caricature spin models for glass transitions [2, 11] because the quenched disorder is actually not needed. We show that the quenched disorder, if present, add on top of the self-generated randomness.

Glass transition of the rotational or spin degrees of freedom is an important problem by itself and can be found not only in the spinglasses but also in many other real systems. It should be noted first that most of the molecules and colloidal particles in glass-forming liquids are not simply spherical but have rotational degrees of freedom because of their shapesq or patches on their surfaces (see Fig. 1c)) and the rotational degree of freedom can exhibit glass transitions simultaneously or separately from that of the translational degrees of freedom. Sometimes the rotational degrees of freedoms alone exhibit glassiness on top of crystalline long-ranged order of the translational degrees of freedom. This happens for instance in the so called plastic crystals where the rotations of molecules slow down and eventually exhibit glass transitions [28]. Another important problem is the spinglass transition found in frustrated magnets but without quenched disorder (Fig. 1b)) [29, 30]. Possibilities of disorder-free spinglass transitions have been a matter of long debate in the field of frustrated magnets. We expect our results provide a useful basis to tackle these problems theoretically.

Within our formalism we consider $p$-body interactions through generic non-linear potentials. In particular we apply the scheme to the case of a $M$-component continuous spins interacting with each other through a hardcore potential which enables jamming transition of the vectorial spins. Here jamming means to loose thermal fluctuations by tightening the constraints. This is relevant in the continuous constrained satisfaction problems such as the circular coloring of graphs or periodic scheduling [31] (Fig. 1 a)): the problem is to put continuous colors parametrized by "color angle" $0 < \theta < 2\pi$ on the vertexes of a given graph such that

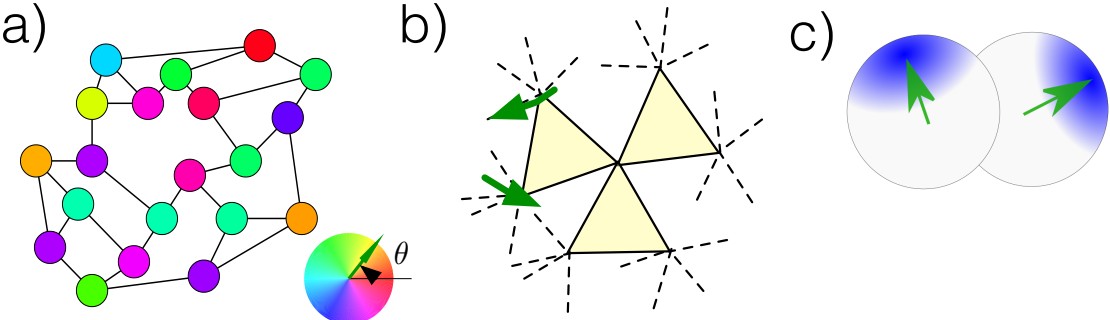

Figure 1: Glassy systems carrying 'spins' representing rotational degree of freedoms. a) **Continuous coloring of a graph**: The color angle $0 < \theta < 2\pi$, as in the standard HSV color map, can be represented by a XY spin, i.e. a vector with $M = 2$ component (green arrow). The example shown here is a solution to the requirement that color angle on adjacent vertexes must be greater than or equal to $2\pi/3$. b) **Geometrically frustrated magnets**: vectorial spins (green arrows) with anti-ferromagnetic couplings on adjacent vertexes on corner sharing triangles (e.g. kagome lattice), tetrahedra (e.g. pyrochlore lattice). The ground states are highly degenerate due to the loose connectivity of the lattices. c) **Glass forming liquids of molecules or colloidal particles with 'spins'**: a simple molecule or a colloidal particle, like the Janus particle, is symmetric under rotation around an axis whose direction can be specified by a spin.

angles on adjacent vertexes are sufficiently separated from each other. This is exactly a continuous version of the usual coloring problem where one is allowed to use only discrete colors like red, green and blue [32, 33]. Remarkably a recent study has shown that a discretized version of the circular coloring problem exhibits a complex free-energy landscape reminiscent of continuous replica symmetry breaking [34].

Increasing the coordination number $c$ of the graph, the solution space exhibit clustering transition (glass transition) and eventually SAT/UNSAT transition (jamming) above which one cannot find a solution which satisfies the constraints. Given the continuous variables, an interesting question is the universality class of the SAT/UNSAT transition. Closely following the analysis done on hardspheres in the $d \to \infty$ limit [16], we will show that the jamming criticality of our model belong indeed to the same universality of the hardspheres. Our result extends the result on the perceptron problem [35–37] which can be regarded as a special case $p = 1$ of our models.

The organization of this paper is as follows. In sec. 2 we introduce a family of large $M$-component vectorial Ising/continuous spin models with a generalized $p$-body interaction described by linear/non-linear potentials. We introduce a disorder-free model that has no quenched disorder and also a model which interpolates between the disorder-free model and a fully disordered spinglass model. In sec. 3 we discuss possible crystalline orderings in our disorder-free models and possibility to realize supercooled paramagnetic states, which are crucial as the basis for glass transitions to take place without the quenched disorder. In sec. 4 we show that the random energy model can be recovered from a $M$-component $p$-spin Ising ferromagnetic model with a linear potential in the limit $M \to \infty$ and $p \to \infty$. This demonstrates the presence of self-generated randomness in our models. In sec. 5 we derive the replicated free-energy functional in terms of the glass and crystalline order parameters. We also discuss stability of the supercooled paramagnetic state and the glassy states against crystallization. In sec. 6 we establish the connection between our model with linear potential and the standard

$p$-spin spinglass models. Then in the subsequent sections, we turn to study glassy phases of our model with non-linear potentials limiting our selves to the case of continuous spins. In sec. 7 we discuss some general results within the replica symmetric (RS) ansatz. In sec. 8 we discuss some general results within 1 step and continuous replica symmetry breaking (RSB) ansatz. In sec. 9 we analyze the model with a quadratic potential as the simplest case of non-linear potential. In sec. 10 we analyze in detail the model with a hardcore potential which exhibit jamming. Finally in sec. 11 we conclude this paper with some summary and remarks. Some technical details are reported in the appendices.

## 2 Vectorial spin model

### 2.1 Generic model

Let us now introduce the models that we study in this paper. We consider vectorial spins with $M$ components $\mathbf{S}_i = (S_i^1, S_i^2, \ldots, S_i^M)$ ($i = 1, 2, \ldots, N$) normalized such that

$$|\mathbf{S}|^2 = \sum_{\mu=1}^M (S^\mu)^2 = M. \tag{1}$$

More specifically we consider two types of spins,

- **Ising spin**

  $M$-component Ising spin with $S_i^\mu \in (-1, 1)$ for $\mu = 1, 2, \ldots, M$.

- **Continuous spin** $M$-component continuous spin with length $|\mathbf{S}| = \sqrt{M}$ which can continuously rotate in the $M$-dimensional space. It is known in some models that this case is closely related to the 'spherical model' which has just $M = 1$ component spins $S_i$ normalized by a global constraint $\sum_{i=1}^N S_i^2 = N$ [38, 39].

The spins are put on the vertexes of lattices (graphs) which are locally tree-like with no closed loops as shown in Fig. 2. Spins are involved in $p$-body interactions represented by factor nodes (interaction node) ■ in the figure. Each spin is involved in $c = \alpha M$ $p$-tuples. Thus the number of the $p$-tuples is given by

$$N_■ = NM(\alpha/p). \tag{2}$$

In the present paper we take not only the thermodynamic limit $N \to \infty$ but also the limit of large number of spin components $M \to \infty$, which scales independently of $N$. As we will find below this brings about important consequences. Later we will consider a special limit $p, \alpha \to \infty$ with the ratio $\gamma = \alpha/p$ fixed to a constant of $O(1)$ in sec. 4. Otherwise the parameters $\alpha$ and $p$ are both constants of $O(1)$.

The interaction between the spins is given by a generalized $p$-body interaction,

$$H = \sum_{■=1}^{N_■} V\left(r_■\right), \tag{3}$$

where

$$r_■ = \delta - \frac{1}{\sqrt{M}} \sum_{\mu=1}^M X_■^\mu S_{1(■)}^\mu S_{2(■)}^\mu \cdots S_{p(■)}^\mu. \tag{4}$$

Here $1(\blacksquare), 2(\blacksquare), \ldots, p(\blacksquare)$ represent the spins involved in a given $p$-tuple $\blacksquare$. The function $V(r)$ represents a generic interaction potential. We will call the argument variable $r_\blacksquare$ as 'gap', whose meaning will become clear later, with $\delta \in \mathcal{R}$ being a control parameter.

In the present paper we mainly study models without quenched disorder (disorder-free model) but we also discuss models with quenched disorder (disordered model).

- **disorder-free model**

$$X_\blacksquare^\mu = 1 \tag{5}$$

- **disordered model**

$$X_\blacksquare^\mu = \frac{\lambda}{\sqrt{M}} + \sqrt{1 - \left(\frac{\lambda}{\sqrt{M}}\right)^2} \xi_\blacksquare^\mu \qquad \left(0 \le \frac{\lambda}{\sqrt{M}} \le 1\right) \tag{6}$$

Here $\xi_\blacksquare^\mu$s are mutually independent, quenched random variables which obey the Gaussian distribution with zero mean and unit variance. The parameter $\lambda$ represents the strength of the 'disorder-free' part in the disordered model. Note that the disorder-free model is recovered by choosing $\lambda/\sqrt{M} = 1$. In the other limit $\lambda/\sqrt{M} = 0$ we have completely disordered, spinglass model. Thus we have a smooth interpolation between the two limits with this parametrization.

The free-energy $F$ of the system can be written as,

$$-\beta F = \log Z, \tag{7}$$

where $\beta$ is the inverse temperature. The partition function $Z$ is defined as

$$Z = \left(\prod_i \mathrm{Tr}_{\mathbf{S}_i}\right) \prod_\blacksquare e^{-\beta V(r_\blacksquare)}$$

$$= \prod_\blacksquare \left\{\int_{-\infty}^\infty \frac{d\kappa_\blacksquare}{2\pi} Z_{\kappa_\blacksquare} e^{i\kappa_\blacksquare \delta}\right\} \left(\prod_i \mathrm{Tr}_{\mathbf{S}_i}\right) \exp\left[\frac{1}{\sqrt{M}} \sum_{\mu=1}^M \sum_\blacksquare (-i\kappa_\blacksquare) X_\blacksquare S_{1(\blacksquare)}^\mu S_{2(\blacksquare)}^\mu \cdots S_{p(\blacksquare)}^\mu\right]. \tag{8}$$

Here $\mathrm{Tr}_{\mathbf{S}}$ represents a trace over the spin space of the spin $\mathbf{S}$,

$$\text{(Ising)} \qquad \mathrm{Tr}_{\mathbf{S}} = \prod_{\mu=1}^M \sum_{S^\mu = \pm 1}, \tag{9}$$

$$\text{(Continuous)} \qquad \mathrm{Tr}_{\mathbf{S}} = \int d\mathbf{S} = \left(\prod_{\mu=1}^M \int_{-\infty}^\infty dS^\mu\right) M \int_{-i\infty}^{i\infty} \frac{d\lambda}{2\pi} e^{\lambda(M - \sum_{\mu=1}^M (S^\mu)^2)}$$

$$= M \int_{-i\infty}^{i\infty} \frac{d\lambda}{2\pi} e^{M\lambda} \prod_{\mu=1}^M \int_{-\infty}^\infty dS^\mu e^{-\lambda(S^\mu)^2}, \tag{10}$$

where $\int d\mathbf{S}_i$ is an integration over the surface of the $M$-dimensional sphere with diameter $\sqrt{M}$. We have also introduced a Fourier transform of the Boltzmann's factor,

$$Z_\kappa \equiv \int_{-\infty}^\infty dh e^{-i\kappa h} e^{-\beta V(h)}. \tag{11}$$

Lastly let us note the similarity of our model to the so called $M - p$ spinglass model [40–42]. In the $M - p$ spinglass model, one considers $M$-component Ising spins on each vertex much as

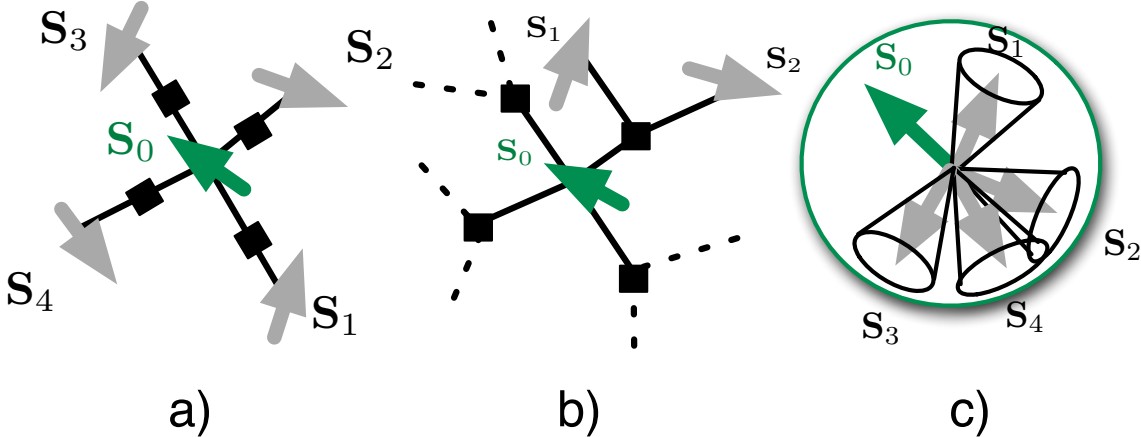

Figure 2: A schematic figure of the model. Panel a) is for the cases of $p = 2$ and b) is for the case of $p = 3$-body interaction on a graph with connectivity $c = 4$. Vectorial spins with $M$ components, in this example $M = 3$ (Heisenberg spins), are put on the vertexes of a lattice or a graph as shown in the left panel a). The filled square represents the interaction nodes each of which connects a set of $p$ spins on the vertexes (variable nodes) interacting with each other. For the hardcore potential given by Eq. (224) the spin $\mathbf{S}_0$ in panel c) is excluded from the cones around each of the neighboring spins $\mathbf{S}_1, \mathbf{S}_2, \mathbf{S}_3$. (Note that, for instance, $\mathbf{S}_2$ and $\mathbf{S}_4$ can overlap if they are not directly connected by a link). The size of the cones grows with decreasing the parameter $\delta$. Thus the excluding volume effect becomes larger by decreasing $\delta$ or increasing the connectivity $c$.

in our model. Then $p \geq 2$-body interactions are introduced between a pair of sites, say $i$ and $j$, taking possible $p$-tuples using the $2M$ components of the spins. The model becomes exactly solvable in the $M \to \infty$ limit [42] much as in our model. Moreover the model is very useful to study finite dimensional effects [40, 41]. Although it is slightly different from our model, we anticipate that much of the analysis we perform in the following could be done also in the geometry of the $M - p$ spinglass model.

## 2.2 Linear and non-linear potentials

The most simple potential is the linear potential,

$$\text{(linear potential)} \qquad V(x) = Jx \qquad J > 0. \tag{12}$$

This is a $p$-spin ferromagnetic model. We will use this potential in order to establish connections to the random energy model (sec. 4) and the $p$-spin spinglass models (sec. 6).

As a simplest non-linear potential, we will consider briefly in sec. 9 the quadratic potential,

$$\text{(quadratic potential)} \qquad V(x) = \frac{\epsilon}{2}x^2 \qquad \epsilon > 0. \tag{13}$$

We will study in detail in sec. 10 the case of more strongly non-linear potential,

$$\text{(soft/hardcore potential)} \qquad V(x) = \epsilon x^2 \theta(-x). \tag{14}$$

The hardcore potential is obtained in the $\epsilon \to \infty$ limit. This amount to bring in an excluded volume effect in the spin space similarly to the interaction between the hardspheres (See Fig. 2

c)). With $p = 2$ body interaction it can be used for the continuous coloring problem shown in Fig. 1 a)): spins representing the color angles on adjacent vertexes are forced to be separated in angle *larger than* $\cos^{-1}(\delta/\sqrt{M})$ for the hardcore potential (See Fig. 2 c)). In the case of $p = 1$, and in the presence of quenched disorder $\xi^\mu$'s in Eq. (89), the problem becomes the perceptron problem [43][35]. The case $p = 2$ was also studied in part by a seminal work [44]. In the present paper we study the cases of $p \geq 2$.

## 2.3 Pressure, distribution of gaps and isostaticity

With the soft/hardcore potential given by Eq. (14), the system becomes more constrained as we decrease the parameter $\delta$ much as an assembly of hardspheres becomes more constrained as the diameter of the spheres increase so that the volume fraction increases. This motivates us to introduce 'pressure' as an analogue of that in particulate systems,

$$\Pi = -\frac{1}{N_\blacksquare}\frac{\partial \beta F}{\partial \delta}. \tag{15}$$

The normalization factor $N_\blacksquare$ is simply the number of interaction links in the system which is given by Eq. (2). Then it is also useful to introduce the distribution function of the gap,

$$g(r) \equiv \left\langle \delta\left(r - r(\mathbf{S}_{i_1},\ldots,\mathbf{S}_{i_p})\right)\right\rangle \tag{16}$$

$$= \frac{1}{N_\blacksquare}\frac{\delta(-\beta F)}{\delta \ln e^{-\beta V(r)}}. \tag{17}$$

In the 1st equation $\langle\ldots\rangle$ is the thermal average. In the 2nd equation $\delta/\delta \ln e^{-\beta V(r)}$ is a functional derivative. Apparently the distribution function of the gap $g(r)$ is analogous to the radial distribution function in the particulate systems. The pressure given by Eq. (15) can be rewritten using $\partial(-\beta F)/\partial\delta = \int_{-\infty}^{\infty} dr \frac{\delta(-\beta F)}{\delta \ln e^{-\beta V(r)}}(\ln e^{-\beta V(r)})'$ and $g(r)$ defined above as,

$$\Pi = \int_{-\infty}^{\infty} dr g(r)(\ln e^{-\beta V(r)})' = \int_{-\infty}^{\infty} dr g(r)(-\beta V'(r)). \tag{18}$$

This is the analogue of the virial equation for the pressure in the liquid theory [45].

Given $N$ spins $\mathbf{S}_i$ ($i = 1, 2, \ldots, N$) with $M$ components, which are normalized such that $|\mathbf{S}_i|^2 = M$, the total number of the degrees of freedom is $N(M-1)$. Each spin is involved in $c = \alpha M$ sets of $p$-body interactions (See Fig. 2). We say the gap associated with such an interaction is *closed* if $r(\mathbf{S}_{i_1},\ldots,\mathbf{S}_{i_p}) < 0$. The fraction of the interactions or contacts whose gaps are closed can be written as

$$f_{\text{closed}} = \lim_{\epsilon \to 0}\int_{-\infty}^{\epsilon} dr g(r), \tag{19}$$

where $g(r)$ is the distribution function of the gap defined in Eq. (17). This means there are $N_\blacksquare f_{\text{closed}}$ constrains. Then isostaticity implies

$$N(M-1) = N_\blacksquare \lim_{\epsilon \to 0}\int_{-\infty}^{\epsilon} dr g(r) \tag{20}$$

or

$$1 = \frac{\alpha}{p}\lim_{\epsilon \to 0}\int_{-\infty}^{\epsilon} dr g(r) \tag{21}$$

in the $M \to \infty$ limit.

## 3 Supercooled spin liquid states, crystalline states and their stability

In this section we focus on the crystallization and possibility of super-cooling, i. e. realization of supercooled paramagnetic state which is at least locally stable against crystallization. This is an important step toward realization of glasses without quenched disorder. In the present section we consider the disorder-free model given by Eq. (5). The effect of quenched disorder will be discussed in sec. 5.2.

### 3.1 Crystalline order parameter and the free-energy functional

Our disorder-free models given by the Hamiltonian Eq. (3), Eq. (4) and Eq. (5) have the following global symmetries. In sec. 2.1 we introduced two types of spins: Ising and continuous spins. In the cases of Ising spins $S_i^\mu = \pm 1$, and for even $p$, the system has a global symmetry with respect to $S^\mu \to -S^\mu$ for each component $\mu$. Such symmetry is absent for the cases of odd $p$. In the cases of continuous spins $S_i^\mu \in \mathcal{R}$, and for $p = 2$, the system has a global continuous symmetry with respect to rotations of spins in the $M$-dimensional spin space. The continuous rotational symmetry is lost for $p > 2$ [1] and the residual global symmetries become just the same as those in the Ising cases.

To be more specific, suppose that the system has a ferromagnetic ground state $\mathbf{S}_i = (1, 1, \ldots, 1)$ for $\forall i$. This is achieved for example by choosing the linear potential $V(x) = Jx$ with $J > 0$ in Eq. (12). Because of the global symmetries mentioned above, there can be other equivalent ground states, e. g. $\mathbf{S}_i = (-1, -1, \ldots, -1)$ for $\forall i$ (for even $p$). In order to study the possibility of spontaneous symmetry breaking which select one ground state out of the equivalent ones (if they exist), we may apply an external field of strength $h > 0$ parallel to the ground state $(1, 1, \ldots, 1)$,

$$\beta H = -\sum_{\blacksquare=1}^{N_\blacksquare} \beta V\left(r_\blacksquare\right) - h \sum_{i=1}^{N} \sum_{\mu=1}^{M} S_i^\mu \tag{22}$$

and examine the behavior of an order parameter,

$$m = \lim_{h\to 0} \lim_{N\to\infty} \frac{1}{NM} \sum_{i=1}^{N} \sum_{\mu=1}^{M} \langle S_i^\mu \rangle_h, \tag{23}$$

where $\langle \cdots \rangle_h$ represents a thermal average in the presence of the symmetry breaking field. The standard procedure to analyze the problem is as follows. 1) One first construct a free-energy $-\beta G(h)$ in the presence of the field $h$ and then perform a Legendre transform to obtain $-\beta F(m) = -\beta G(m) + Nmh$ and then 2) seek for a solution $m$ which solves $\partial_m(-\beta F(m)) = h = 0$.

In addition, since we are considering to take the limit of large number of components $M \to \infty$, we may also define a *local order parameter*,

$$m_i = \lim_{h\to 0} \lim_{M\to\infty} \frac{1}{M} \sum_{\mu=1}^{M} \langle S_i^\mu \rangle_h \qquad (i = 1, 2, \ldots, N). \tag{24}$$

Let us emphasize again that $M$ scales independently of $N$, which will bring about important consequences below.

---

[1] Suppose that a rotation is defined by a $M \times M$ matrix $\hat{R}$, which is orthogonal $\hat{R}^t = \hat{R}^{-1}$. Vectors are transformed by the rotation as $S^\mu \to \sum_{\nu=1}^{M} R^{\mu\nu} S^\nu$. For instance, it can be easily checked that $\sum_{\mu=1}^{M} S_i^\mu S_j^\mu$ remains invariant under the rotation but $\sum_{\mu=1}^{M} S_i^\mu S_j^\mu S_k^\mu S_l^\mu$ does not.

### 3.1.1 Spin trace

The above discussion motivates us to introduce an identity,

$$1 = \int_{-\infty}^{\infty} dm_i \delta(Mm_i - \sum_{\mu=1}^{M} S_i^{\mu}) = M \int_{-i\infty}^{i\infty} \frac{dh_i}{2\pi} \int_{-\infty}^{\infty} dm_i e^{h_i(Mm_i - \sum_{\mu=1}^{M} S_i^{\mu})} \qquad (i = 1, 2, \ldots, N). \tag{25}$$

The integration over $h$ and $m$ corresponds to the steps 1) and 2) mentioned above. Using the identity spin traces can be expressed formally in the $M \to \infty$ limit as,

$$\text{Tr}_S \cdots = M \int_{-i\infty}^{i\infty} \frac{dh}{2\pi} \int_{-\infty}^{\infty} dm \exp\left[ Mhm + \ln \text{Tr}_S e^{-h\sum_{\mu=1}^{M} S^{\mu}} \right] \cdots = \int_{-\infty}^{\infty} dm e^{Ms_{\text{ent}}(m)} \prod_{\mu} \langle \cdots \rangle_{\mu}. \tag{26}$$

Here the integration over $h$ can be done (formally) by the saddle point method in the limit $M \to \infty$. The saddle point $h^*(m)$ is given by the saddle point equation,

$$m = \left. \frac{\text{Tr}_S e^{-h^* \sum_{\mu=1}^{M} S^{\mu}} S^{\mu}}{\text{Tr}_S e^{-h^* \sum_{\mu=1}^{M} S^{\mu}}} \right|_{h^* = h^*(m)} \tag{27}$$

and we find,

$$s_{\text{ent}}(m) = h^* m + \ln \text{Tr}_S e^{-h^* \sum_{\mu=1}^{M} S^{\mu}} \qquad \langle \cdots \rangle_{\mu} = \frac{\text{Tr}_S e^{-h^* S^{\mu}} \cdots}{\text{Tr}_S e^{-h^* S^{\mu}}}, \tag{28}$$

where $h^* = h^*(m)$ is given by Eq. (27). Using Eq. (26) we find, for example,

$$\text{Tr}_S S^{\mu} = \int_{-\infty}^{\infty} dm e^{Ms_{\text{ent}}(m)} m. \tag{29}$$

More specifically, by taking the spin traces explicitly we obtain the following expressions for the Ising and continuous spin systems,

- **Ising spin**

  We find using Eq. (9),

  $$s_{\text{ent}}(m) = -m \tanh^{-1}(m) + \ln[2 \cosh(\tanh^{-1} m)], \qquad h^* = \tanh^{-1}(m)). \tag{30}$$

- **Continuous spin**

  We find using Eq. (10),

  $$s_{\text{ent}}(m) = \frac{1}{2} + \frac{1}{2} \ln(2\pi) + \frac{1}{2} \ln(1 - m^2),$$
  $$\langle \cdots \rangle_{\mu} \equiv \frac{\int_{-\infty}^{\infty} dS^{\mu} e^{-\lambda^*(S^{\mu})^2 - h^* S^{\mu}} \cdots}{\int_{-\infty}^{\infty} dS^{\mu} e^{-\lambda^*(S^{\mu})^2 - h^* S^{\mu}}}, \qquad h^* = -2m\lambda^*, \qquad \lambda^* = \frac{1}{2(1 - m^2)}. \tag{31}$$

  Here we performed the integrations $\int_{-\infty}^{\infty} dS^{\mu}$ assuming $\lambda > 0$. Then we performed integrations over $\lambda$ and $h$ by the saddle point method.

In Eq. (26) we notice that different spin components $\mu$ are decoupled in the average $\prod_\mu \langle \dots \rangle_\mu$. Then we obtain the following cumulant expansion which will become very useful in the following,

$$\ln \langle e^{\frac{1}{\sqrt{M}} \sum_{\mu=1}^{M} A_\mu} \rangle = \frac{1}{\sqrt{M}} \sum_{\mu=1}^{M} \langle A_\mu \rangle + \frac{1}{2!} \left( \frac{1}{\sqrt{M}} \right)^2 \sum_{\mu=1}^{M} (\langle A_\mu^2 \rangle - \langle A_\mu \rangle^2)$$

$$+ \frac{1}{3!} \left( \frac{1}{\sqrt{M}} \right)^3 \sum_{\mu=1}^{M} (\langle A_\mu^3 \rangle - 3 \langle A_\mu^2 \rangle^2 \langle A_\mu \rangle + 2 \langle A_\mu \rangle^3)$$

$$+ \frac{1}{4!} \left( \frac{1}{\sqrt{M}} \right)^4 \sum_{\mu=1}^{M} (\langle A_\mu^4 \rangle - 4 \langle A_\mu^3 \rangle \langle A_\mu \rangle - 3 \langle A_\mu^2 \rangle^2 + 12 \langle A_\mu^2 \rangle \langle A_\mu \rangle^2 - 6 \langle A_\mu \rangle^3) + \cdots \quad (32)$$

Here we just used the fact that $\langle A^\mu A^\nu \rangle = \langle A^\mu \rangle \langle A^\nu \rangle$ holds for $\mu \neq \nu$.

### 3.1.2 Evaluation of the free-energy

Using Eq. (26), Eq. (29) and the cumulant expansion Eq. (32) we find,

$$\prod_i \mathrm{Tr}_{\mathbf{S}_i} \exp \left[ \frac{1}{\sqrt{M}} \sum_{\mu=1}^{M} \sum_{\blacksquare} (-i\kappa_\blacksquare) S_{1(\blacksquare)}^\mu S_{2(\blacksquare)}^\mu \cdots S_{p(\blacksquare)}^\mu \right]$$

$$\xrightarrow{M \to \infty} \left( \prod_i \int_{-\infty}^{\infty} dm_i \right) e^{M \sum_i s_{\mathrm{ent}}(m_i)} \exp \left[ \sqrt{M} \sum_{\blacksquare} (-i\kappa_\blacksquare) m_{1(\blacksquare)} m_{2(\blacksquare)} \cdots m_{p(\blacksquare)} \right]. \quad (33)$$

Now the partition function given by Eq. (8) can be rewritten formally in the $M \to \infty$ limit as,

$$Z = \left( \prod_i \int_{-\infty}^{\infty} dm_i \right) e^{M \sum_i s_{\mathrm{ent}}(m_i)} \prod_\blacksquare \left\{ \int_{-\infty}^{\infty} \frac{d\kappa_\blacksquare}{2\pi} Z_{\kappa_\blacksquare} e^{i\kappa_\blacksquare \delta} \right\} \times \cdots$$

$$\times \exp \left[ \sqrt{M} \sum_\blacksquare (-i\kappa_\blacksquare) m_{1(\blacksquare)} m_{2(\blacksquare)} \cdots m_{p(\blacksquare)} \right]$$

$$= \left( \prod_{i=1}^{N} \int_{-\infty}^{\infty} dm_i \right) e^{NMs(\{m_i\})}, \quad (34)$$

where we defined

$$s(\{m_i\}) = \frac{1}{N} \sum_{i=1}^{N} \left\{ s_{\mathrm{ent}}(m_i) - \frac{1}{pM} \sum_{\blacksquare \in \partial_i} \beta V(\delta - \sqrt{M} m_{1(\blacksquare)} m_{2(\blacksquare)} \cdots m_{p(\blacksquare)}) \right\}, \quad (35)$$

where $\partial_i$ represents the set of interactions which involve $\mathbf{S}_i$. Now we are left with the integrations over $m_i$s in Eq. (34) which can be done by the saddle point method in the $M \to \infty$ limit. The saddle point equation reads as,

$$0 = \left. \frac{\partial s(\{m_i\})}{\partial m_j} \right|_{\{m_i = m_i^*\}} \qquad \text{for} \qquad (j = 1, 2, \dots, N). \quad (36)$$

Since the system is regular and every vertex is exactly equivalent to each other in our system, it is natural to expect a uniform solution $m_i^* = m$ for $\forall i$. Moreover, since each spin

is connected to $c = \alpha M$ neighbors which is a large number, one can show that the effect of possible site-to-site fluctuation of $m_i$ can be neglected in the $M \to \infty$ limit. In addition, possible small fluctuations of the coordination number $c$ can also be neglected for the same reason.

We obtain the free-energy associated with such a uniform saddle point as,

$$-\beta \frac{F}{NM} = s(m), \tag{37}$$

with

$$s(m) \equiv s_{\text{ent}}(m) - \frac{\alpha}{p}\beta V(\delta - \sqrt{M}m^p), \tag{38}$$

where $m$ must satisfy the saddle point equation

$$0 = \frac{ds(m)}{dm}. \tag{39}$$

It is also required to satisfy the stability condition,

$$d^2 s(m)/dm^2 \le 0. \tag{40}$$

## 3.2 Possibilities of the crystalline states

So far we have just considered a ferromagnetic phase with the ground state $\mathbf{S}_i = (1, 1, \ldots, 1)$ for $\forall i$ but we can also consider other crystalline states. For example, suppose that there is a crystalline ground state in which the spin configuration can be represented by some configuration $S_i^\mu = (\sigma_i)_0 \in (-1, 1)$ which is independent of $\mu$ but depends on the vertex $i$. Just for simplicity we are limiting ourselves to the cases that the ground state configuration have the collinear spin structure, i. e. spin configuration on different vertexes are either parallel or anti-parallel to each other. The ferromagnetic case discussed in sec. 3.1 corresponds to $(\sigma_i)_0 = 1$ for $\forall i$. Then it is useful to perform a gauge transformation

$$S_i^\mu \to \tilde{S}_u^\mu \equiv \sigma_i S_i^\mu. \tag{41}$$

The crystalline order parameter $m$ can be defined again as Eq. (23) but replacing the spins $S$ by the gauge transformed ones $\tilde{S}$. Here the spins $S_i^\mu$ can be either the Ising type or continuous type. The gauge transformation defined above does not change the character of the spins including the spin normalization Eq. (1) which reads $\sum_{\mu=1}^{M}(S_i^\mu)^2 = M$.

By the same gauge transformation the gap given by Eq. (4) (with $X_\blacksquare^\mu = 1$) is transformed to

$$r_\blacksquare \to \tilde{r}_\blacksquare = \delta - \frac{\eta_\blacksquare}{\sqrt{M}} \sum_\mu \tilde{S}_{1(\blacksquare)}^\mu \tilde{S}_{2(\blacksquare)}^\mu \cdots \tilde{S}_{p(\blacksquare)}^\mu, \tag{42}$$

where we defined

$$\eta_\blacksquare = \sigma_{1(\blacksquare)}\sigma_{2(\blacksquare)}\cdots\sigma_{p(\blacksquare)}. \tag{43}$$

The variable $\eta_\blacksquare$ takes $\pm 1$ values. For simplicity we limit ourselves to the ground states such that it is a constant $\eta_\blacksquare = \eta$ for all the interactions $\blacksquare$. Then the results in the previous section given by Eq. (37)-Eq. (40) holds just by changing the argument of the potential as:

$$V(\delta - \sqrt{M}m^p) \to V(\delta - \eta\sqrt{M}m^p). \tag{44}$$

The simplest example is $p = 2$ model with the linear potential $V(x) = Jx$ but with $J < 0$. Obviously the ground state is the anti-ferromagnetic one: $\sigma_\blacksquare$ alternates the sign across each of the interactions (note that we are considering tree-like lattices with no loops). In this case $\eta_\blacksquare = \sigma_{1(\blacksquare)}\sigma_{2(\blacksquare)} = -1$ so that it becomes essentially the same as a ferromagnetic model with $J > 0$ after the gauge transformation.

## 3.3 Crystalline transitions and possibility of super-cooling

The saddle point equation given by Eq. (39) and the stability condition given by Eq. (40) becomes, including the factor $\eta = \pm 1$ discussed above as the following:

- **Ising spin**

  The saddle point equation given by Eq. (30) becomes,

  $$m = \tanh\left[\eta\alpha\sqrt{M}m^{p-1}\beta V'(\delta - \eta\sqrt{M}m^p)\right]. \tag{45}$$

  The stability condition becomes,

  $$\begin{aligned}\frac{d^2 s(m)}{dm^2} &= -\frac{1}{1-m^2} + \eta\alpha\sqrt{M}(p-1)m^{p-2}\beta V'(\delta - \eta\sqrt{M}m^p) \\ &\quad -\alpha p M m^{2(p-1)}\beta V''(\delta - \eta\sqrt{M}m^p) < 0.\end{aligned} \tag{46}$$

- **Continuous spin**

  The saddle point equation given by Eq. (31) becomes,

  $$0 = m\left(-\frac{1}{1-m^2} + \eta\alpha\sqrt{M}\beta V'(\delta - \eta\sqrt{M}m^p)m^{p-2}\right). \tag{47}$$

  The stability condition becomes,

  $$\begin{aligned}\frac{d^2 s(m)}{dm^2} &= -\frac{1}{1-m^2} + \frac{2m^2}{(1-m^2)^2} + \eta\alpha\sqrt{M}(p-1)m^{p-2}\beta V'(\delta - \eta\sqrt{M}m^p) \\ &\quad -\alpha p M m^{2(p-1)}\beta V''(\delta - \eta\sqrt{M}m^p) < 0.\end{aligned} \tag{48}$$

It can be seen that the paramagnetic solution $m = 0$ always verify the saddle point equations. We are especially interested with the possibility that the paramagnetic state with $m = 0$ remains as a metastable state after the crystalline transitions take place so that glass transitions within the paramagnetic phase become possible.

- $p = 2$ case:

  - if $|V'(\delta)| > 0$, a 2nd order ferromagnetic transition takes place at a critical temperature

    $$k_{\mathrm{B}} T_c = \alpha\sqrt{M}|V'(\delta)| \tag{49}$$

    below which the paramagnetic solution $m = 0$ becomes unstable and the ferromagnetic or anti-ferromagnetic order with $|m| > 0$ emerges continuously. If $V'(\delta)$ is positive (negative) the ordering is ferromagnetic (anti-ferromagnetic) and we should choose $\eta = 1$ ($\eta = -1$). Since the paramagnetic state $m = 0$ is unstable below $T_c$, supper-cooled paramagnetic state is absent and thus glass transitions is not possible without suppressing crystalline states by quenched disorder.

  - If $V'(\delta) = 0$, there will be no ferromagnetic nor anti-ferromagnetic phase transitions at finite temperatures. The $m = 0$ solution remains stable at all finite temperatures,

    $$\left.\frac{d^2 s(m)}{dm^2}\right|_{m=0} = -1 < 0. \tag{50}$$

    This is a very interesting situation where the crystallization is totally suppressed opening possibilities of glass transitions without quenched disorder.

- $p > 2$ case:

  The paramagnetic solution $m = 0$ remains locally stable at all temperatures in the sense of Eq. (50). Thus in this case supercooled paramagnetic state exist opening possibilities of glass transitions without quenched disorder.

### 3.3.1 Linear potential: $p$-spin ferromagnetic model

As a simplest example let us consider the case of the linear potential $V(x) = Jx$ where $J > 0$, which means $V'(\delta) > 0$. It is a ferromagnetic model so we choose $\eta = 1$. The saddle point equation becomes for the Ising spins,

$$\text{(Ising)} \qquad m = \tanh\left[\alpha\sqrt{M}\beta Jm^{p-1}\right] \tag{51}$$

and for the continuous spins,

$$\text{(Continuous)} \qquad 0 = m\left(-\frac{1}{1-m^2} + \alpha\sqrt{M}\beta Jm^{p-2}\right). \tag{52}$$

We see $m = 0$ always verifies the saddle point equations as it should,

- For $p = 2$ case, a 2nd order ferromagnetic transition takes place at a critical temperature $k_{\mathrm{B}}T_c/J = \alpha\sqrt{M}$. Supper-cooled paramagnet and thus glass transitions without quenched disorder are not possible as discussed above.

- For $p > 2$, a 1st order ferromagnetic transition take place at $k_{\mathrm{B}}T_c/J = O(\alpha\sqrt{M})$. On the other hand the paramagnetic state $m = 0$ remains locally stable at all temperatures as discussed above.

  Quite interestingly in [27] a $p = 3$ Ising ferromagnet with $M = 1$ component was studied via cavity method and Monte Carlo simulations and the supercooled paramagnetic state and the glass transition were discovered. Our result is consistent with this observation.

- In the $p \to \infty$ limit with the Ising spins, the exact solution can be easily obtained. The saddle point equation given by Eq. (51) admits only $m = \pm 1$ except for $m = 0$. The paramagnetic free-energy (free-entropy) is obtained as $s(m = 0) = \ln 2$ while that for the ferromagnetic phase is obtained as $s(m = 1) = \gamma\beta J\sqrt{M}$. Here we introduced a parameter

$$\gamma = \frac{\alpha}{p}. \tag{53}$$

  Let us consider the $p \to \infty$ limit with $\gamma$ fixed. Then we easily see that a 1st order ferromagnetic phase transition takes place at

$$k_{\mathrm{B}}T_c/J = \gamma\sqrt{M}/\ln 2. \tag{54}$$

  In the next section 4 we will find that the system becomes equivalent to the random energy model (REM) [4] by excluding the ferromagnetic state.

### 3.3.2 Non-linear potentials with flatness

If the potential $V(x)$ has a flat part where $V'(x) = 0$, it tends to suppress crystallization and thus enhances the possibility to realize glass transitions inside the paramagnetic phase.

The simplest example may be the quadratic potential,

$$V(x) = \frac{\epsilon}{2}\frac{x^2}{2} \to V'(x) = \epsilon x. \tag{55}$$

Thus for $p = 2$ and $\delta = 0$, the system should remain paramagnetic at all finite temperatures.

More interesting case is the soft/hard core potential given by Eq. (14) which is completely flat for $x > 0$,

$$V(x) = \epsilon x^2\theta(-x) \to V'(x) = 2\epsilon x\theta(-x). \tag{56}$$

Let us consider again $p = 2$ case. For $\delta > 0$, $V'(\delta) = 0$ so that the system is paramagnetic at all finite temperatures. On the other hand for $\delta < 0$, $V'(x) < 0$ anti-ferromagnetic phase emerges via 2nd order transition. Then we choose $\eta = -1$. The transition temperature given by Eq. (49) is found as,

$$k_B T_c(\delta) = 2\alpha\sqrt{M}\epsilon(-\delta)\theta(-\delta). \tag{57}$$

In the hardcore limit $\epsilon \to \infty$, the anti-ferromagnetic transition takes place as $\delta \to 0^+$.

## 4 Self-generated randomness: connection to the random energy model in $p \to \infty$ ferromagnetic Ising model with $M \to \infty$

Let us start looking for possible glass transitions within the supercooled paramagnetic phase. In this section we study the ferromagnetic $p$-spin $M$-component Ising model, with the linear potential $V(x) = Jx$ with $J > 0$ discussed in sec 3.3.1. There we have seen that supercooled paramagnetic states with $m = 0$ exist for $p > 2$ below the ferromagnetic transition temperatures $k_T T_c/J \sim O(\alpha\sqrt{M})$. In the following we will find that the system becomes essentially identical to the random energy model (REM) [4] in the $p \to \infty$ limit as far as the supercooled states are concerned. This proves the existence of the self-generated randomness.

The hamiltonian is given by

$$H_{\{S\}} = -\frac{J}{\sqrt{M}} \sum_{\blacksquare} \sum_{\mu=1}^{M} S^{\mu}_{1(\blacksquare)} S^{\mu}_{2(\blacksquare)} \cdots S^{\mu}_{p(\blacksquare)} \qquad S^{\mu}_i \in (-1, 1), \tag{58}$$

with $J > 0$. Here we are especially interested with the $p \to \infty$ limit with $\gamma = \alpha/p$ introduced in Eq. (53) fixed. As we discussed in sec 3.3 it exhibits a ferromagnetic phase transition at $k_B T_c = \gamma J\sqrt{M}/\ln 2$.

We examine the distribution of the energies of the disorder-free model performing a similar analysis done for the $p$-spin Ising spinglass model *with* quenched disorder in the original work by Derrida [4]. To this end let us first introduce a flat average over the $2^{NM}$ spin configurations,

$$\langle \ldots \rangle_S \equiv \frac{\prod_{i=1}^{N} \prod_{\mu=1}^{M} \sum_{S^{\mu}_i = \pm 1} \cdots}{2^{NM}}.$$

Then the distribution of energy among all configurations is obtained as,

$$
\begin{aligned}
P(E) &= \langle \delta(E - H_{\{S\}}) \rangle_S = \int_{-\infty}^{\infty} \frac{d\kappa}{2\pi} e^{i\kappa E} \left\langle \exp\left[ i\kappa \frac{J}{\sqrt{M}} \sum_{\blacksquare} \sum_{\mu=1}^{M} S^{\mu}_{1(\blacksquare)} S^{\mu}_{2(\blacksquare)} \cdots S^{\mu}_{p(\blacksquare)} \right] \right\rangle_S \\
&\xrightarrow[M \to \infty]{} \frac{e^{-\frac{E^2}{2N_{\blacksquare}J^2}}}{\sqrt{2\pi N_{\blacksquare}J^2}},
\end{aligned}
\tag{59}
$$

where $N_{\blacksquare} = NM(\alpha/p)$ is the number of interactions given by Eq. (2). Here we evaluated the expectation value $\langle \ldots \rangle_S$ by performing expansion in power series of $1/\sqrt{M}$ (see Eq. (32) for

the cumulant expansion),

$$\ln \left\langle \exp\left[ i\kappa \frac{J}{\sqrt{M}} \sum_{\blacksquare} \sum_{\mu=1}^{M} S_{1(\blacksquare)}^{\mu} S_{2(\blacksquare)}^{\mu} \cdots S_{p(\blacksquare)}^{\mu} \right] \right\rangle_{S}$$

$$= \ln\left[ 1 + \frac{(i\kappa)^2}{2!}\left(\frac{J}{\sqrt{M}}\right)^2 \left\langle \left(\sum_{\blacksquare}\sum_{\mu} S_{1(\blacksquare)}^{\mu} S_{2(\blacksquare)}^{\mu} \cdots S_{p(\blacksquare)}^{\mu}\right)^2 \right\rangle_{S}\right.$$

$$\left. + \frac{(i\kappa)^4}{4!}\left(\frac{J}{\sqrt{M}}\right)^4 \left\langle \left(\sum_{\blacksquare}\sum_{\mu} S_{1(\blacksquare)}^{\mu} S_{2(\blacksquare)}^{\mu} \cdots S_{p(\blacksquare)}^{\mu}\right)^4 \right\rangle_{S} + \dots \right]$$

$$= \frac{(i\kappa)^2}{2!}\left(\frac{J}{\sqrt{M}}\right)^2 N_{\blacksquare}M + \frac{\kappa^4}{4!}\left(\frac{J}{\sqrt{M}}\right)^4 \left(N_{\blacksquare}M\right) + \dots \xrightarrow[M\to\infty]{} -\frac{\kappa^2}{2}N_{\blacksquare}J^2. \tag{60}$$

Here $N_{\blacksquare} = NM(\alpha/p) = NM\gamma$ (see Eq. (2)), since we take the $p \to \infty$ limit with fixed $\gamma$ as defined in Eq. (53).

Next let us examine simultaneous distribution of energy $E_1$ associated with an arbitrary chosen spin configuration $(\mathbf{S}_1, \mathbf{S}_2, \dots, \mathbf{S}_N)$ and the energy $E_2$ of another configuration $(\mathbf{S}'_1, \mathbf{S}'_2, \dots, \mathbf{S}'_N)$. Here the latter is created from the former by flipping, say according to a deterministic rule, a fraction $(1-q)/2$ with $0 < q < 1$ of the elements of the former. In other words the overlap between the two configurations is $q = (1/(NM)) \sum_{i=1}^{N} \sum_{\mu=1}^{M} S_i^{\mu}(S')_i^{\mu}$. We find,

$$P(E, E') = \langle \delta(E - H_{\{S\}}))\delta(E' - H_{\{S'\}}))\rangle_S$$

$$= \int_{-\infty}^{\infty}\int_{-\infty}^{\infty} \frac{d\kappa}{2\pi}\frac{d\kappa'}{2\pi} e^{i\kappa E + i\kappa' E'} \left\langle \exp\left[ \frac{J}{\sqrt{M}}\sum_{\blacksquare}\sum_{\mu=1}^{M}\left( i\kappa S_{1(\blacksquare)}^{\mu} S_{2(\blacksquare)}^{\mu} \cdots S_{p(\blacksquare)}^{\mu} \right. \right. \right.$$

$$\left. \left. \left. + i\kappa'(S')_{1(\blacksquare)}^{\mu}(S')_{2(\blacksquare)}^{\mu} \cdots (S')_{p(\blacksquare)}^{\mu} \right) \right] \right\rangle_S$$

$$\xrightarrow[M\to\infty]{} \frac{1}{2}\frac{1}{\sqrt{N_{\blacksquare}\pi J^2 A_+}}\frac{1}{\sqrt{N_{\blacksquare}\pi J^2 A_-}} \exp\left[ -\frac{E_+^2}{NJ^2 A_+} - \frac{E_-^2}{NJ^2 A_-} \right], \tag{61}$$

with $A_{\pm} = \frac{1}{2}[1 \pm q^p]$ and $E_{\pm} = (E \pm E')/2$. Here we evaluated the expectation value $\langle\dots\rangle_S$ by performing expansion in power series of $1/\sqrt{M}$,

$$\lim_{M\to\infty} \ln \left\langle \exp\left[ \frac{J}{\sqrt{M}}\sum_{\blacksquare}\sum_{\mu=1}^{M}\left( i\kappa S_{1(\blacksquare)}^{\mu} S_{2(\blacksquare)}^{\mu} \cdots S_{p(\blacksquare)}^{\mu} + i\kappa'(S')_{1(\blacksquare)}^{\mu}(S')_{2(\blacksquare)}^{\mu} \cdots (S')_{p(\blacksquare)}^{\mu} \right) \right] \right\rangle_S$$

$$= \lim_{M\to\infty} \frac{1}{2}\left(\frac{J}{\sqrt{M}}\right)^2 \left\langle \left\{\sum_{\blacksquare}\sum_{\mu}\left( i\kappa S_{1(\blacksquare)}^{\mu} S_{2(\blacksquare)}^{\mu} \cdots S_{p(\blacksquare)}^{\mu} + i\kappa'(S')_{1(\blacksquare)}^{\mu}(S')_{2(\blacksquare)}^{\mu} \cdots (S')_{p(\blacksquare)}^{\mu} \right)\right\}^2 \right\rangle_S$$

$$= -\frac{1}{2}[\kappa^2 + (\kappa')^2 + 2\kappa\kappa' q^p]N_{\blacksquare}J^2. \tag{62}$$

Here we realize that in the $p \to \infty$ limit, the distribution function decouples,

$$P(E, E') \xrightarrow[p\to\infty]{} P(E)P(E'), \tag{63}$$

because $0 < q < 1$.

The above observations imply that the present ferromagnetic system without any quenched disorder behaves essentially as a REM in the $p \to \infty$ limit: over the majority of the $2^{NM}$ spin configurations , excluding the negligible fraction of the spin configurations close to the ferromagnetic ground state, it is as if each microscopic states is assigned a random energy drawn from a Gaussian distribution with 0 mean and variance $\sqrt{\gamma} J$. Here we notice that all the exact results of the standard version of the REM [4], which corresponds to $\gamma = 1/2$, can be used in the present system just by replacing $J$ of the standard model by $\sqrt{2\gamma} J$. Then we readily find that the system exhibit the Kauzmann transition, i. e. ideal static glass transition at,

$$k_B T_K / J = \sqrt{\gamma} / \sqrt{2 \ln 2} \tag{64}$$

within the supercooled paramagnetic states. At temperatures below $T_K$, the internal energy becomes stuck at $E/NM = -\sqrt{2\gamma \ln 2} J$ among the disordered states, while the ferromagnetic ground state energy is given by $E_g / NM = -\sqrt{M} \gamma J$.

Readers would have noticed that the derivation of the REM discussed above is quite similar to the standard procedure to prove the central limit theorem (CLT). The underlying reason can be traced back to the tree-like structure of our system.

# 5 Replicated system

In this section we setup a formalism to study the glass transitions using the replica method. We first develop a free-energy functional of the disorder-free model given by Eq. (5). Then we also consider the model with quenched disorder given by Eq. (6).

## 5.1 Disorder free model

We consider a system of replicas $a = 1, 2, \ldots, n$ of the disorder-free model given by Eq. (5),

$$H = -\sum_{a=1}^{n} \sum_{\blacksquare=1}^{N_\blacksquare} V\left(r_\blacksquare^a\right), \tag{65}$$

where

$$r_\blacksquare^a = \delta - \frac{1}{\sqrt{M}} \sum_{\mu=1}^{M} (S^a)_{1(\blacksquare)}^\mu (S^a)_{2(\blacksquare)}^\mu \cdots (S^a)_{p(\blacksquare)}^\mu. \tag{66}$$

The free-energy of the replicated system can be expressed as

$$-\beta F = \log Z = \partial_n Z^n \big|_{n=0},$$

with the replicated partition function

$$Z^n = \left( \prod_{i=1}^{N} \prod_{a=1}^{n} \text{Tr}_{\mathbf{S}_{i,a}} \right) \prod_\blacksquare e^{-\sum_{a=1}^{n} \beta V(r_\blacksquare^a)} = \prod_{a=1}^{n} \prod_\blacksquare \left\{ \int_{-\infty}^{\infty} \frac{d\kappa_\blacksquare}{2\pi} Z_{\kappa_{\blacksquare,a}} e^{i\kappa_{\blacksquare,a}\delta} \right\}$$

$$\left( \prod_a \prod_i \text{Tr}_{\mathbf{S}_{i,a}} \right) \exp \left[ \frac{1}{\sqrt{M}} \sum_{a=1}^{n} \sum_{\mu=1}^{M} \sum_\blacksquare (-i\kappa_{\blacksquare,a})(S^a)_{1(\blacksquare)}^\mu (S^a)_{2(\blacksquare)}^\mu \cdots (S^a)_{p(\blacksquare)}^\mu \right], \tag{67}$$

where $\text{Tr}_i^a$ represents a trace over the spin space in replica $a$.

In order to detect the spontaneous glass transition, we can follow steps analogous to the one we took in sec. 3.1 for the crystalline (ferromagnetic) transition. Namely we can explicitly

break the replica symmetry as [46],

$$\beta H = -\sum_{a=1}^{n}\sum_{\blacksquare=1}^{N_{\blacksquare}}\beta V\left(r_{\blacksquare}^{a}\right) - \sum_{a<b}\epsilon_{ab}\sum_{i}\sum_{\mu}(S^{a})_{i}^{\mu}(S^{b})_{i}^{\mu} \tag{68}$$

and study the behavior of the glass order parameter matrix $\hat{Q}$

$$Q_{ab} = \lim_{\epsilon_{ab}\to 0}\lim_{N\to\infty}\frac{1}{NM}\sum_{i=1}^{N}\sum_{\mu=1}^{M}\langle(S^{a})_{i}^{\mu}(S^{b})_{i}^{\mu}\rangle_{\epsilon}. \tag{69}$$

Here $\langle\ldots\rangle_{\epsilon}$ represents the thermal average in the presence of the symmetry breaking field $\epsilon_{ab}$. Although Eq. (69) is meat for $a \neq b$, it is convenient to extend it to include the diagonal elements

$$Q_{aa} = 1 \tag{70}$$

to reflect the spin normalization Eq. (1).

Just as the case of ferromagnetic transition discussed in sec. 3.1, we can consider the following steps to analyze the problem: 1) One first construct a free-energy $-\beta G(\hat{\epsilon})$ in the presence of the field $\hat{\epsilon}$ and then perform a Legendre transform to obtain $-\beta F(\hat{Q}) = -\beta G(\hat{Q}) + N\sum_{a<b}\epsilon_{ab}Q_{ab}$ and then 2) seek for a solution which solves $\partial_{Q_{ab}}(-\beta F(\hat{Q})) = \epsilon_{ab} = 0$.

Since we are considering the $M \to \infty$ limit we may also define a *local glass order parameter*,

$$(Q_{i})_{ab} = \lim_{\epsilon_{ab}\to 0}\lim_{M\to\infty}\frac{1}{M}\sum_{\mu=1}^{M}\langle(S^{a})_{i}^{\mu}(S^{b})_{i}^{\mu}\rangle_{\epsilon} \qquad (i=1,2,\ldots,N). \tag{71}$$

### 5.1.1 Spin trace

The above discussion motivate us to introduce an identity,

$$1 = \int_{-\infty}^{\infty}d(Q_{ab})_{i}\delta\left(M(Q_{ab})_{i} - \sum_{\mu=1}^{M}(S^{a})_{i}^{\mu}(S^{b})_{i}^{\mu}\right) \tag{72}$$

$$= M\int_{-i\infty}^{i\infty}\frac{d(\epsilon_{ab})_{i}}{2\pi}\int_{-\infty}^{\infty}d(Q_{ab})_{i}e^{(\epsilon_{ab})_{i}(M(Q_{ab})_{i}-\sum_{\mu=1}^{M}(S^{a})_{i}^{\mu}(S^{b})_{i}^{\mu})} \qquad (i=1,2,\ldots,N)$$

for $a < b$. The integration over $\epsilon_{ab}$ and $Q_{ab}$ corresponds to the steps 1) and 2) mentioned above. Using the latters, spin traces can be expressed formally in the $M \to \infty$ limit as,

$$\prod_{c=1}^{n}\mathrm{Tr}_{\mathbf{S}_{c}}\cdots =$$

$$M\left(\prod_{a<b}\int_{-\infty}^{\infty}\frac{d\epsilon_{ab}}{2\pi}\int_{-\infty}^{\infty}d(Q_{ab})\right) \quad \exp\left[M\sum_{a<b}\epsilon_{ab}Q_{ab} + \ln\prod_{c=1}^{n}\mathrm{Tr}_{\mathbf{S}_{c}}e^{-\epsilon_{ab}\sum_{\mu=1}^{M}(S^{a})^{\mu}(S^{b})^{\mu}}\right]\cdots$$

$$= \left(\prod_{a<b}\int_{-\infty}^{\infty}d(Q_{ab})\right)e^{Ms_{\mathrm{ent}}[\hat{Q}]}\prod_{\mu}\langle\cdots\rangle_{\mu}. \tag{73}$$

Here the integration over $\epsilon_{ab}$ can be done by the saddle point method in $M \to \infty$. The saddle point equation which determines the saddle point $\epsilon_{ab}^{*}(\hat{Q})$ is given by,

$$Q_{ab} = \left.\frac{\prod_{c}\mathrm{Tr}_{S^{c}}e^{-\sum_{a<b}\epsilon_{ab}S^{a}S^{b}}S^{a}S^{b}}{\prod_{c}\mathrm{Tr}_{S^{c}}e^{-\sum_{a<b}\epsilon_{ab}S^{a}S^{b}}}\right|_{\epsilon_{ab}=\epsilon_{ab}^{*}(\hat{Q})} \tag{74}$$

and we find,

$$s_{\text{ent}}[\hat{Q}] = \sum_{a<b} \epsilon^*_{ab} Q_{ab} + \ln \prod_{c=1}^{n} \text{Tr}_{S^c} e^{-\sum_{a<b} \epsilon^*_{ab} S^a S^b} \qquad \langle \cdots \rangle_\mu = \frac{\prod_{c=1}^{n} \text{Tr}_{S^c} e^{-\sum_{a<b} \epsilon^*_{ab} (S^a)^\mu (S^b)^\mu} \cdots}{\prod_{c=1}^{n} \text{Tr}_{S^c} e^{-\sum_{a<b} \epsilon^*_{ab} (S^a)^\mu (S^b)^\mu}}$$

(75)

where $\epsilon^*_{ab} = \epsilon^*_{ab}(\hat{Q})$ determined by Eq. (74). Using Eq. (73) we find, for example,

$$\prod_c \text{Tr}_{\mathbf{S}_c} (S^a)^\mu = 0 \qquad \prod_c \text{Tr}_{\mathbf{S}_c} (S^a)^\mu (S^b)^\mu = \int_{-\infty}^{\infty} d(Q_{ab}) e^{M s_{\text{ent}}[\hat{Q}]} Q_{ab}.$$

(76)

More precisely, by taking the spin traces we obtain the following expressions for the Ising and continuous spin systems,

- **Continuous spin**: We find using Eq. (10) and introducing $\epsilon_{aa} = \lambda_a$ and $Q_{aa} = 1$ (spin normalization, see Eq. (70)),

$$s_{\text{ent}}[\hat{Q}] = \sum_{a,b} \frac{1}{2} \epsilon^*_{ab} Q_{ab} + \ln \sqrt{\frac{(2\pi)^n}{\det(\hat{\epsilon})^*}} = \frac{n}{2} + \frac{n}{2} \ln(2\pi) + \frac{1}{2} \ln \det \hat{Q}$$

$$\langle \cdots \rangle_\mu \equiv \frac{\int_{-\infty}^{\infty} dS^\mu e^{-\frac{1}{2} \sum_{a,b} \epsilon^*_{ab} (S^a)^\mu (S^b)^\mu} \cdots}{\int_{-\infty}^{\infty} dS^\mu e^{-\frac{1}{2} \sum_{a,b} \epsilon^*_{ab} (S^a)^\mu (S^b)^\mu}}.$$

(77)

Here we have performed integration over $\epsilon_{ab}$ by the saddle point method which yields a saddle point

$$\hat{\epsilon}^* = \hat{Q}^{-1}.$$

(78)

- **Ising spin**: We find using Eq. (9),

$$s_{\text{ent}}[\hat{Q}] = \sum_{a \neq b} \frac{1}{2} \epsilon^*_{ab} Q_{ab} + \ln e^{-\sum_{a<b} \epsilon^*_{ab} \frac{\partial^2}{\partial h_a \partial h_b}} \prod_a 2\cosh(h_a) \Bigg|_{\{h_a=0\}}.$$

(79)

Here we performed the spin trace formally as

$$\text{Tr}_{\mathbf{S}_c} e^{-\sum_{a<b} \epsilon_{ab} S^a S^b} = \text{Tr}_{\mathbf{S}_c} e^{-\sum_{a<b} \epsilon_{ab} \frac{\partial^2}{\partial h_a \partial h_b}} e^{\sum_a h_a S^a} \Bigg|_{\{h_a=0\}}$$

$$= e^{-\sum_{a<b} \epsilon_{ab} \frac{\partial^2}{\partial h_a \partial h_b}} \prod_a 2\cosh(h_a) \Bigg|_{\{h_a=0\}}.$$

For the integration over $\epsilon_{ab}$, the saddle point $\epsilon^*_{ab} = \epsilon^*_{ab}[\hat{Q}]$ is obtained formally as,

$$Q_{ab} = -\frac{\delta}{\delta \epsilon_{ab}} \ln e^{-\sum_{a<b} \epsilon_{ab} \frac{\partial^2}{\partial h_a \partial h_b}} \prod_a 2\cosh(h_a) \Bigg|_{\{h_a=0\}} \Bigg|_{\epsilon_{ab}=\epsilon^*_{ab}[\hat{Q}]}.$$

(80)

### 5.1.2 Evaluation of the free-energy

In Eq. (73) we notice again that different spin components $\mu$ are decoupled in the average $\prod_\mu \langle \cdots \rangle_\mu$. Then we can evaluate the spin trace in the replicated partition function given by

Eq. (67) in the $M \to \infty$ limit using the cumulant expansion given by Eq. (32) and Eq. (76) as,

$$\prod_a \prod_i \text{Tr}_{S_{i,a}} \exp\left[\frac{1}{\sqrt{M}} \sum_a \sum_{\mu=1}^M \sum_\blacksquare (-i\kappa_\blacksquare^a)(S^a)_{1(\blacksquare)}^\mu (S^a)_{2(\blacksquare)}^\mu \cdots (S^a)_{p(\blacksquare)}^\mu\right] \tag{81}$$

$$\xrightarrow[M\to\infty]{} \left(\prod_i \prod_{a<b} \int_{-\infty}^\infty d(\hat{Q}_i)_{ab}\right) e^{M \sum_i s_{\text{ent}}[\hat{Q}_i]} \times \cdots$$

$$\cdots \times \exp\left[\sum_\blacksquare \sum_{a,b} \frac{(i\kappa_a)(i\kappa_b)}{2}(Q_{1(\blacksquare)})_{ab}(Q_{2(\blacksquare)})_{ab}\cdots(Q_{p(\blacksquare)})_{ab}\right].$$

In the exponent we assume $Q_{aa} = 1$ (see Eq. (70)) for the diagonal terms. The above expression is a crucial result because it reveals the self-generated randomness in our 'disorder-free' model.

Collecting the above results, the partition function given by Eq. (67) can be rewritten formally in the $M \to \infty$ limit as,

$$\begin{aligned} Z &= \left(\prod_i \prod_{a<b} \int_{-\infty}^\infty d(\hat{Q}_i)_{ab}\right) e^{M \sum_i s_{\text{ent}}[\hat{Q}_i]} \prod_{\blacksquare,a} \left\{\int_{-\infty}^\infty \frac{d\kappa_\blacksquare^a}{2\pi} Z_{\kappa_\blacksquare^a} e^{i\kappa_\blacksquare^a \delta}\right\} \times \cdots \\ &\qquad \cdots \times \exp\left[\sum_\blacksquare \sum_{a,b} \frac{(i\kappa_a)(i\kappa_b)}{2}(Q_{1(\blacksquare)})_{ab}(Q_{2(\blacksquare)})_{ab}\cdots(Q_{p(\blacksquare)})_{ab}\right] \\ &= \left(\prod_i \prod_{a<b} \int_{-\infty}^\infty d(\hat{Q}_i)_{ab}\right) e^{NM s[(\hat{Q}_i)]}, \end{aligned} \tag{82}$$

where we defined

$$s[\hat{Q}_i] = \frac{1}{N} \sum_{i=1}^N \left\{ s_{\text{ent}}[\hat{Q}_i] + \frac{1}{pM} \sum_{\blacksquare \in \partial_i} e^{\frac{1}{2}\sum_{a,b} \frac{\partial^2}{\partial h_a \partial h_b}(Q_{1(\blacksquare)})_{ab}(Q_{2(\blacksquare)})_{ab}\cdots(Q_{p(\blacksquare)})_{ab}} \prod_{a=1}^n e^{-\beta V(\delta + h_a)}\bigg|_{\{h_a=0\}} \right\}. \tag{83}$$

To derive the last expression we used Eq. (11) and performed integrations by parts (see Eq. (355) for the same calculation.).

The integrations over each $(Q_i)_{ab}$ can be performed in the $M \to \infty$ limit by the saddle point method. The saddle point equation read,

$$0 = \frac{\partial s[\hat{Q}_i]}{\partial (Q_j)_{ab}}\bigg|_{[\hat{Q}_i=\hat{Q}_i^*]} \qquad \text{for} \qquad (j=1,2,\ldots,N). \tag{84}$$

Now repeating the same argument as in sec.3.1.2, we can assume that the equations admit a uniform solution $\hat{Q}_i^* = \hat{Q}$ for $\forall i$ since in our system every vertex is equivalent to each other. As the result we obtain the free-energy associated with such a saddle point as

$$-\beta \frac{F}{NM} = \partial_n s_n[\hat{Q}]\bigg|_{n=0}, \tag{85}$$

with

$$s_n[\hat{Q}] \equiv s_{\text{ent}}[\hat{Q}] - \frac{\alpha}{p}\mathcal{F}_{\text{int}}[\hat{Q}],$$

$$-\mathcal{F}_{\text{int}}[\hat{Q}] \equiv \left. \exp\left( \frac{1}{2}\sum_{a,b=1}^{n} Q_{ab}^p \frac{\partial^2}{\partial h_a \partial h_b} \right) \prod_{a=1}^{n} e^{-\beta V(\delta+h_a)} \right|_{h_a=0} m \qquad (86)$$

where $-\mathcal{F}_{\text{int}}$ represents the interaction part of the free-energy (free-entropy). Importantly $Q_{ab}$ must satisfy the saddle point equations,

$$0 = \frac{\partial s[\hat{Q}]}{\partial Q_{ab}}. \qquad (87)$$

It is also required to satisfy the stability condition, i.e. the eigen values of the the Hessian matrix,

$$H_{(ab),(cd)} \equiv -\frac{\partial^2 s_n[\hat{Q}]}{\partial Q_{ab} \partial Q_{cd}} \qquad (88)$$

in the $n \to 0$ limit, must be non negative.

The exact free-energy functional given by Eq. (85) with Eq. (86) can be derived also using a density functional approach as we show in appendix A for the case of continuous spins. It is done closely following the strategy used in the recent replicated liquid theory for hardsphere glass in the large dimensional limit [14–16].

## 5.2 Interpolation between disorder-free and completely disordered model

In the present paper we are most concerned with systems where the disorder is self-generated. However it is very instructive to consider also the model with quenched disorder given by Eq. (6),

$$r_{\blacksquare} = \delta - \frac{1}{\sqrt{M}}\sum_{\mu=1}^{M}\left[ \frac{\lambda}{\sqrt{M}} + \sqrt{1-\left(\frac{\lambda}{\sqrt{M}}\right)^2}\,\xi_{\blacksquare}^{\mu} \right] S_{1(\blacksquare)}^{\mu} S_{2(\blacksquare)}^{\mu}\cdots S_{p(\blacksquare)}^{\mu} \qquad \left( 0 \le \frac{\lambda}{\sqrt{M}} \le 1 \right). \qquad (89)$$

Here $\xi_{\blacksquare}^{\mu}$ is a random variable with Gaussian distribution with zero mean and unit variance. With this parametrization, we have a continuous interpolation between the disorder-free model $\lambda/\sqrt{M} = 1$ and completely disordered, spinglass model $\lambda/\sqrt{M} = 0$.

Analysis of this model is useful for the following reasons.

- We can show that the self-generated disorder and the quenched disorder act additively.

- In the disorder-free model given by Eq. (5), which corresponds to $\lambda/\sqrt{M} = 1$, the energy scale of glass transition and crystalline transitions are widely separated. For instance the $p$-spin ferromagnetic Ising model in the $p \to \infty$ limit exhibit a ferromagnetic transition at $k_B T_c/J = O(\sqrt{M})$ given by Eq. (54) while the glass transition takes place in the supercooled paramagnetic sector at $k_B T_K/J = O(1)$ given by Eq. (64). Actually the fact that $T_K$ is lower than $T_c$ is natural by itself but they become too much separated in the disorder-free model in the $M \to \infty$ limit. With the choice of the disordered model given by Eq. (6) we can bring the energy scales of two transitions much closer. This is achieved in two steps: (i) reduce the interaction energy scale down to order $O(1/\sqrt{M})$ of the original 'disorder-free' model (ii) then add additional quenched disorder such that the effective energy scale for the glass transition is brought back to the original level of

$O(1)$. Bringing the crystallization and glass transition temperatures closer, it becomes easier to investigate competitions between liquid, glass and crystalline phases. Note that similar treatments for the energy scales are considered in standard spinglass models with ferromagnetic biases [47].

Averaging over the quenched disorder of the replicated partition function given by Eq. (67), we obtain,

$$\overline{Z^n}^{\xi} = \left( \prod_i \prod_{a=1}^{n} \text{Tr}_i^a \right) \prod_{\blacksquare} \left\{ \prod_a \int_{-\infty}^{\infty} \frac{d\kappa_{\blacksquare,a}}{2\pi} Z_{\kappa_{\blacksquare,a}} e^{i\kappa_{\blacksquare,a}\delta} e^{\mathcal{L}_{\blacksquare}} \right\} \tag{90}$$

where the overline denotes the average over the disorder and we introduced,

$$\begin{aligned} \mathcal{L}_{\blacksquare} \equiv{} & \lambda \sum_{a=1}^{n} (-i\kappa_{\blacksquare,a}) \frac{1}{M} \sum_{\mu=1}^{M} (S^a)^{\mu}_{1(\blacksquare)} (S^a)^{\mu}_{2(\blacksquare)} \cdots (S^a)^{\mu}_{p(\blacksquare)} + \dots \\ & \cdots + \sum_{a,b=1}^{n} \frac{(-i\kappa_{\blacksquare,a})(-i\kappa_{\blacksquare,b})}{2} \left( 1 - \left( \frac{\lambda}{\sqrt{M}} \right)^2 \right) \times \dots \\ & \cdots \times \frac{1}{M} \sum_{\mu=1}^{M} (S^a)^{\mu}_{1(\blacksquare)} (S^b)^{\mu}_{1(\blacksquare)} \cdots (S^a)^{\mu}_{p(\blacksquare)} (S^b)^{\mu}_{p(\blacksquare)}. \end{aligned} \tag{91}$$

For the order parameters we may consider both the glass order parameters given by Eq. (69) and the crystalline one given by Eq. (23). Since we are considering the $M \to \infty$ limit we naturally define the following set of local order parameters,

$$(Q_i)_{ab} = \lim_{\epsilon_{ab} \to 0} \lim_{M \to \infty} \frac{1}{M} \sum_{\mu=1}^{M} \langle (S^a)^{\mu}_i (S^b)^{\mu}_i \rangle_{\epsilon} \tag{92}$$

$$m_a = \lim_{h_a \to 0} \lim_{M \to \infty} \frac{1}{M} \sum_{\mu=1}^{M} \langle (S^a)^{\mu}_i \rangle_h \tag{93}$$

and the corresponding identities,

$$\begin{aligned} 1 ={} & \int_{-\infty}^{\infty} d(Q_{ab})_i \delta \left( M(Q_{ab})_i - \sum_{\mu=1}^{M} (S^a)^{\mu}_i (S^b)^{\mu}_i \right) \\ ={} & M \int_{-i\infty}^{i\infty} \frac{d(\epsilon_{ab})_i}{2\pi} \int_{-\infty}^{\infty} d(Q_{ab})_i e^{(\epsilon_{ab})_i (M(Q_{ab})_i - \sum_{\mu=1}^{M}(S^a)^{\mu}_i (S^b)^{\mu}_i)} \end{aligned} \tag{94}$$

$$1 = \int_{-\infty}^{\infty} d(m_a)_i \delta(M(m_a)_i - \sum_{\mu=1}^{M} S^{\mu}_i) = M \int_{-i\infty}^{i\infty} \frac{dh_i^a}{2\pi} \int_{-\infty}^{\infty} d(m_a)_i e^{h_i^a (M(m_a)_i - \sum_{\mu=1}^{M} S^{\mu}_i)}. \tag{95}$$

### 5.2.1 Spin trace

Using the identities shown above spin traces can be expressed in the $M \to \infty$ limit as,

$$\prod_{c=1}^{n} \text{Tr}_{\mathbf{S}_c} \cdots = \left( \prod_{a<b} \int_{-\infty}^{\infty} dQ_{ab} \right) \left( \prod_a \int_{-\infty}^{\infty} dm_a \right) e^{M s_{\text{ent}}[\hat{Q}, \hat{m}]} \prod_{\mu} \langle \cdots \rangle_{\mu}, \tag{96}$$

where

$$s_{\text{ent}}[\hat{Q},\hat{m}] = \sum_{a<b} \epsilon^*_{ab}Q_{ab} + \sum_a h^*_a m_a + \ln \prod_{c=1}^n \text{Tr}_{S^c} e^{-\sum_{a<b}\epsilon^*_{ab}S^a S^b - \sum_a h^*_a S^a}$$

$$\langle\cdots\rangle_\mu = \frac{\prod_{c=1}^n \text{Tr}_{\mathbf{S}_c} e^{\mathcal{L}^{\text{ent}}_\mu}\cdots}{\prod_{c=1}^n \text{Tr}_{\mathbf{S}_c} e^{\mathcal{L}^{\text{ent}}_\mu}} \qquad \mathcal{L}^{\text{ent}}_\mu = -\sum_{a<b}\epsilon^*_{ab}(S^a)^\mu (S^b)^\mu - \sum_a h^*_a (S^a)^\mu$$

$$Q_{ab} = \frac{\prod_c \text{Tr}_{\mathbf{S}_c} e^{\mathcal{L}^{\text{ent}}} S^a S^b}{\prod_c \text{Tr}_{\mathbf{S}_c} e^{\mathcal{L}^{\text{ent}}}} \qquad m_a = \frac{\prod_c \text{Tr}_{\mathbf{S}_c} e^{\mathcal{L}^{\text{ent}}} S^a}{\prod_c \text{Tr}_{\mathbf{S}_c} e^{\mathcal{L}^{\text{ent}}}}. \tag{97}$$

Here we introduced a short hand notation $\hat{m} = (m_1, m_2, \ldots, m_n)$. The last equations are the saddle point equations for the integrations over $\epsilon_{ab}$ and $h_a$ which fix the saddle points $\epsilon^*_{ab} = \epsilon^*_{ab}[\hat{Q},\hat{m}]$ and $h^*_a = h^*_a[\hat{Q},\hat{m}]$. Using Eq. (96) we find, for example,

$$\prod_c \text{Tr}_{\mathbf{S}_c}(S^a)^\mu = \left(\prod_{a<b}\int_{-\infty}^\infty d(Q_{ab})\int_{-\infty}^\infty \prod_a d(m_a)\right) e^{Ms_{\text{ent}}[\hat{Q},\hat{m}]} m_a$$

$$\prod_c \text{Tr}_{\mathbf{S}_c}(S^a)^\mu_i (S^b)^\mu = \left(\prod_{a<b}\int_{-\infty}^\infty d(Q_{ab})\right)\left(\prod_a \int_{-\infty}^\infty d(m_a)\right) e^{Ms_{\text{ent}}[\hat{Q},\hat{m}]} Q_{ab}. \tag{98}$$

More precisely, by taking the spin traces we obtain the following expressions for the Ising and continuous spin systems,

- **Continuous spin**: We find similarly to Eq. (77),

$$\begin{aligned} s_{\text{ent}}[\hat{Q},\hat{m}] &= \frac{1}{2}\sum_{a,b}\epsilon^*_{ab}Q_{ab} + \sum_a h^*_a m_a + \ln\sqrt{\frac{(2\pi)^n}{\det\epsilon^*}} + \frac{1}{2}\sum_{ab}h^*_a(\epsilon^*)^{-1}_{ab}h^*_b \\ &= \frac{n}{2} + \frac{n}{2}\ln(2\pi) + \frac{1}{2}\ln\det(\hat{Q}-\hat{m}^T\hat{m}) \end{aligned} \tag{99}$$

and

$$\langle\cdots\rangle_\mu \equiv \frac{\int_{-\infty}^\infty dS^\mu e^{\mathcal{L}_\mu}\cdots}{\int_{-\infty}^\infty dS^\mu e^{\mathcal{L}_\mu}} \qquad \mathcal{L}_\mu = -\frac{1}{2}\sum_{a,b}\epsilon^*_{ab}(S^a)^\mu (S^b)^\mu - \sum_a h^*_a (S^a)^\mu. \tag{100}$$

Here we have performed integration over $\epsilon_{ab}$ and $h_a$ by the saddle point method which yield a saddle point

$$(\hat{\epsilon}^*)^{-1}_{ab} = Q_{ab} - m_a m_b \qquad h^*_a = -\sum_b \epsilon^*_{ab} m_b. \tag{101}$$

- **Ising spin**: We find similarly to Eq. (75),

$$s_{\text{ent}}[\hat{Q},\hat{m}] = \sum_{a\neq b}\frac{1}{2}\epsilon^*_{ab}Q_{ab} + \sum_a h^*_a m_a + \ln e^{-\sum_{a<b}\epsilon^*_{ab}\frac{\partial^2}{\partial h_a \partial h_b}}\prod_a 2\cosh(h_a)\Bigg|_{\{h_a = h^*_a\}}, \tag{102}$$

where we performed the spin trace formally as

$$\text{Tr}_{\mathbf{S}_c} e^{-\sum_{a<b}\epsilon_{ab}S^a S^b - \sum_a h_a S^a} = e^{-\sum_{a<b}\epsilon_{ab}\frac{\partial^2}{\partial h_a \partial h_b}}\prod_a 2\cosh(h_a). \tag{103}$$

For the integration over $\epsilon_{ab}$ and $h_a$, the saddle points $\epsilon^*_{ab} = \epsilon^*_{ab}[\hat{Q}, \hat{m}]$ and $h^*_a = h^*_a[\hat{Q}, \hat{m}]$ are obtained formally as,

$$
\begin{aligned}
Q_{ab} &= -\frac{\delta}{\delta \epsilon_{ab}} \ln e^{-\sum_{a<b} \epsilon_{ab} \frac{\partial^2}{\partial h_a \partial h_b}} \prod_a 2\cosh(h_a) \Bigg|_{\{\epsilon_{ab}=\epsilon^*_{ab}[\hat{Q},\hat{m}], h_a=h^*_a[\hat{Q},\hat{m}]\}} \\
m_a &= -\frac{\delta}{\delta h_a} \ln e^{-\sum_{a<b} \epsilon_{ab} \frac{\partial^2}{\partial h_a \partial h_b}} \prod_a 2\cosh(h_a) \Bigg|_{\{\epsilon_{ab}=\epsilon^*_{ab}[\hat{Q},\hat{m}], h_a=h^*_a[\hat{Q},\hat{m}]\}} .
\end{aligned}
\tag{104}
$$

### 5.2.2 Evaluation of the free-energy

In Eq. (96) we notice again that different spin components $\mu$ are decoupled in the average $\prod_\mu \langle \dots \rangle_\mu$. Then we can evaluate the spin trace in the replicated partition function given by Eq. (67) in the $M \to \infty$ limit using the cumulant expansion given by Eq. (32) and Eq. (98) as,

$$
\prod_a \prod_i \mathrm{Tr}_{\mathbf{S}_{i,a}} \prod_{\blacksquare} e^{\mathcal{L}_{\blacksquare}} \xrightarrow[M\to\infty]{} \left( \prod_i \prod_{a<b} \int_{-\infty}^{\infty} d(Q_i)_{ab} \prod_a \int_{-\infty}^{\infty} d(m_i)_a \right) \times \dots
$$

$$
\dots \times \exp\left[ \lambda \sum_{a=1}^n (-i\kappa_{\blacksquare,a})(m_{1(\blacksquare)})_a (m_{2(\blacksquare)})_a \cdots (m_{p(\blacksquare)})_a + \dots \right.
\tag{105}
$$

$$
\left. \dots + \sum_{a,b=1}^n \frac{(-i\kappa_{\blacksquare,a})(-i\kappa_{\blacksquare,b})}{2}(Q_{1(\blacksquare)})_{ab}(Q_{2(\blacksquare)})_{ab} \cdots (Q_{p(\blacksquare)})_{ab} \right].
$$

Here we point out that the last term in the exponent of the last equation is the result of a summation of the contributions of two different different kinds of disorder: (1) quenched disorder of amplitude $1 - (\lambda/\sqrt{M})^2$ (see the 2nd term in Eq. (92)) (2) self-generated disorder of amplitude $(\lambda/\sqrt{M})^2$. Now it is clear that parametrization given by Eq. (6) is chosen such that the energy scale of the glass transition does not change between the disorder-free limit ($\lambda/\sqrt{M} = 1$) and completely disordered limit $\lambda/\sqrt{M} = 0$.

Collecting the above results, the disorder averaged replicated partition function given by Eq. (90) can be rewritten formally in the $M \to \infty$ limit as,

$$
\overline{Z^n}^\xi = \prod_i \left( \prod_{a<b} \int_{-\infty}^{\infty} d(Q_i)_{ab} \prod_a \int_{-\infty}^{\infty} d(m_i)_a \right) e^{NMs[\hat{Q}_i, \hat{m}_i]},
\tag{106}
$$

with

$$
s[\hat{Q}_i, \hat{m}_i] = \frac{1}{N} \sum_{i=1}^N \left\{ s_{\text{ent}}[\hat{Q}_i, \hat{m}_i] + \frac{1}{pM} \sum_{\blacksquare \in \partial_i} \frac{1}{2} \sum_{a,b} \frac{\partial^2}{\partial h_a \partial h_b}(Q_{1(\blacksquare)})_{ab} \cdots (Q_{p(\blacksquare)})_{ab} \times \dots \right.
$$

$$
\left. \dots \times \prod_{a=1}^n e^{-\beta V(\delta - \lambda(m_{1(\blacksquare)})_a \dots (m_{p(\blacksquare)})_a + h_a)} \Bigg|_{\{h_a=0\}} \right\}
\tag{107}
$$

The integrations over each $(Q_i)_{ab}$ and $(m_i)_a$ can be performed in the $M \to \infty$ limit by the

saddle point method. The saddle point equations read,

$$0 = \frac{\partial s[\hat{Q}_i, \hat{m}_i]}{\partial (Q_j)_{ab}}\Bigg|_{[\hat{Q}_i = \hat{Q}_i^*, \hat{m}_i = \hat{m}_i^*]} \qquad \text{for} \qquad (j = 1, 2, \ldots, N)$$

$$0 = \frac{\partial s[\hat{Q}_i, \hat{m}_i]}{\partial (m_j)_a}\Bigg|_{[\hat{Q}_i = \hat{Q}_i^*, \hat{m}_i = \hat{m}_i^*]} \qquad \text{for} \qquad (j = 1, 2, \ldots, N). \tag{108}$$

After the average over the quenched disorder every vertex has become again identical to each other. Then we can repeat the same argument as in sec. 3.1.2 and assume uniform solutions: $\hat{Q}_i^* = \hat{Q}$ and $\hat{m}_i^* = \hat{m}$ for $\forall i$. As the result we obtain the free-energy associated with such a saddle point as

$$-\beta \frac{F}{NM} = \partial_n s_n[\hat{Q}, \hat{m}]\Big|_{n=0}, \tag{109}$$

with

$$s_n[\hat{Q}] \equiv s_{\text{ent}}[\hat{Q}, \hat{m}] - \frac{\alpha}{p}\mathcal{F}_{\text{int}}[\hat{Q}, \hat{m}]$$

$$-\mathcal{F}_{\text{int}}[\hat{Q}, \hat{m}] \equiv \exp\left(\frac{1}{2}\sum_{a,b=1}^{n}(\hat{Q})_{ab}^p \frac{\partial^2}{\partial h_a \partial h_b}\right)\prod_{a=1}^{n} e^{-\beta V(\delta - \lambda m_a^p + h_a)}\Bigg|_{h_a=0}, \tag{110}$$

where $Q_{ab}$ and $m_a$ must satisfy the saddle point equations

$$0 = \frac{\partial s[\hat{Q}, \hat{m}]}{\partial Q_{ab}}, \qquad 0 = \frac{\partial s[\hat{Q}, \hat{m}]}{\partial m_a}. \tag{111}$$

It is also required to satisfy the stability condition, i.e. the eigenvalues of the Hessian matrix,

$$H_{(ab),(cd)} \equiv -\frac{\partial^2 s_n[\hat{Q}, \hat{m}]}{\partial Q_{ab}\partial Q_{cd}}, \qquad H_{(ab),c} \equiv -\frac{\partial^2 s_n[\hat{Q}, \hat{m}]}{\partial Q_{ab}\partial m_c}, \qquad H_{a,b} \equiv -\frac{\partial^2 s_n[\hat{Q}, \hat{m}]}{\partial m_a \partial m_b}. \tag{112}$$

in the $n \to 0$ limit, must be non negative.

Finally let us note again that the disorder-free model can be recovered by choosing $\lambda/\sqrt{M} = 1$ in the above expressions. For the disorder-free model discussed in previous sections, we gave free-energy functional in terms of the crystalline order parameter $m$ in Eq. (38) and that in terms of the glass order parameter $Q_{ab}$ in Eq. (86) separately just to simplify the presentations. In any case here we now have complete free-energy functional where both the crystalline and glass order parameters are present.

## 5.3 Stability against crystallization

Given the complete free-energy functional in term of both the crystalline and glass order parameters, we can now investigate the stability of glassy phases against crystallization extending the analysis in sec. 3.3 which was limited to the liquid phase. Here we limit ourselves with a glassy phase without crystalline order parameter $m = 0$ and do not consider possible 'glassy crystals' with $m > 0$. First we note that,

$$H_{(ab),c}\Big|_{\{m_a=0\}} = -\frac{\partial^2 s_n[\hat{Q}, \hat{m}]}{\partial Q_{ab}\partial m_c}\Bigg|_{\{m_a=0\}} = 0 \tag{113}$$

holds. This can be checked by taking the derivatives explicitly. For the entropic part we find,

$$\frac{\partial^2 s_{\text{ent}}[\hat{Q},\hat{m}]}{\partial Q_{ab}\partial m_c}\bigg|_{\{m_a=0\}} = \frac{\partial}{\partial Q_{ab}}h^*[\hat{Q},\hat{m}]\bigg|_{\{m_a=0\}} = 0 \tag{114}$$

The last equation follows from the last equation of Eq. (97) which implies that $m_a = 0$ requires $h_a^* = 0$. For the interaction part of the free-energy we find,

$$-\frac{\partial^2 \mathcal{F}_{\text{int}}[\hat{Q},\hat{m}]}{\partial Q_{ab}\partial m_c}\bigg|_{\{m_a=0\}} = pQ_{ab}^{p-1}\frac{\partial^2}{\partial h_a\partial h_b}e^{\frac{1}{2}\sum_{a,b=1}^n(\hat{Q})_{ab}^p\frac{\partial^2}{\partial h_a\partial h_b}}(-\lambda pm_c^{p-1})\times\ldots \tag{115}$$

$$\cdots\times(-\beta V'(\delta-\lambda m_c^p+h_c))\prod_{a=1}^n e^{-\beta V(\delta-\lambda m_a^p+h_a)}\bigg|_{\{h_a,m_a=0\}}$$

$$= 0. \tag{116}$$

The last equation holds for $p > 1$. Thus Eq. (113) must hold.

Then the local stability of the glassy phase with $m = 0$ against crystallization is solely determined by the matrix,

$$H_{a,b} = -\frac{\partial^2 s_n[\hat{Q},\hat{m}]}{\partial m_a\partial m_b}\bigg|_{\{m_a=0\}}$$

$$= Q_{ab}^{-1} - \delta_{ab}\lambda\alpha(p-1)\, m_a^{p-2}\big|_{m_a=0}\, e^{\frac{1}{2}\sum_{a,b=1}^n(\hat{Q})_{ab}^p\frac{\partial^2}{\partial h_a\partial h_b}}\times\ldots \tag{117}$$

$$\cdots\times(-\beta V'(\delta+h_a))\prod_{a=1}^n e^{-\beta V(\delta+h_a)}\bigg|_{h_a=0},$$

where we assumed $p > 1$ and we used

$$-\frac{\partial^2 s_{\text{ent}}[\hat{Q},\hat{m}]}{\partial m_a\partial m_b}\bigg|_{\{m_a=0\}} = Q_{ab}^{-1}. \tag{118}$$

which follows from Eq. (97).[2]

In the liquid phase we have $m_a = 0$ and $Q_{ab} = \delta_{ab}$ and thus $Q_{ab}^{-1} = \delta_{ab}$. due to spin normalization (see Eq. (70)). There one can check that non-negativeness of the eigenvalues of the matrix Eq. (117) in $n \to 0$ limit becomes equivalent to the stability conditions Eq. (46) and Eq. (48) of the paramagnetic solution $m = 0$ as it should.

Here we see that the 2nd term on the r.h.s of Eq. (117), which is due to the interaction part of the free-energy, vanishes in two cases (i) $p > 2$ (ii) $p = 2$ with non-linear potential with the flatness $V'(\delta) = 0$. Remarkably in these cases the matrix becomes independent of $\lambda$ and its the eigen values are simply the inverse of the eigen values of the matrix $Q_{ab}$. We expect the latters are positive for physical solutions.[3] This is very interesting because including the regime of large enough $\lambda$, especially the disorder-free case $\lambda = \sqrt{M}$, where we naturally expect crystalline order as the true equilibrium phase, paramagnetic phase $m = 0$ (for which $Q_{ab} = \delta_{ab}$ due to the spin normalization) and also the glassy phase with $m = 0$ (for which $Q_{ab} \neq \delta_{ab}$) remains locally stable against crystallization for the two cases: (i) and (ii).

---

[2]Taking $\partial_{m_b}$ on both sides of the last equation of Eq. (97) we find $\delta_{ab} = \sum_c \frac{\partial h_c^*}{\partial m_b}(Q_{ab} - m_a m_b)$ thus $-\frac{\partial^2 s_{\text{ent}}[\hat{Q},\hat{m}]}{\partial m_a\partial m_b}\big|_{\{m_a=0\}} = \frac{\partial h_a^*}{\partial m_b}\big|_{\{m_a=0\}} = Q_{ab}^{-1}$.

[3] For instance note that in the case of the continuous spins the entropic part of the free-energy has a term $\ln\det(\hat{Q})$ with $m_a = 0$ (see Eq. (99)). Thus the eigenvalues of the matrix $Q_{ab}$ are needed to be positive.

Contrarily, in the case of $p = 2$ without the flat potential the $m = 0$ solution cannot be stable against crystallization if $\lambda/\sqrt{M}$ is finite in $M \to \infty$ limit, including the in particular the disorder-free case $\lambda = \sqrt{M}$. Thus in these cases the quenched disorder is necessary to realize the glass phases. The range of the stability of the liquid $Q_{ab} = \delta_{ab}$ and glass phase $Q_{ab} \neq \delta_{ab}$ with a given $\lambda$ must be examined analyzing the eigenvalues of Eq. (117).

Finally we note that the situation can change in systems with finite connectivity. The supercooled paramagnetic phase can disappear for sufficiently large $\lambda$. In the context of statistical inference problems this is an important issue because one has to find the hidden crystalline state (ground truth) in the immense sea of wrong solutions (glasses) [48].

## 6 Linear potential: connection to the standard $p$-spin Ising/ spherical spinglass models and the random energy model

Let us discuss here the simplest case, the linear potential given by Eq. (12) which reads,

$$V(x) = Jx. \tag{119}$$

The interaction part of the free-energy given by Eq. (109) becomes,

$$-\mathcal{F}_{\text{int}}[\hat{Q}, \hat{m}] = e^{\lambda(\beta J)\sum_a m_a^p + \frac{(\beta J)^2}{2}\sum_{a,b=1}^n Q_{ab}^p} \tag{120}$$

then we find

$$-\frac{\beta F}{NM} = \partial_n s_n|_{n=0} \qquad s_n = s_{\text{ent}}(\hat{Q}, \hat{m}) + \lambda(\beta J)\sum_a m_a^p + \frac{(\beta J)^2}{2}\frac{\alpha}{p}\sum_{a,b} Q_{ab}^p \tag{121}$$

where $s_{\text{ent}}(\hat{Q}, \hat{m})$ is given in Eq. (99) for the continuous spin case and Eq. (102) for the Ising case. This is exactly the same as those of the standard $p$-spin Ising/spherical spinglass models with $M = 1$ but with global couplings [4, 9, 49] by choosing $\alpha/p = 1/2$. Note that such a correspondence has been known for the case of a $p = 2$ continuous spin model with global coupling [39].

Let us summarize below some important known results of the $p$-spin spinglass models. The case $p = 2$ with the Ising spin corresponds to the SK (Sherrington-Kirkpatrick) model [47]. It exhibits a continuous phase transition from the paramagnetic to the spinglass phase accompanying the continuous replica symmetry breaking (RSB) [50] while the spherical version of it exhibits a continuous phase transition but without RSB [51]. The SK model is the standard mean-field model for spinglasses [22, 52]. On the other hand $p > 2$ system exhibit 1 step RSB [4, 5, 9] with a discontinuous transition from the paramagnetic to the spinglass phase. These models show the essence of the glass phenomenology such as the dynamical and static glass transitions so that they are regarded as prototypical theoretical model to capture the physics of structural glasses [2, 8, 11, 53]. Among the latter models those with the Ising spin exhibit yet another glass transition to enter the continuous RSB phase at lower temperatures [54]. In the $p \to \infty$ limit of the Ising case, the random energy model is recovered [4].

We emphasize that the result given by Eq. (121) is valid also in the disorder-free limit $\lambda/\sqrt{M} = 1$. Indeed in sec. 4 we have shown that the random energy model is recovered in the disorder-free limit. Thus the disorder-free model have sufficient amount of self-generated disorder to realize glass transitions. The supercooled paramagnetic state and the glass phase which emerge there are stable against crystallization for $p > 2$ as discussed in sec. 5.3. In the case $p = 2$, however, we have to invoke the quenched disorder to suppress the crystalline (ferromagnetic) states. (This amount to yield nothing but the SK model for the Ising spins and spherical SK model for the continuous spins mentioned above.)

# 7 Replica symmetric (RS) ansatz

For the rest of the present paper we study glass transitions of our model, which emerge within the supercooled paramagnetic phase with no crystalline order $m = 0$. And we limit our selves with the continuous spin models for the rest of the present paper. In the present section and in the next section we derive some generic results within the replica symmetric (RS) and replica symmetry breaking (RSB) ansatz. We apply these schemes to systems with non-linear potentials in later sections.

Our starting point is the free-energy functional given by Eq. (85) which reads,

$$-\beta f[\hat{Q}] = -\beta \frac{F[\hat{Q}]}{NM} = \partial_n s_n[\hat{Q}]\Big|_{n=0} \tag{122}$$

with Eq. (86) which reads,

$$
\begin{aligned}
s_n[\hat{Q}] &\equiv \frac{1}{2}\ln\det(\hat{Q}) - \frac{\alpha}{p}\mathcal{F}_{\text{int}}[\hat{Q}] \\
-\mathcal{F}_{\text{int}}[\hat{Q}] &\equiv \exp\left(\frac{1}{2}\sum_{a,b=1}^{n} Q_{ab}^p \frac{\partial^2}{\partial h_a \partial h_b}\right) \prod_{a=1}^{n} e^{-\beta V(\delta + h_a)}\Bigg|_{\{h_a = 0\}} .
\end{aligned}
\tag{123}
$$

Here $\mathcal{F}_{\text{int}}$ represents the interaction part of the free-energy. For the entropic part in Eq. (86) we used the expression given by Eq. (77) and we omitted irrelevant constants $\frac{n}{2} + \frac{n}{2}\ln(2\pi)$ for simplicity.

The pressure given by Eq. (15) can be computed as

$$\Pi = -\frac{p}{\alpha}\frac{\partial \beta f[\hat{Q}]}{\partial \delta} = -\frac{\partial}{\partial \delta}\,\partial_n \mathcal{F}_{\text{int}}\big|_{n=0} \tag{124}$$

and similarly the distribution function of the gap given by Eq. (17) as

$$g(r) = -\frac{p}{\alpha}\frac{\delta \beta f[\hat{Q}]}{\delta(-\beta V(r))} = -\frac{\delta}{\delta(-\beta V(r))}\,\partial_n \mathcal{F}_{\text{int}}\big|_{n=0}. \tag{125}$$

Before passing let us recall the discussion in sec 3.3.2 and sec. 5.3 that the supercooled paramagnetic $m = 0$ states and glassy states of the model is locally stable against crystallization if $p > 2$. But for the case $p = 2$ we must have the flatness $V'(\delta) = 0$ or quenched disorder. Although we may not mention these points often in the following, we must keep these in our minds.

## 7.1 Formulation

In the replica symmetric (RS) ansatz we assume the following form of the overlap matrix parametrized by a single parameter $q$,

$$Q_{ab}^{\text{RS}} = (1-q)\delta_{ab} + q. \tag{126}$$

Note that diagonal part $Q_{aa} = 1$ reflects the spin normalization.

### 7.1.1 Free-energy

First let us compute the free-energy given by Eq. (122) -Eq. (123) within the RS ansatz. Using Eq. (126) we find,

$$\ln\det\hat{Q}^{\text{RS}} = \ln[1 + (n-1)q] + (n-1)\ln(1-q) \tag{127}$$

so that the entropic part of the free-energy is obtained as

$$\frac{1}{2}\partial_n \ln \det \hat{Q}^{\mathrm{RS}}\Big|_{n=0} = \frac{1}{2}\left(\frac{q}{1-q} + \ln(1-q)\right). \tag{128}$$

The interaction part of the free-energy is obtained as

$$
\begin{aligned}
-\mathcal{F}_{\mathrm{int}}[\hat{Q}^{\mathrm{RS}}] &= \exp\left(\frac{1}{2}\sum_{a,b=1}^{n}[(1-q^p)\delta_{ab} + q^p]\frac{\partial^2}{\partial h_a \partial h_b}\right)\prod_{a=1}^{n} e^{-\beta V(\delta+h_a)}\Bigg|_{\{h_a=0\}} \\
&= \exp\left(\frac{1}{2}q^p\sum_{a,b=1}^{n}\frac{\partial^2}{\partial h_a \partial h_b}\right)\prod_{a=1}^{n}\left\{\exp\left(\frac{1}{2}(1-q^p)\frac{\partial^2}{\partial h_a^2}\right)e^{-\beta V(\delta+h_a)}\right\}\Bigg|_{h_a=0} \\
&= \gamma_{q^p}\otimes(\gamma_{1-q^p}\otimes e^{-\beta V(\delta)})^n,
\end{aligned}
\tag{129}
$$

where we used the formula

$$\exp\left(\frac{a}{2}\frac{\partial^2}{\partial h^2}\right)A(h) = \gamma_a \otimes A(h) \tag{130}$$

and the following short hand notations: $\gamma_a(x)$ is a Gaussian with zero mean and variance $a$ [55],

$$\gamma_a(x) = \frac{1}{\sqrt{2\pi a}}e^{-\frac{x^2}{2a}}, \tag{131}$$

by which we write a convolution of a function $A(x)$ with the Gaussian as,

$$\gamma_a \otimes A(x) \equiv \int dy \frac{e^{-\frac{y^2}{2a}}}{\sqrt{2\pi a}}A(x-y) = \int \mathcal{D}z A(x-\sqrt{a}z), \tag{132}$$

where

$$\int \mathcal{D}z \ldots \equiv \int dz \frac{e^{-\frac{z^2}{2}}}{\sqrt{2\pi}}\ldots \tag{133}$$

Collecting the above results we obtain the variational free-energy given by Eq. (122)–Eq. (123) within the RS ansatz as

$$
\begin{aligned}
-\beta f_{\mathrm{RS}}(q) &= \partial_n s_{\mathrm{RS}}(q)|_{n=0} \\
&= \frac{1}{2}\left(\frac{q}{1-q} + \ln(1-q)\right) + \frac{\alpha}{p}\int \mathcal{D}z_0 \ln \int \mathcal{D}z_1 e^{-\beta V(\delta-\sqrt{1-q^p}z_1 - \sqrt{q^p}z_0)}.
\end{aligned}
\tag{134}
$$

### 7.1.2 The saddle point equation

The saddle point equation for the order parameter $q$ is obtained as,

$$
\begin{aligned}
0 &= \frac{\partial(-\beta f_{\mathrm{RS}}(q))}{\partial q} \\
&= \frac{1}{2}\frac{q}{(1-q)^2} - \frac{\alpha}{p}\frac{pq^{p-1}}{2}\int \mathcal{D}z_0 \left(\frac{\int \mathcal{D}z_1 (e^{-\beta V(x)})'}{\int \mathcal{D}z_1 e^{-\beta V(x)}}\right)^2\Bigg|_{x=\delta-\sqrt{1-q^p}z_1 - \sqrt{q^p}z_0} \\
&= \frac{1}{2}\frac{q}{(1-q)^2}\mathcal{G}(q),
\end{aligned}
\tag{135}
$$

where we introduced

$$\mathcal{G}(q) \equiv 1 - \alpha(1-q)^2 q^{p-2} \int \mathcal{D}z_0 \left( \frac{\int \mathcal{D}z_1 (e^{-\beta V(x)})'}{\int \mathcal{D}z_1 e^{-\beta V(x)}} \right)^2 \Bigg|_{x=\delta-\sqrt{1-q^p}z_1-\sqrt{q^p}z_0} . \tag{136}$$

### 7.1.3 Pressure and distribution of gap

Using Eq. (134) we obtain the pressure Eq. (15) as,

$$\Pi = \int \mathcal{D}z_0 \frac{\int \mathcal{D}z_1 (e^{-\beta V(x)})'}{\int \mathcal{D}z_1 e^{-\beta V(x)}} \Bigg|_{x=\delta-\sqrt{1-q^p}z_1-\sqrt{q^p}z_0} \tag{137}$$

and similarly the distribution of the gap given by Eq. (125) as

$$
\begin{aligned}
g(r) &= \int \mathcal{D}z_0 \frac{\int \mathcal{D}z_1 \delta(x-r) e^{-\beta V(x)} \big|_{x=\delta-\sqrt{1-q^p}z_1-\sqrt{q^p}z_0}}{\int \mathcal{D}z_1 e^{-\beta V(x)} \big|_{x=\delta-\sqrt{1-q^p}z_1-\sqrt{q^p}z_0}} \\
&= e^{-\beta V(r)} \int \mathcal{D}z_0 \frac{\gamma_{1-q^p}(\delta - \sqrt{q^p}z_0 - r)}{\int \mathcal{D}z_1 e^{-\beta V(x)} \big|_{x=\delta-\sqrt{1-q^p}z_1-\sqrt{q^p}z_0}} ,
\end{aligned}
\tag{138}
$$

which is properly normalized such that $\int dr\, g(r) = 1$. One can also check easily that the 'virial equation' given by Eq. (18) for the pressure is satisfied, as it should be.

## 7.2 The liquid phase: $q=0$ solution

Apparently $q=0$ representing the liquid state is always a solution of the RS saddle point equation given by Eq. (135) for $p \geq 2$. The stability of the solution must be examined by studying the eigenvalues of the Hessian matrix reported in the appendix B.1.

### 7.2.1 $p=2$ case

From the results in appendix B.1 we find for the $q=0$ solution for $p=2$,

$$
\begin{aligned}
M_1 &= 2 - 2\alpha \left( \frac{\gamma_1 \otimes (e^{-\beta V(\delta)})'}{\gamma_1 \otimes e^{-\beta V(\delta)}} \right)^2 \tag{139} \\
M_2 &= M_3 = 0, \tag{140}
\end{aligned}
$$

from which we find the eigenvalues of the Hessian matrix as (see Eq. (385)),

$$\lambda_R = \lambda_L = \lambda_A = 2 - 2\alpha \left( \frac{\gamma_1 \otimes (e^{-\beta V(\delta)})'}{\gamma_1 \otimes e^{-\beta V(\delta)}} \right)^2 , \tag{141}$$

which vanishes at

$$\alpha_c(\delta) = \left( \frac{\gamma_1 \otimes (e^{-\beta V(\delta)})'}{\gamma_1 \otimes e^{-\beta V(\delta)}} \right)^{-2} = \left( \frac{\int \frac{dz}{\sqrt{2\pi}} e^{-z^2/2} (e^{-\beta V(\delta-z)})'}{\int \frac{dz}{\sqrt{2\pi}} e^{-z^2/2} e^{-\beta V(\delta-z)}} \right)^{-2} . \tag{142}$$

For $\alpha < \alpha_c(\delta)$, the eigenvalues are positive so that the $q=0$ solution is stable but it becomes unstable for $\alpha > \alpha_c(\delta)$.

Interestingly we see that at the critical point $\alpha = \alpha_c(\delta)$, $q = 0$ solves also $\mathcal{G}(q) = 0$ (see Eq. (136)). Since $q \neq 0$ solution must solve $\mathcal{G}(q) = 0$, this suggests a possibility of continuous phase transition at the critical point such that $q \neq 0$ solution emerges continuously. The situation is similar to the Sherrington-Kirkpatrick model which exhibits a continuous spinglass transition at the d'Almeida-Thouless (AT) instability [56] line.

### 7.2.2 $p > 2$ **case**

For $p > 2$, using the results reported in appendix B.1, we find $\lambda_R = \lambda_L = \lambda_A = 2 > 0$ (see Eq. (385)) so that the $q = 0$ solution is always stable. Thus contrary to the $p = 2$ model, we find the liquid phase described by the $q = 0$ RS solution is always (meta)stable. The situation is very similar to the usual $p$-spin spherical spinglass models [9]. Then we are naturally lead to consider the possibility of a discontinuous glass phase represented by 1 step replica symmetry breaking (1RSB) much as the usual $p$-spin SG models (see sec 6) including the random energy model (see sec. 4)). The latter is the standard random first order (RFOT) scenario [8, 53] which is established theoretically for the hardspheres in the $d \to \infty$ limit [14].

## 8 Replica symmetry breaking (RSB) ansatz

Here we continue the previous section and obtain some generic results based on the replica symmetry breaking (RSB) ansatz for the $M$-component continuous vector spin system with generic potential $V(x)$.

### 8.1 Parisi's ansatz

We assume the following structure of the glass order parameter in the glass phase which is the Parisi' ansatz [50]. It reads,

$$Q_{ab}^{k-\mathrm{RSB}} = q_0 + \sum_{i=1}^{k+1}(q_i - q_{i-1})I_{ab}^{m_i} = \sum_{i=0}^{k+1}q_i(I_{ab}^{m_i} - I_{ab}^{m_{i+1}}), \qquad (143)$$

where $I_{ab}^m$ is a kind of generalized ('fat') identity matrix of size $n \times n$ composed of blocks of size $m \times m$. (see Fig. 3) The matrix elements in the diagonal blocks, are all 1 while those in the off-diagonal blocks are all 0. The Parisi's matrix has a hierarchical structure such that

$$1 = m_{k+1} < m_k < \ldots < m_1 < m_1 < m_0 = n, \qquad (144)$$

which becomes

$$0 = m_0 < m_1 < \ldots < m_k < m_{k+1} = 1 \qquad (145)$$

in the $n \to 0$ limit. The expression given by Eq. (143) becomes valid also for the diagonal part by introducing

$$q_{k+1} = 1, \qquad (146)$$

which reflects the normalization of the spins. Let us note that we may sometimes extend the labels $m$'s in Eq. (145) introducing an additional label $m_{k+2}$ just for conveniences. In the last equation of Eq. (143), $I_{ab}^{m_{k+2}} = 0$.

Note that the replica symmetric (RS) ansatz corresponds to $k = 0$. Thus we should be able to recover the results discussed in the previous section 7 by taking $k = 0$ in the following.

The Parisi's ansatz describes the hierarchical organization of the free-energy landscape in glass phases. The value of the matrix elements which belong to the off-diagonal part but most

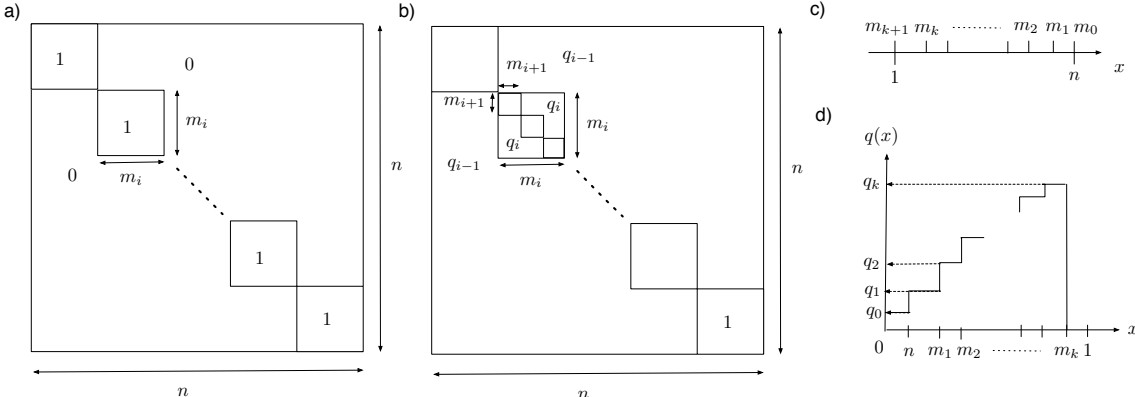

Figure 3: Parametrization of the Parisi's matrix a) the 'fat' identity matrix $I_{ab}^{m_i}$ b) Parisi's order parameter matrix given by Eq. (143) c) the hierarchy of the sizes $m_i$ of the sub-matrices d) the $q(x)$ function with $0 < n < 1$.

close to the diagonal, namely $q_k$, is interpreted as the self-overlap of the glassy states or the Edwards-Anderson order parameter $q_{\text{EA}}$ [21]. Within the RS ansatz discussed in sec 7, $q_{\text{EA}} = q$.

In the $k \to \infty$ limit, the matrix elements can be parametrized by a function $q(x)$ defined in the range $0 \leq x \leq 1$ (See Fig. 3 d)). It encodes the information of the self and mutual overlaps among glassy metastable states. For instance, it is known that [57,58] the probability distribution function $P(q)$ of the overlap $q$ between two independently sampled thermalized spin configurations, say $S^a$ and $S^b$,

$$P(q) = \langle \delta(q - q_{ab}) \rangle, \qquad q_{ab} = \frac{1}{NM} \sum_{i=1}^{N} \sum_{\mu=1}^{M} (S^a)_i^\mu (S^b)_i^\mu, \tag{147}$$

where $\langle \ldots \rangle$ is meant for the thermal average, can be related to the $q(x)$ function as,

$$P(q) = \frac{dx(q)}{dq}, \tag{148}$$

where $x(q)$ is the inverse of $q(x)$. In general the Edwards-Anderson parameter $q_{\text{EA}}$ appears as a plateau height $q(x_1) = q_{\text{EA}}$ of the $q(x)$ function in some range $x_1 < x < 1$ (where $x_1$ corresponds to $m_k$, See Fig. 3 d)). This leads to a delta-peak $\propto \delta(q - q_{\text{EA}})$ in the overlap distribution function $P(q)$. On the other hand, the behavior of the $q(x)$ function at $0 < x < x_1$ encodes non-trivial features of the distribution function $P(q)$ of the mutual overlaps between different glassy metastable states. See [59] for more discussions on the physical consequences of the Parisi's ansatz.

Finally let us note that 'Jamming' simply means disappearance of thermal fluctuations within glassy states due to tightening of constraints. This means the self-overlap saturates to 1, $q_{\text{EA}} = q_k = q(x_1) \to 1$.

## 8.2 Free-energy

Let us evaluate the free-energy given by Eq. (122) -Eq. (123) using the above ansatz. To compute the entropic part of the free-energy one needs to evaluate $\ln \det \hat{Q}^{k-\text{RSB}}$. Given the hierarchical structure of the Parisi's matrix, one can obtain it in a recursive fashion and one

finds [60],

$$\ln \det \hat{Q}^{k-\text{RSB}} = \ln\left(1 + \sum_{j=0}^{k}(m_j - m_{j+1})q_j\right) + \dots$$
$$\dots + n\sum_{i=0}^{k}\left(\frac{1}{m_{i+1}} - \frac{1}{m_i}\right)\ln\left(1 + \sum_{j=i}^{k}(m_j - m_{j+1})q_j - m_i q_i\right) \tag{149}$$

Remembering that $m_0 = n$ we find,

$$\partial_n \ln \det \hat{Q}^{k-\text{RSB}}\Big|_{n=0} = \frac{q_0}{G_0} + \frac{1}{m_1}\ln G_0 + \sum_{i=1}^{k}\left(\frac{1}{m_{i+1}} - \frac{1}{m_i}\right)\ln G_i, \tag{150}$$

with

$$G_i = 1 + \sum_{j=i}^{k}(m_j - m_{j+1})q_j - m_i q_i \qquad i = 0, 1, \dots, k \tag{151}$$

which implies

$$q_i = 1 - G_k + \sum_{j=i+1}^{k}\frac{1}{m_j}(G_j - G_{j-1}) \qquad i = 0, 1, \dots, k \tag{152}$$

The interaction part of free-energy can also be evaluated in a recursive fashion [55]. One finds

$$-\mathcal{F}_{\text{int}}[\hat{Q}^{k-\text{RSB}}] = \prod_{l=0}^{k+1}\exp\left(\frac{\Lambda_l}{2}\sum_{a,b=1}^{n}I_{ab}^{m_l}\frac{\partial^2}{\partial h_a \partial h_b}\right)\prod_{a=1}^{n}e^{-\beta V(\delta+h_a)}\Bigg|_{\{h_a=0\}}$$
$$= \prod_{l=0}^{k}\exp\left(\frac{\Lambda_l}{2}\sum_{a,b=1}^{n}I_{ab}^{m_l}\frac{\partial^2}{\partial h_a \partial h_b}\right)\prod_{a=1}^{n}g(m_{k+1}, h_a)\Bigg|_{\{h_a=0\}}, \tag{153}$$

where we introduced
$$\Lambda_0 \equiv \lambda_0$$
$$\Lambda_i \equiv \lambda_i - \lambda_{i-1} \qquad i = 1, 2, \dots, k+1, \tag{154}$$

with
$$\lambda_i \equiv q_i^p. \tag{155}$$

In the 2nd equation of Eq. (153) we used $I_{ab}^{m_{k+1}=1} = \delta_{ab}$ and introduced

$$g(m_{k+1}, h) \equiv \gamma_{\Lambda_{k+1}} \otimes e^{-\beta V(h)} = \int \mathcal{D}z_{k+1} e^{-\beta V(h-\sqrt{\Lambda_{k+1}}z_{k+1})}, \tag{156}$$

where we used the formula given by Eq. (130).

The expression given by Eq. (153) naturally motivates a family of functions $g$'s which obey a recursion relation,

$$g(m_l, h) = e^{\frac{\Lambda_l}{2}\frac{\partial^2}{\partial h^2}} g^{\frac{m_l}{m_{l+1}}}(m_{l+1}, h) = \gamma_{\Lambda_l} \otimes g^{\frac{m_l}{m_{l+1}}}(m_{l+1}, h) \tag{157}$$

$$= \int \mathcal{D}z_l\, g^{\frac{m_l}{m_{l+1}}}(m_{l+1}, h - \sqrt{\Lambda_l}z_l) \tag{158}$$

for $l = 0, 1, \ldots, k$. Then it is easy to see that

$$
\begin{aligned}
-\mathcal{F}_{\text{int}}[\hat{Q}^{k-\text{RSB}}] &= \gamma_{\Lambda_0} \otimes \left( \gamma_{\Lambda_1} \otimes (\ldots \gamma_{\Lambda_k} \otimes g^{m_k}(m_{k+1}, \delta) \ldots)^{m_1/m_2} \right)^{m_0/m_1} \\
&= \ldots \\
&= \gamma_{\Lambda_0} \otimes \left( \gamma_{\Lambda_1} \otimes g^{m_1/m_2}(m_2, \delta) \right)^{m_0/m_1} \\
&= \gamma_{\Lambda_0} \otimes g^{m_0/m_1}(m_1, \delta) \\
&= g(m_0, \delta).
\end{aligned}
\tag{159}
$$

Equivalently we can introduce a related family of functions $f$'s,

$$
f(m, h) \equiv -\frac{1}{m} \ln g(m, h),
\tag{160}
$$

which follows a recursion relation,

$$
e^{-m_i f(m_i, h)} = \gamma_{\Lambda_i} \otimes e^{-m_i f(m_{i+1}, h)} = \int \mathcal{D}z_i e^{-m_i f(m_{i+1}, h - \sqrt{\Lambda_i} z_i)}
\tag{161}
$$

for $i = 0, 1, \ldots, k+1$ with the boundary condition,

$$
e^{-m_{k+1} f(m_{k+1}, h)} = \int \mathcal{D}z_{k+1} e^{-m_{k+1} \beta V(h - \sqrt{\Lambda_{k+1}} z_{k+1})}.
\tag{162}
$$

where $m_{k+1} = 1$. We may also express the boundary condition as

$$
f(m_{k+2}, h) = \beta V(h)
\tag{163}
$$

by introducing $m_{k+2}$ just as an additional label for convenience. Remembering that $m_0 = n$ we find the interaction part of the free-energy becomes,

$$
\begin{aligned}
-\partial_n \mathcal{F}_{\text{int}}[\hat{Q}^{k-\text{RSB}}]\Big|_{n=0} &= \partial_{m_0} g(m_0, \delta)\Big|_{m_0 = n = 0} = -f(m_0 = 0, \delta) \\
&= \gamma_{\Lambda_0} \otimes (-f(m_1, \delta)) = -\int \mathcal{D}z_0 f(m_1, \delta - \sqrt{\Lambda_0} z_0).
\end{aligned}
\tag{164}
$$

Finally collecting the above results we obtain the free-energy within the $k$-RSB ansatz as

$$
\begin{aligned}
-\beta f_{k-\text{RSB}}[\hat{Q}] &= \frac{1}{2} \frac{q_0}{G_0} + \frac{1}{2} \frac{1}{m_1} \ln G_0 + \frac{1}{2} \sum_{i=1}^{k} \left( \frac{1}{m_{i+1}} - \frac{1}{m_i} \right) \ln G_i + \ldots \\
&\quad \ldots + \frac{\alpha}{p} \int \mathcal{D}z_0 (-f(m_1, \delta - \sqrt{\Lambda_0} z_0)).
\end{aligned}
\tag{165}
$$

### 8.2.1 $k = 0$ case: RS

Let us check if $k = 0$ case recovers the result we obtained previously for the replica symmetric (RS) ansatz.

$$
-\beta f_{0-\text{RSB}}(q_0) = \frac{1}{2} \frac{q_0}{(1 - q_0)} + \frac{1}{2} \ln(1 - q_0) + \frac{\alpha}{p} \int \mathcal{D}z_0 \ln \int \mathcal{D}z_1 e^{-\beta V(\Xi(\delta, q_0))},
\tag{166}
$$

with

$$
\Xi(\delta, q_0) = \delta - \sqrt{1 - q_0^p} z_1 - \sqrt{q_0^p} z_0,
\tag{167}
$$

where we used $G_0 = 1 - (m_1 - m_0) q_0$ and that $m_0 = 0$ and $m_1 = 1$. In the 2nd equation we used Eq. (162). The result agrees with Eq. (134) as it should.

### 8.2.2 $k = 1$ case: 1 RSB

For the $k = 1$ RSB case we find,

$$
\begin{aligned}
-\beta f_{1-\text{RSB}}(q_0, q_1, m_1) = {} & \frac{1}{2} \frac{q_0}{1 - m_1 q_0 + (m_1 - 1)q_1} + \ldots \\
& \cdots + \frac{1}{2} \frac{1}{m_1} \ln(1 - m_1 q_0 + (m_1 - 1)q_1) + \ldots \\
& \cdots + \frac{1}{2} \left(1 - \frac{1}{m_1}\right) \ln(1 - q_1) + \ldots \\
& \cdots + \frac{\alpha}{p} \frac{1}{m_1} \int \mathcal{D}z_0 \ln \int \mathcal{D}z_1 \left[ \int \mathcal{D}z_2 e^{-\beta V(\Xi(\delta, q_0, q_1))} \right]^{m_1},
\end{aligned}
\tag{168}
$$

with

$$
\Xi(\delta, q_0, q_1) = \delta - \sqrt{q_0^p} z_0 - \sqrt{q_1^p - q_0^p} z_1 - \sqrt{1 - q_1^p} z_2.
\tag{169}
$$

where $q_0$ and $q_1$ must be determined through the saddle point equations which we discuss in sec. 8.4.2.

An important quantity is the complexity or the configurations entropy $\Sigma(f)$, which describes the exponentially large number of states $\propto e^{N\Sigma(f)} df$ with free-energy density between $f$ and $f + df$ in the glass phase. Using Monasson's prescription [61], which is equivalent to the approach based on the Franz-Paisi's potential [62, 63], one can construct the complexity function $\Sigma(f)$ treating $m = m_1$ as a parameter;

$$
\Sigma^*(m) = m^2 \partial_m \beta f_{1-\text{RSB}}(q_0, q_1, m)
\tag{170}
$$

$$
\beta f^*(m) = \partial_m (m \beta f_{1-\text{RSB}})(q_0, q_1, m).
\tag{171}
$$

Thus extremization of the free-energy with respect to $m$, $0 = \partial_m \beta f_{1-\text{RSB}}(q_0, q_1, m)$ amounts to force the complexity to vanish $\Sigma^*(m)$ [61].

We readily find the following explicit expressions,

$$
\begin{aligned}
\beta f^*(m_1) = {} & -\frac{1}{2} \frac{q_0}{1 - m_1 q_0 + (m_1 - 1)q_1} - \frac{1}{2} \ln(1 - q_1) - \ldots \\
& \cdots - \frac{1}{2} \left( -\frac{m_1 q_0}{[1 - m_1 q_0 + (m_1 - 1)q_1]^2} + \frac{1}{1 - m_1 q_0 + (m_1 - 1)q_1} \right)(q_1 - q_0) - \ldots \\
& \cdots - \frac{\alpha}{p} \int \mathcal{D}z_0 \frac{\int \mathcal{D}z_1 \left[ \int \mathcal{D}z_2 e^{-\beta V(\Xi(\delta, q_0, q_1))} \right]^{m_1} \ln \left[ \int \mathcal{D}z_2 e^{-\beta V(\Xi(\delta, q_0, q_1))} \right]}{\int \mathcal{D}z_1 \left[ \int \mathcal{D}z_2 e^{-\beta V(\Xi(\delta, q_0, q_1))} \right]^{m_1}}
\end{aligned}
\tag{172}
$$

and

$$
\begin{aligned}
\Sigma^*(m_1) = {} & \frac{1}{2} \ln[1 - m_1 q_0 + (m_1 - 1)q_1] - \frac{1}{2} \ln(1 - q_1) + \ldots \\
& \cdots + \frac{\alpha}{p} \int \mathcal{D}z_0 \ln \left[ \int \mathcal{D}z_1 \left[ \int \mathcal{D}z_2 e^{-\beta V(\Xi(\delta, q_0, q_1))} \right]^{m_1} \right] + \ldots \\
& \cdots + m_1 \left\{ -\frac{1}{2} \left( -\frac{m_1 q_0}{[1 - m_1 q_0 + (m_1 - 1)q_1]^2} + \frac{1}{1 - m_1 q_0 + (m_1 - 1)q_1} \right)(q_1 - q_0) - \ldots \right. \\
& \left. \cdots - \frac{\alpha}{p} \int \mathcal{D}z_0 \frac{\int \mathcal{D}z_1 \left[ \int \mathcal{D}z_2 e^{-\beta V(\Xi(\delta, q_0, q_1))} \right]^{m_1} \ln \left[ \int \mathcal{D}z_2 e^{-\beta V(\Xi(\delta, q_0, q_1))} \right]}{\int \mathcal{D}z_1 \left[ \int \mathcal{D}z_2 e^{-\beta V(\Xi(\delta, q_0, q_1))} \right]^{m_1}} \right\}.
\end{aligned}
\tag{173}
$$

### 8.2.3 $k = \infty$ case: continuous RSB

In the limit $k \to \infty$, the overlap matrix $\hat{Q}$ is parametrized by a function $q(x)$ with $n < x < 1$. Then Eq. (149) becomes,

$$\ln \det \hat{Q}^{\infty-\text{RSB}} = \ln \left( 1 - \int_n^1 dx\, q(x) \right) - n \int_n^1 \frac{dx}{x^2} \ln \left( 1 - \int_x^1 dy\, q(y) - x q(x) \right). \tag{174}$$

From the above expression we find

$$\partial_n \ln \det \hat{Q}^{\infty-\text{RSB}} \bigg|_{n=0} = \frac{q(0)}{G(0)} + \ln G(1) + \int_0^1 \frac{dx}{G(x)}, \tag{175}$$

with

$$G(x) \equiv 1 - \int_x^1 dy\, q(y) - x q(x). \tag{176}$$

Then the free-energy given by Eq. (165) can be written as,

$$-\beta f_{\infty-\text{RSB}}[\hat{Q}] = \frac{1}{2} \left[ \frac{q(0)}{G(0)} + \frac{1}{2} \ln G(1) + \dots \right.$$
$$\left. \dots + \int_0^1 \frac{dx}{G(x)} \right] + \frac{\alpha}{p} \int \mathcal{D}z_0 (-f(m(0), \delta - \sqrt{q^p(0)} z_0)) \tag{177}$$

where the function $f(x, h)$ obeys

$$\dot{f}(x, h) = -\frac{1}{2} \dot{\lambda}(x) \left[ f''(x, h) - x \left( f'(x, h) \right)^2 \right], \tag{178}$$

with

$$\lambda(x) \equiv q^p(x). \tag{179}$$

Here and in the following we denote a partial derivative with respect to the 1st argument by a dot, e. g. $\partial_x f(x, h) = \dot{f}(x, h)$ and that with respect to the 2nd argument by a dash e. g. $\partial_h f(x, h) = f'(x, h)$. The partial differential equation given by Eq. (178) is the continuous limit of recursion formula Eq. (161). The boundary condition given by Eq. (162) becomes,

$$f(1, h) = -\ln \int \mathcal{D}z\, e^{-\beta V(h - \sqrt{1 - q^p(1)} z)}. \tag{180}$$

## 8.3 Variation of the interaction part of the free-energy

Later we will often meet the needs to consider variation of the free-energy. This happens when we solve the saddle point equation for the order parameters $q_l$'s (sec. 8.4), analyze the stability of the saddle point solutions (sec. 8.5), compute the pressure $\Pi$ given by Eq. (124) and the distribution of the gap $g(r)$ given by Eq. (125). We note that a variation scheme for continuous RSB has been formulated originally in [64].

### 8.3.1 Some useful functions

Here we consider a strategy make variation of the interaction part of the free-energy given by Eq. (153). Within the Parisi's ansatz it becomes Eq. (164) which reads as,

$$-\partial_n \mathcal{F}_{\text{int}}[\hat{Q}^{k-\text{RSB}}] \bigg|_{n=0} = -f(m_0 = n = 0, \delta) = -\int \mathcal{D}z_0 f(m_1, \delta - \sqrt{\Lambda_0} z_0). \tag{181}$$

As we discussed in sec. 8.2, the interaction part of the free-energy is is constructed in a recursive way and the functions $f(m_i, h)$ follows a recursion formula Eq. (161). This fact naturally motivates us to introduce for $0 \leq i \leq j \leq k+1$,

$$P_{i,j}(y,h) \equiv \frac{\delta f(m_i, y)}{\delta f(m_j, h)}. \tag{182}$$

Using the chain rule we can write,

$$P_{i,j}(y,z) = \int dx \, P_{i,j-1}(y,x) P_{j-1,j}(x,z), \tag{183}$$

where

$$P_{j-1,j}(x,z) = \frac{\delta f(m_{j-1}, x)}{\delta f(m_j, z)} = e^{m_{j-1}(f(m_{j-1},x) - f(m_j,z))} \frac{e^{-\frac{(x-z)^2}{2\Lambda_{j-1}}}}{\sqrt{2\pi\Lambda_{j-1}}} \tag{184}$$

as one can easily find from the recursion relation given by Eq. (161). Then we find a recursion relation

$$P_{ij}(y,z) = e^{-m_{j-1}f(m_j,z)} \gamma_{\Lambda_{j-1}} \otimes_z \frac{P_{i,j-1}(y,z)}{e^{-m_{j-1}f(m_{j-1},z)}}, \tag{185}$$

with the 'boundary condition'

$$P_{ii}(y,h) = \delta(y-h). \tag{186}$$

Here $\otimes_h$ stands for a convolution with respect to the variable $h$. A useful property to note is that the recursion relation given by Eq. (185) preserves the 'normalization',

$$\int dh \, P_{i,j}(y,h) = 1, \tag{187}$$

which can be easily proved using Eq. (161).

For a convenience, let us introduce another quantity which is related to $P_{ij}(y,h)$,

$$P(m_j, h) \equiv \frac{\delta f(m_0, \delta)}{\delta f(m_{j+1}, h)} = P_{0,j+1}(\delta, h) = \int dx \, P_{0,1}(\delta, x) P_{1,j+1}(x, h)$$
$$= \int \mathcal{D}z_0 \, P_{1,j+1}(\delta - \sqrt{\Lambda_0} z_0, h), \tag{188}$$

where we used the chain rule, Eq. (184) and set $m_0 = n \to 0$. Clearly it follows the same recursion formula as Eq. (185),

$$P(m_j, h) = e^{-m_j f(m_{j+1}, h)} \gamma_{\Lambda_j} \otimes_h \frac{P(m_{j-1}, h)}{e^{-m_j f(m_j, h)}}, \qquad j = 1, 2, \ldots, k+1, \tag{189}$$

with the 'boundary condition'

$$P(m_0, h) = \frac{1}{\sqrt{2\pi\Lambda_0}} e^{-\frac{(\delta-h)^2}{2\Lambda_0}}, \tag{190}$$

which follows from Eq. (188) and Eq. (186). The functions $P(m_i, h)$ is also normalized such that

$$\int dh \, P(m_i, h) = 1, \tag{191}$$

reflecting Eq. (187).

In the $k \to \infty$ limit, the function $P(x,h)$ can be obtained by solving a differential equation,

$$\dot{P}(x,h) = \frac{1}{2}\dot{\lambda}(x)\left[P''(x,h) - 2x(P(x,h)\pi(x,h))'\right], \tag{192}$$

which is the continuous limit of Eq. (189). The boundary condition given by Eq. (190) becomes,

$$P(0,h) = \frac{1}{\sqrt{2\pi q^P(0)}} e^{-\frac{(\delta-h)^2}{2q^P(0)}}. \tag{193}$$

### 8.3.2 Distribution of the gap

Now the distribution function of the gap $g(r)$ given by Eq. (125) for the generic k-RSB ansatz is obtained using Eq. (165), Eq. (163) which reads $\beta V(r) = f(m_{k+2}, r)$, Eq. (182), Eq. (188) and Eq. (189) as

$$\begin{aligned}
g(r) &= -\frac{p}{\alpha}\frac{\delta\beta f_{k-RSB}[\hat{Q}^*]}{\delta(-\beta V(r))} = \int \mathcal{D}z_0 \frac{\delta f(m_1, \delta - \sqrt{\Lambda_0}z_0)}{\delta f(m_{k+2}, r)} \\
&= \int \mathcal{D}z_0 P_{1,k+2}(\delta - \sqrt{\Lambda_0}z_0, r) = P(m_{k+1}, r) = e^{-\beta V(r)}\gamma_{\Lambda_{k+1}} \otimes_r \frac{P(m_k, r)}{e^{-f(m_{k+1}=1,r)}}.
\end{aligned} \tag{194}$$

We used $m_{k+1} = 1$ in the last equation. It can be seen that $g(r)$ is properly normalized such that $\int dr\, g(r) = 1$ reflecting Eq. (191). The previous result in the RS case ($k=0$) Eq. (138) can be recovered using Eq. (190), Eq. (161), Eq. (162) in Eq. (194) as expected.

### 8.3.3 Pressure

The pressure given by Eq. (124) for the generic $k$-RSB ansatz is obtained using and Eq. (165) as

$$\Pi = -\frac{p}{\alpha}\frac{\partial\beta f_{k-RSB}[\hat{Q}^*]}{\partial\delta} = -\int \mathcal{D}z_0 \frac{\partial f(m_1, \delta - \sqrt{\Lambda_0}z_0)}{\partial\delta} = \int \mathcal{D}z_0 \pi(m_1, \delta - \sqrt{\Lambda_0}z_0), \tag{195}$$

where we introduced

$$\pi(m,h) \equiv -f'(m,h). \tag{196}$$

Here and in the following the prime stands for taking a partial derivative with respect to the 2nd argument $h$.

The function $\pi(m,h)$ introduced above also follows a recursion formula which can be obtained using Eq. (161) and Eq. (162) as

$$\pi(m_i, h) = e^{m_i f(m_i, h)}\gamma_{\Lambda_i} \otimes \pi(m_{i+1}, h)e^{-m_i f(m_{i+1}, h)} \tag{197}$$

for $i = 1, 2, \ldots, k$ with the 'boundary condition'

$$\pi(m_{k+1}, h) = \frac{\int \mathcal{D}z_{k+1}(e^{-\beta V(h - \sqrt{\Lambda_{k+1}}z_{k+1})})'}{\int \mathcal{D}z_{k+1}e^{-\beta V(h - \sqrt{\Lambda_{k+1}}z_{k+1})}}. \tag{198}$$

The previous result in the RS case ($k=0$) given by Eq. (137) can be recovered using Eq. (198) in Eq. (195).

In the $k \to \infty$ limit, the function $\pi(x,h) \equiv -f'(x,h)$ obeys a differential equation

$$\dot{\pi}(x,h) = -\frac{\dot{\lambda}(x)}{2}\left(\pi''(x,h) + 2x\pi(x,h)\pi'(x,h)\right), \tag{199}$$

which is the continuous limit of Eq. (197) and can be obtained from Eq. (178). The boundary condition for the latter is given by Eq. (198) which reads

$$\pi(1, h) = \frac{\int \mathcal{D}z(e^{-\beta V(h - \sqrt{1 - q^p(1)}z)})'}{\int \mathcal{D}z e^{-\beta V(h - \sqrt{1 - q^p(1)}z)}}. \tag{200}$$

It is instructive to verify that the pressure given by Eq. (195) can be recovered through the virial equation for the pressure given by Eq. (18). In fact the pressure can be expressed as

$$\Pi = \int dh P(m_{i-1}, h) \pi(m_i, h), \tag{201}$$

with *any* $i = 1, 2, \ldots, k+2$ (so that $\pi = \int dh P(x, h) \pi(x, h)$ for any $0 < x < 1$ in the $k \to \infty$ limit). Using the recursion formulas given by Eq. (189) and Eq. (197) one can check that $\int dh P(m_{i-1}, h) \pi(m_i, h) = \int dh' P(m_i, h') \pi(m_{i+1}, h')$ so that the r.h.s. of the above equation does not depend on the level $i$ of the hierarchy. The case of the virial equation for the pressure $\Pi = \int dr g(r)(-\beta V'(r))$ given by Eq. (18) corresponds to the case $i = k+2$ which can be seen by noting $\pi(m_{k+2}, h) = -f'(m_{k+2}, h) = -\beta V'(h)$ (see Eq. (162)) and Eq. (194). On the other hand, the case $i = 1$ corresponds to the expression given by Eq. (195) which can be seen using Eq. (190).

## 8.4 Saddle point equations for the order parameters

Here we derive variational equations to determine $q_i$ for $i = 0, 1, 2, \ldots, k$. Since $q_i$'s are related linearly to $G_i$'s through Eq. (152), the saddle point equations can be written as

$$\begin{aligned}
0 &= \frac{\partial(-\beta f_{k-RSB}[\hat{Q}])}{\partial G_i} \\
&= \frac{1}{2}\left[-\frac{q_0}{G_0^2}\delta_{i,0} + \left(\frac{1}{m_{i+1}} - \frac{1}{m_i}\right)\left(\frac{1}{G_i} - \frac{1}{G_0}\right)(1 - \delta_{i,0})\right] + \ldots \\
&\quad \cdots + \left[-\left(\frac{1}{m_{i+1}} - \frac{1}{m_i}\right)\sum_{j=0}^{i-1} pq_j^{p-1}\frac{\partial}{\partial \lambda_j} - \frac{1}{m_{i+1}}pq_i^{p-1}\frac{\partial}{\partial \lambda_i}\right] \times \ldots \\
&\quad \cdots \times \frac{\alpha}{p}\int \mathcal{D}z_0(-f(m_1, \delta - \sqrt{\Lambda_0}z_0)).
\end{aligned} \tag{202}$$

In the last equation we used Eq. (155) and Eq. (154). As we show in appendix C we find,

$$-\frac{\partial}{\partial \lambda_j}f(m_i, y) = \frac{1}{2}(m_j - m_{j+1})\int dh P_{i,j+1}(y, h)\pi^2(m_{j+1}, h), \tag{203}$$

where $P_{i,j}(y, h)$ is defined in Eq. (182) and $\pi(m, h)$ is defined in Eq. (196).

Collecting the above results we obtain the variational equations as

$$\frac{q_0}{G_0^2} = \kappa_0$$

$$\frac{1}{G_i} - \frac{1}{G_0} = \sum_{j=0}^{i-1}(m_j - m_{j+1})\kappa_j + m_i\kappa_i \qquad i = 1, 2, \ldots, k, \tag{204}$$

where we introduced

$$\kappa_j \equiv \alpha q_j^{p-1}\int dh P(m_j, h)\pi^2(m_{j+1}, h), \qquad j = 0, 1, \ldots, k. \tag{205}$$

Note that $P(m_j, h)$ and $\pi(m_i, h)$ used here can be obtained by solving the recursion formulas given by Eq. (189) and Eq. (197) respectively together with their boundary conditions.

Finally we note that for $p > 1$, $q_0 = 0$ always solves the 1st equation of Eq. (204).

### 8.4.1 $k = 0$ case: RS

Let us check if $k = 0$ case recovers the result we obtained previously for the replica symmetric (RS) ansatz. In this case we just need the 1st equation of Eq. (204) which becomes,

$$
\begin{aligned}
\frac{q_0}{(1-q_0)^2} &= \alpha q_0^{p-1} \int dh P(m_0, h)(\pi(m_1, h))^2 \\
&= \alpha q_0^{p-1} \int \mathcal{D}z_0 \left( \frac{\int \mathcal{D}z_1 (e^{-\beta V(x)})'}{\int \mathcal{D}z_1 e^{-\beta V(x)}} \right)^2 \Bigg|_{x = \delta - \sqrt{1-q_0^p} z_1 - \sqrt{q_0^p} z_0},
\end{aligned}
\tag{206}
$$

where we used $G_0 = 1 - (m_1 - m_0)q_0$ and that $m_0 = 0$ and $m_1 = 1$. In the 2nd equation we used Eq. (198) and Eq. (190). The result agrees with Eq. (135) as it should.

### 8.4.2 $k = 1$ case: 1RSB

For the $k = 1$ case (1RSB) Eq. (204) becomes,

$$
\begin{aligned}
\frac{q_0}{G_0^2} &= \kappa_0, \\
\frac{1}{G_1} - \frac{1}{G_0} &= m_1(\kappa_1 - \kappa_0),
\end{aligned}
\tag{207}
$$

with

$$
\begin{aligned}
\kappa_0 &= \alpha q_0^{p-1} \int dh P(m_0, h) \pi^2(m_1, h), \\
\kappa_1 &= \alpha q_1^{p-1} \int dh P(m_1, h) \pi^2(m_2, h).
\end{aligned}
\tag{208}
$$

After solving the above equations for $G_0$ and $G_1$, the order parameters $q_0$ and $q_1$ can be obtained as (See Eq. (152))

$$
\begin{aligned}
q_0 &= 1 - G_1 + \frac{1}{m_1}(G_1 - G_0), \\
q_1 &= 1 - G_1.
\end{aligned}
\tag{209}
$$

To evaluate $\kappa_0$ and $\kappa_1$ in Eq. (208) we need more information. Suppose that we are given some initial guess for the values of $q_0$ and $q_1$. Then we can recursively obtain functions $f(m_2, h)$ and $f(m_1, h)$ (see Eq. (162)) and Eq. (161)) as

$$
\begin{aligned}
e^{-m_2 f(m_2, h)} &= \int \mathcal{D}z_2 e^{-\beta V(h - \sqrt{1-q_1^p} z_2)}, \\
e^{-m_1 f(m_1, h)} &= \int \mathcal{D}z_1 e^{-m_1 f(m_2, h - \sqrt{q_1^p - q_0^p} z_1)},
\end{aligned}
\tag{210}
$$

where $m_2 = 1$. Similarly we can recursively obtain functions $\pi(m_1, h)$ and $\pi(m_2, h)$ (See Eq. (197) and Eq. (198)) as

$$
\begin{aligned}
\pi(m_2, h) &= \frac{\int \mathcal{D}z_2 (e^{-\beta V(h - \sqrt{1-q_1^p} z_2)})'}{\int \mathcal{D}z_2 e^{-\beta V(h - \sqrt{1-q_1^p} z_2)}}, \\
\pi(m_1, h) &= e^{m_1 f(m_1, h)} \int \mathcal{D}z_1 \, \pi(m_2, h') e^{-m_1 f(m_2, h')} \Bigg|_{h' = h - \sqrt{q_1^p - q_0^p} z_1}.
\end{aligned}
\tag{211}
$$

Next we can recursively obtain functions $P(m_0, h)$ and $P(m_1, h)$ (see Eq. (189) and Eq. (190)) as

$$P(m_0, h) = \frac{1}{\sqrt{2\pi q_0^p}} e^{-\frac{(\delta - h)^2}{2q_0^p}},$$

$$P(m_1, h) = e^{-m_1 f(m_2, h)} \int \mathcal{D}z_1 \, P(m_0, h') e^{m_1 f(m_1, h')} \Big|_{h' = h - \sqrt{q_1^p - q_0^p} z_1}.$$

(212)

With these we are now readily to evaluate $\kappa_0$ and $\kappa_1$ using Eq. (208).

To sum up, we can evaluate the 1RSB solution numerically as follows: (0) make some initial guess for the values of $q_0$ and $q_1$ (1) obtain functions $f(m_2, h) \to f(m_1, h)$ using Eq. (210) (2) obtain functions $\pi(m_2, h) \to \pi(m_1, h)$ using Eq. (211) (3) obtain functions $P(m_0, h) \to P(m_1, h)$ using Eq. (212) (4) Compute $\kappa_0$ and $\kappa_1$ using Eq. (208) (4) solve for $G_0$ and $G_1$ using Eq. (207) and Eq. (208) (5) compute $q_0$ and $q_1$ using Eq. (209) (6) return to (1). The procedure has to be repeated until the solution converges.

We note that the parameter $m_1$ remains. In order to study the equilibrium state $m_1$ is fixed by the condition of vanishing complexity $\Sigma(m_1) = 0$. (See sec. 8.2.2)

### 8.4.3 $k > 1$ case

The saddle point equations for a generic finite $k$-RSB ansatz with some fixed values of $0 < m_1 < m_2 < \ldots < m_k < 1$ can be solved numerically generalizing the procedure explained above.

0. Make some guess for the initial values of $q_i$ $(i = 0, 1, 2, \ldots, k)$. Then compute $G_i$ for $i = 0, 1, \ldots, k$ using Eq. (151).

1. Compute function $f(m_i)$ recursively for $i = k + 1, k, \ldots, 0$ using Eq. (161) with the boundary condition given by Eq. (163). Compute also functions $\pi(m_i, h)$ recursively for $i = k, k - 1, \ldots, 2, 1$ using Eq. (197) with the boundary condition given by Eq. (198).

2. Compute functions $P(m_i, h)$ recursively for $i = 1, \ldots, k + 1$ using Eq. (189) with the boundary condition given by Eq. (190).

3. Compute $\kappa_i$ for $i = 0, 1, \ldots, k$ using Eq. (205), $G_i$ for $i = 0, 1, \ldots, k + 1$ using using Eq. (204), $q_i$ for $i = 0, 1, 2, \ldots, k$ using Eq. (152) and finally $\Lambda_i$ for $i = 0, 1, \ldots, k + 1$ using Eq. (154) and Eq. (155).

4. Return to 1.

The above procedure 1-4 must be repeated until the solution converges.

### 8.4.4 $k = \infty$ case: continuous RSB

In the limit $k \to \infty$, the variational equations given by Eq. (204) become,

$$\frac{q(0)}{G^2(0)} = \kappa(0),$$

(213)

$$\frac{1}{G(x)} - \frac{1}{G(0)} = -\int_0^x dy \kappa(y) + x\kappa(x),$$

(214)

with

$$\kappa(x) \equiv \alpha q^{p-1}(x) \int dh P(x, h) \pi^2(x, h).$$

(215)

From the above equations we can derive some exact identities which become useful later. Taking a derivative with respect to $x$ on both sides of Eq. (214) and using Eq. (215), Eq. (176), Eq. (192), Eq. (199), we find after some integrations by parts,

$$
\begin{aligned}
1 = \alpha(p-1)q^{p-2}(x)G^2(x)\int dh P(x,h)\pi^2(x,h) + \dots \\
\dots + \alpha p q^{2(p-1)}(x)G^2(x)\int dh P(x,h)(\pi'(x,h))^2.
\end{aligned}
\tag{216}
$$

Then taking another derivative on both sides of the above equation we find after some integrations by parts,

$$
\begin{aligned}
0 = (p-1)q^{p-3}(x)[(p-2)G^2(x) - 2q(x)xG(x)]\int dh P(x,h)\pi^2(x,h) + \dots \\
\dots + 3p(p-1)q^{2p-3}(x)G^2(x)\int dh P(x,h)(\pi'(x,h))^2 + \dots \\
\dots + pq^{2(p-1)}(x)(-2xG(x))\int dh P(x,h)(\pi'(x,h))^2 + \dots \\
\dots + p^2 q^{3(p-1)}(x)G^2(x)\int dh P(x,h)\big[(\pi''(x,h))^2 - 2x(\pi'(x,h))^3\big].
\end{aligned}
\tag{217}
$$

## 8.5 Stability of the kRSB solution

Stability of the $k(\geq 1)$-RSB ansatz must be examined by studying the eigenvalues of the Hessian matrix. As we note in appendix B.2, there is a residual replica symmetry within each of the inner-core part of the replica groups. Here we do not study the complete spectrum of the eigen-modes of the Hessian matrix but focus on the so called replicon eigenvalue $\lambda_R$ which is responsible for the replica symmetry breaking of the residual replica symmetry.

### 8.5.1 $k = 1$ case: 1RSB

For the $k = 1$ case we find from Eq. (399),

$$
\begin{aligned}
\lambda_R = \frac{2}{(1-q_1)^2} - \dots \\
\dots - 2\frac{\alpha}{p}\int dh P(m_1,h)\big[p(p-1)q^{p-2}(\pi(m_2,h))^2 + (pq^{p-1})^2(\pi'(m_2,h))^2\big],
\end{aligned}
\tag{218}
$$

where $m_2 = 1$. Here we used $\pi(x,h) \equiv -f'(x,h)$ defined in Eq. (196). The functions $\pi(m_2,h)$ and $P(m_1,h)$ are given by Eq. (211) and Eq. (212) respectively.

The vanishing on $\lambda_R$ signals the Gardner's transition [54]: instability to further breaking of the replica symmetry.

### 8.5.2 $k = \infty$ case: continuous RSB

From Eq. (399) we find for $k = \infty$, by which $m_k \to 1$,

$$
\lambda_R = \frac{2}{G(1)^2} - 2\frac{\alpha}{p}\int dh P(1,h)\big[p(p-1)q^{p-2}(\pi(1,h))^2 + (pq^{p-1})^2(\pi'(1,h))^2\big],
\tag{219}
$$

where we used $\pi(x,h) \equiv -f'(x,h)$ defined in Eq. (196) and $G(1) = 1 - q(1)$ which follows from Eq. (176). Now using the exact identity given by Eq. (216) which holds for the continuous RSB system, we find it vanishes exactly: $\lambda_R = 0$. Thus the continuous RSB solution is marginally stable.

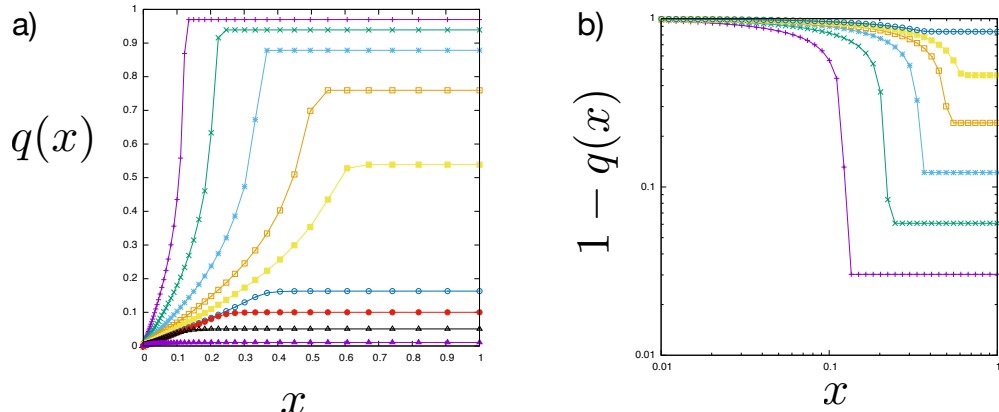

Figure 4: The $q(x)$ function of the quadratic potential with $p = 2$, $\delta = 1.0$, $\alpha = 2$ for which $T_c/\epsilon = \sqrt{2} - 1 = 0.414...$  a) linear plot: $T/\epsilon = 0.00625, 0.0125, 0.025, 0.05, 0.25, 0.3, 0.35, 0.4$ from top to bottom. The points represent the solution of the continuous RSB equation (obtained by solving the recursion formulas with $k = 200$ steps). b) double logarithmic of the function $1 - q(x)$ vs $x$.

## 9  Quadratic potential

Now we are ready to study specific problems with non-linear potentials $V(x)$ and continuous spins. First let us briefly examine the simplest one,

$$V(x) = \frac{\epsilon}{2}x^2, \qquad \epsilon > 0. \tag{220}$$

As we show below it is already a non-trivial problem. To see this it is useful to expand the interaction part of the free-energy given by Eq. (123) in power series of the order parameter,

$$-\partial_n \mathcal{F}_{\text{int}}\big|_{n=0} = \partial_n \left[ n\frac{(\beta\epsilon)^2}{2}(1+\delta^2) + \delta^2\frac{(\beta\epsilon)^2}{2}\sum_{ab}Q_{ab}^p + \frac{(\beta\epsilon)^2}{2}\sum_{ab}Q_{ab}^{2p} + \dots \right.$$

$$\left. \dots + \frac{2}{3}(\beta\epsilon)^3 \sum_{abc}Q_{ab}^p Q_{bc}^p Q_{ca}^p + \dots \right]\Bigg|_{n=0}. \tag{221}$$

In contrast to the case of the linear potential discussed in sec 6, higher order terms of $Q_{ab}^p$ appears. Thus even with $p = 2$, for which system remains RS for the linear potential case (spherical SK model [51]), one can expect that the quadratic potential allows RSB since the above expression is somewhat similar to the interaction part of the free-energy of the $2 + p$ spherical model [65], which exhibits various types of RSB.

   In the present paper we do not explore the whole phase diagram but let us show the existence of continuous RSB for the $p = 2$ case. From Eq. (142) we find the critical point,

$$\alpha_c(\delta) = \left(1 + \frac{1}{\beta\epsilon}\right)^2 |\delta|^{-2}, \tag{222}$$

which implies the glass transition temperature,

$$T_c(\alpha, \delta) = \sqrt{\alpha}|\delta| - 1. \tag{223}$$

Above the critical temperature, i. e. $T > T_c$ the $q = 0$ RS solution (liquid phase) is valid (sec. 7.2). One naturally expects continuous glass transition takes place at $T_c$.

We solved numerically the continuous RSB equation approximated by $k$-step RSB (With $k = 200$) recursion formulas with appropriate boundary conditions as explained in sec. 8.4.3. In Fig. 4 we show the continuous RSB solution obtained numerically for the case of $p = 2$, $\alpha = 2$ and $\delta = 1.0$ at various temperatures below $T_c/\epsilon = \sqrt{2} - 1 = 0.414...$ As expected the glass transition takes place continuously. Moreover it accompanies continuous RSB as evident in the figure. The $q(x)$ function has a plateau $q(x) = q_1$ for some range $x_1 < x < 1$. The plateau height $q_1$ is interpreted as the self-overlap of the glassy states or the Edwards-Anderson order parameter $q_{EA}$ [21] while the continuous part at $x < x_1$ describes the hierarchical organization of the glassy states [59].

Finally let us consider stability of the glass phase against crystallization. From the analysis in sec. 5.3 we know that for the case $p = 2$ the flatness of the potential $V'(\delta) = 0$ is needed to ensure the locally stability against crystallization in the disorder-free model. In the case of the quadratic potential Eq. (220) the condition is met only for $\delta = 0$. However the above results imply $\alpha_c(\delta = 0) = \infty$. Thus for the $p = 2$ case with the quadratic potential, we cannot avoid using the disordered model given by Eq. (6) in order to allow the desired glass transition. The free-energy functional of the disordered model is given by Eq. (109) with Eq. (110). The solution we obtained just above amount to assume $m = 0$. Such a solution is certainly locally stable for the fully disordered case $\lambda/\sqrt{M} = 0$ and presumably also for small enough $\lambda$ (see Eq. (117)).

# 10 Hardcore potential

We will now focus on the continuous spin system subjected to a more strongly non-linear potential, namely soft/hardcore potential,

$$V(x) = \epsilon x^2 \theta(-x), \qquad \epsilon > 0, \tag{224}$$

which becomes a hardcore potential in the limit $\epsilon \to \infty$. Much as in hardspheres [13,16] we can expect jamming $q_{EA} \to 1$, i. e. vanishing of the thermal fluctuation due to tightening of the constraints. This would happen in two ways: by decreasing $\delta$ or increasing $\alpha$ (connectivity). In the context of the coloring problems decreasing $\delta$ is analogous to decreasing the number of colors allowed to use (see Fig. 1 a)).

Note that in the special case $p = 1$ and with fully disordered choice $\lambda/\sqrt{M} = 0$ in Eq. (6), the model becomes identical to the perceptron problem [43]. Recent works [35, 37] have established that the universality class of the jamming in the perceptron model with $\delta < 0$ is the same as that of hardspheres [16]. We wish to clarify if the same universality holds for $p \geq 2$ or not.

The soft/hardcore potential defined above is flat such that $V'(x) = 0$ for $x > 0$. Thus for $\delta > 0$, the supercooled paramagnetic phase and glassy phases are locally stable against crystallization even in the $p = 2$ system (See Eq. (46), Eq. (48) and Eq. (117)). In contrast, if $\delta < 0$ and $p = 2$, the quenched disorder is needed to allow the glassy phases. For $p > 2$, the the supercooled paramagnetic states and glassy states are always locally stable against crystallization. In the following we study both $\delta > 0$ and $\delta < 0$ assuming $m = 0$ but we should keep these points in our mind.

Below we closely follow the analysis done on hardspheres in $d \to \infty$ limit [16] and find indeed that many aspects are quite similar to those found there, especially at jamming.

## 10.1 Replica symmetric solution

Let us first study the RS solution discussed in sec. 7 in the case of the hardcore model.

### 10.1.1 Free-energy

For the hardcore potential we find the RS free-energy given by Eq. (134) as

$$-\beta f_{\rm RS}(q) = \frac{1}{2}\left(\frac{q}{1-q} + \ln(1-q)\right) + \frac{\alpha}{p}\int \mathcal{D}z_0 \ln\Theta\left(\frac{\delta - \sqrt{q^p}z}{\sqrt{2(1-q^p)}}\right), \qquad (225)$$

where we introduced a function $\Theta(x)$

$$\Theta(x) \equiv \int_{-\infty}^{x}\frac{dz}{\sqrt{\pi}}e^{-z^2} = \gamma_{1/2}\otimes\theta(x) = \frac{1}{2}(1+\mathrm{erf}(x)), \qquad (226)$$

with $\mathrm{erf}(x)$ being the error function,

$$\mathrm{erf}(x) = \frac{2}{\sqrt{\pi}}\int_0^x dy\, e^{-y^2} = -\mathrm{erf}(-x), \qquad (227)$$

which behaves for $x\to\infty$ as

$$\mathrm{erf}(x) = 1 - \frac{1}{\sqrt{\pi}}\frac{e^{-x^2}}{x}\left(1 - \frac{1}{2x^2} + \frac{3}{(2x^2)^2} + \dots\right). \qquad (228)$$

This implies,

$$\Theta(x) \simeq \begin{cases} \frac{1}{2}\frac{e^{-x^2}}{(-x)\sqrt{\pi}}\left[1 - \frac{1}{2x^2} + \frac{3}{(2x^2)^2} + \dots\right] & x\to-\infty, \\ 1 & x\to\infty. \end{cases} \qquad (229)$$

The function $\mathcal{G}(q)$ defined in Eq. (136) becomes

$$\mathcal{G}(q) = 1 - \alpha(1-q)^2 q^{p-2}\frac{1}{2(1-q^p)}\int \mathcal{D}z_0\, r^2(x)\Big|_{x=\frac{\delta - \sqrt{q^p}z_0}{\sqrt{2(1-q^p)}}}, \qquad (230)$$

where we introduced

$$r(x) \equiv \frac{\Theta'(x)}{\Theta(x)} = \frac{e^{-x^2}}{\sqrt{\pi}}/\Theta(x), \qquad (231)$$

which behaves asymptotically as

$$r(x) \simeq \begin{cases} -2x\left(1 - \frac{1}{2x^2} + \frac{3}{(2x^2)^2} + \dots\right)^{-1} & x\to-\infty, \\ 0 & x\to\infty. \end{cases} \qquad (232)$$

### 10.1.2 $q=0$ RS solution and its stability

Within the liquid state $q=0$, the $p$-dependence disappears. The pressure given by Eq. (137) is obtained as

$$\Pi = \frac{1}{\sqrt{2}}r(\delta/\sqrt{2}) = \frac{1}{\sqrt{2}}\frac{\Theta'(\delta/\sqrt{2})}{\Theta(\delta/\sqrt{2})}. \qquad (233)$$

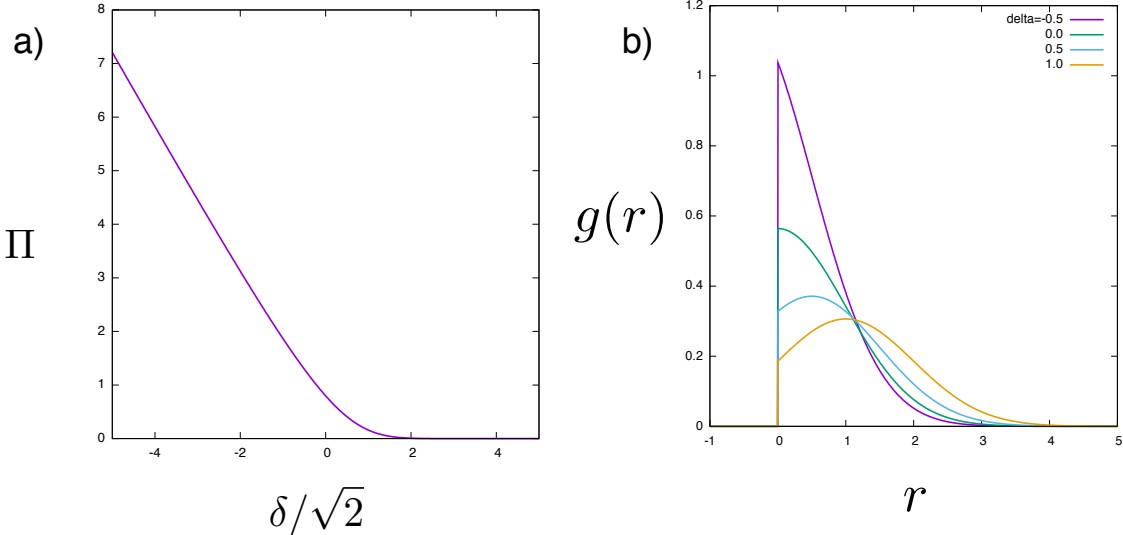

Figure 5: Liquid phase (RS solution with $q = 0$) of the hardcore model: a) behavior of the pressure $\Pi$ and b) the distribution of the gap $g(r)$. Note that these are independent of $p$.

As shown in the left panel of Fig. 5, the pressure monotonically increases by decreasing $\delta$ as expected. We display in the right panel of Fig. 5 b) the behavior of the distribution of gap given by Eq. (138) which becomes

$$g(r) = \frac{\theta(r)}{\Theta(\delta/\sqrt{2})} \frac{e^{-\frac{(\delta - r)^2}{2}}}{\sqrt{2\pi}}. \tag{234}$$

We see that the peak around $r = 0$ develops by decreasing $\delta$ as expected.

As we found in sec. 7.2 $q = 0$ solution is always stable for $p > 2$ body interactions. For the $p = 2$ case, we find the $q = 0$ solution becomes unstable for $\alpha > \alpha_c(\delta)$ with

$$\alpha_c(\delta) = 2r^{-2}(\delta/\sqrt{2}) = 2\left(\frac{\Theta'(\delta/\sqrt{2})}{\Theta(\delta/\sqrt{2})}\right)^{-2}, \tag{235}$$

which is obtained from Eq. (142). The critical line $\alpha_c(\delta)$ is displayed in Fig. 6.

### 10.1.3 Jamming within the RS ansatz

It is possible to look for glass transition within the RS ansatz by looking for $q \neq 0$ solution of the RS saddle point equation given by Eq. (135), which must solve $\mathcal{G}(q) = 0$. In sec 10.4.1 we will examine the phase diagram within the RS ansatz for $p = 2$ case. Here we instead focus on the jamming limit where the EA order parameter saturates $q_{EA} = q \to 1$ signaling vanishing of the thermal fluctuations.

The location of the jamming point can be analyzed as follows. We find $\mathcal{G}(q)$ given in Eq. (230) becomes in the limit $q \to 1$,

$$
\begin{aligned}
\lim_{q \to 1} \mathcal{G}(q) &= 1 - \frac{\alpha}{2p} \lim_{q \to 1} (1-q) \int \mathcal{D}z_0 \, r^2(x)\Big|_{x = \frac{\delta - z_0}{\sqrt{2(1-q^p)}}} \\
&= 1 - \frac{\alpha}{p^2} \int_0^\infty \frac{dy}{\sqrt{2\pi}} e^{-(\delta+y)^2/2} y^2.
\end{aligned} \tag{236}
$$

In the last equation we used the asymptotic behavior of the function $r(x)$ given in Eq. (232). Thus we find the jamming line $\alpha = \alpha_j(\delta)$,

$$\alpha_j(\delta) = \frac{p^2}{\int_0^\infty \frac{dy}{\sqrt{2\pi}} e^{-(\delta+y)^2/2} y^2}, \tag{237}$$

which is also displayed in Fig. 6.

The pressure given by Eq. (137) becomes for the hardcore model,

$$\Pi = \frac{1}{\sqrt{2}} \int \mathcal{D}z_0 \, r(x)\Big|_{x=\frac{\delta-\sqrt{q^p}z_0}{\sqrt{2(1-q^p)}}} \xrightarrow{q\to 1} \int_\delta^\infty \mathcal{D}z_0 \frac{(z_0-\delta)}{\sqrt{1-q^p}} \propto \frac{1}{\sqrt{1-q}}, \tag{238}$$

where we used the asymptotic expansion given by Eq. (232). Thus as expected the pressure diverges by jamming (see Fig. 7 b)).

Next let us examine the distribution of the gap given by Eq. (138) which becomes for the hardcore model,

$$g(r) = \theta(r) \int \mathcal{D}z_0 \frac{\gamma_{1-q^p}(\delta - \sqrt{q^p}z_0 - r)}{\Theta\left(\frac{\delta-\sqrt{q^p}z_0}{\sqrt{2(1-q^p)}}\right)}. \tag{239}$$

The behavior in the jamming limit $q \to 1$ can be viewed in the following two ways (see Fig. 7 c)) much as in the case of hardspheres [16],

1. For *fixed* finite $r$, sending $q \to 1$, we find

$$\lim_{q\to 1} g(r) = \theta(r) \frac{e^{-\frac{(\delta-r)^2}{2}}}{\sqrt{2\pi}}. \tag{240}$$

   This is because $\gamma_{1-q^p}(\delta - \sqrt{q^p}z_0 - r)$ becomes a delta function in the $q \to 1$ limit and $\lim_{X\to\infty} \Theta(X) = 1$.

2. In the vanishing region around $r = 0$ parametrized as $r = (1-q^p)\lambda$ we find a different behavior as follows. Assuming $q \sim 1$ we find for $r > 0$,

$$g(r) \sim \int_\delta^\infty \frac{dz_0}{\sqrt{2\pi}} \frac{e^{-\frac{(\delta-z_0-r)^2}{2(1-q^p)}}}{2\pi(\sqrt{1-q^p})} 2\sqrt{\pi}|X|e^{X^2}\Big|_{X=\frac{\delta-\sqrt{q^p}}{2\sqrt{1-q^p}}} \xrightarrow{q\to 1, \text{fixed }\lambda} \dots$$

$$\dots \xrightarrow{q\to 1, \text{fixed }\lambda} \frac{1}{1-q^p} \int_0^\infty \frac{dy}{\sqrt{2\pi}} e^{-\frac{(\delta+y)^2}{2}} y e^{-\lambda y}, \qquad \lambda = \frac{r}{1-q^p}. \tag{241}$$

   In the 1st equation we dropped contribution from $\int_{-\infty}^\delta dz_0..$ which can be neglected compared with the contribution from $\int_\delta^\infty dz_0..$ and used the asymptotic behavior of the error function given ny Eq. (228) which implies $\Theta(-X) \sim \frac{e^{-X^2}}{2\sqrt{\pi}X}$ for $X \ggg 1$. Thus in the jamming limit $q \to 1$ we find diverging peak in the "contact region" around $r = 0$ whose height diverging as $1/(1-q)$ and the width vanishing as $1-q$.

## 10.2 $k$-step RSB solution

We study now the $k$-RSB solution discussed in sec. 8 in the case of the hardcore model.

### 10.2.1 Inputs

Here we present some necessary inputs to study the glass phase and jamming of the hardcore model within generic $k$-RSB ansatz. Within the $k$-RSB ansatz, jamming means $q_{\text{EA}} = q_k \to 1$ (see sec. 8.1). With the following inputs, 1RSB solution can be obtained following sec. 8.4.2 and generic $k$-RSB solution can be obtained following sec. 8.4.3.

For the hardcore potential given by Eq. (224) we find

$$f(m_{k+1}, h) = -\ln\Theta\left(\frac{h}{\sqrt{2\Lambda_{k+1}}}\right), \tag{242}$$

where $\Theta(x)$ is defined in Eq. (226). Then

$$\pi(m_{k+1}, h) = \frac{1}{\sqrt{2\Lambda_{k+1}}} \frac{\Theta'\left(\frac{h}{\sqrt{2\Lambda_{k+1}}}\right)}{\Theta\left(\frac{h}{\sqrt{2\Lambda_{k+1}}}\right)}. \tag{243}$$

The functions $f(m_i, h)$ and $\pi(m_i, h)$ are determined via recursion formulas given by Eq. (161) and Eq. (197) using the boundary values obtained above.

It is useful to study the asymptotic behavior of the functions $f(m_i, h)$ and $\pi(m_i, h)$ in the limit $h \to -\infty$ both for numerical and analytical purposes. Using Eq. (226) and Eq. (228) and the recursion formula given by Eq. (161) one finds for $i = 1, 2, \ldots, k+1$,

$$f(m_i, h) = \begin{cases} 0 & h \to \infty, \\ \frac{h^2}{2\tilde{\Lambda}_i} & h \to -\infty, \end{cases} \qquad \pi(m_i, h) = \begin{cases} 0 & h \to \infty, \\ -\frac{h}{\tilde{\Lambda}_i} & h \to -\infty, \end{cases} \tag{244}$$

where we introduced

$$\tilde{\Lambda}_i \equiv \sum_{j=i}^{k+1} m_j \Lambda_j. \tag{245}$$

Note that $\tilde{\Lambda}_{k+1} = m_{k+1}\Lambda_{k+1} = \Lambda_{k+1} = 1 - q_k^p$. In the continuous limit $k \to \infty$ this implies,

$$f(x, h) = \begin{cases} 0 & h \to \infty, \\ \frac{h^2}{2\tilde{\Lambda}(x)} & h \to -\infty, \end{cases} \qquad \pi(x, h) = \begin{cases} 0 & h \to \infty, \\ -\frac{h}{\tilde{\Lambda}(x)} & h \to -\infty, \end{cases} \tag{246}$$

with

$$\tilde{\Lambda}(x) = 1 - \int_x^1 dy\, \lambda(y) - x\lambda(x), \tag{247}$$

with $\lambda(x) = q^p(x)$ defined in Eq. (179).

The above observation suggests us to introduce a function $j(m_i, h)$ defined as

$$-f(m_i, h) \equiv -\frac{h^2}{2\tilde{\Lambda}_i}\theta(-h) + j(m_i, h). \tag{248}$$

From Eq. (161) we find that the function $j(m_i, h)$ follows a recursion relation

$$j(m_i, h) = \frac{1}{m_i}\ln\int dy\, K_{i,i+1}(h, y)e^{m_i j(m_{i+1}, y)}, \tag{249}$$

with

$$K_{i,i+1}(y, h) \equiv \frac{1}{\sqrt{2\pi\Lambda_i}}\exp\left[-\frac{(h-y)^2}{2\Lambda_i} - \frac{m_i}{2}\frac{y^2}{\tilde{\Lambda}_{i+1}}\theta(-y) + \frac{m_i}{2}\frac{h^2}{\tilde{\Lambda}_i}\theta(-h)\right], \tag{250}$$

and the boundary condition

$$j(m_{k+1}, h) = \ln \Theta\left(\frac{h}{\sqrt{2\Lambda_{k+1}}}\right) + \frac{h^2}{2\tilde{\Lambda}_{k+1}}\theta(-h). \tag{251}$$

Correspondingly one finds that Eq. (197) becomes

$$\pi(m_i, h) = \int dy\, \pi(m_{i+1}, y) K_{i,i+1}(h, y) e^{m_i(j(m_{i+1}, y) - j(m_i, y))}. \tag{252}$$

### 10.2.2 Rescaled quantities useful close to jamming

Let us show below how to modify the numerical algorithm in sec 8.4.3 to solve the continuous RSB equations close to jamming, where $q_{\text{EA}} = q_k \to 1$. To this end let us first introduce several rescaled quantities.

As mentioned above jamming within the $k$-RSB ansatz means

$$\Delta \equiv 1 - q_k \to 0. \tag{253}$$

Then it is convenient to introduce the following rescaled quantities,

$$\gamma_i = \frac{G_i}{\Delta}, \qquad i = 0, 1, 2, \dots, k \tag{254}$$

with $G_i$ being defined in Eq. (151). Note that

$$\gamma_k = 1, \tag{255}$$

since $G_k = 1 - q_k = \Delta$. We replace $0 = m_0 < m_1 < m_2 < \dots < m_k < m_{k+1} = 1$ by,

$$y_i = \frac{m_i}{\Delta}, \qquad i = 0, 1, 2, \dots, k, k+1. \tag{256}$$

In terms of these we can write (Eq. (152)),

$$q_i = 1 - \Delta + \sum_{j=i+1}^{k} \frac{1}{y_j}(\gamma_j - \gamma_{j-1}), \qquad i = 0, 1, 2, \dots, k \tag{257}$$

which in turn implies

$$\gamma_i = \gamma_{i+1} + y_{i+1}(q_{i+1} - q_i), \qquad i = 0, 1, 2, \dots, k-1. \tag{258}$$

Let us also introduce

$$\begin{aligned}
\hat{f}(y_i, h) &= \Delta f(m_i, h), \\
\hat{j}(y_i, h) &= \Delta j(m_i, h), \qquad i = 1, 2, \dots, k+1. \\
\hat{\pi}(y_i, h) &= \Delta \pi(m_i, h),
\end{aligned} \tag{259}$$

Then Eq. (248) becomes

$$-\hat{f}(y_i, h) = -\frac{h^2}{2}\frac{1}{\hat{\Lambda}_i}\theta(-h) + \hat{j}(y_i, h), \tag{260}$$

with

$$\hat{\Lambda}_i \equiv \frac{\tilde{\Lambda}_i}{\Delta} = \sum_{j=i}^{k+1} y_j \Lambda_j, \qquad \Lambda_j = q_j^p - q_{j-1}^p, \tag{261}$$

where we used Eq. (245), Eq. (154) and Eq. (155). The recursion given by Eq. (249) and Eq. (252) becomes

$$
\hat{j}(m_i, h) = \frac{1}{y_i} \ln \int dh' K_{i,i+1}(h, h') e^{y_i \hat{j}(y_{i+1}, h')}
$$
$$
\hat{\pi}(y_i, h) = \int dh' \hat{\pi}(y_{i+1}, h') K_{i,i+1}(h, h') e^{y_i (\hat{j}(y_{i+1}, h') - \hat{j}(y_i, h'))}
$$
(262)

for $i = 0, 1, \ldots, k$ while Eq. (250) becomes

$$
K_{i,i+1}(h', h) \equiv \frac{1}{\sqrt{2\pi\Lambda_i}} \exp\left[ -\frac{(h - h')^2}{2\Lambda_i} - \frac{y_i}{2} \frac{(h')^2}{\hat{\Lambda}_{i+1}} \theta(-h') + \frac{y_i}{2} \frac{h^2}{\hat{\Lambda}_i} \theta(-h) \right].
$$
(263)

The boundary condition given by Eq. (251) and Eq. (243) becomes

$$
\hat{j}(y_{k+1} = 1/\Delta, h) = \Delta \ln \Theta\left( \frac{h}{\sqrt{2\Delta \hat{\Lambda}_{k+1}}} \right) + \frac{h^2}{2\hat{\Lambda}_{k+1}} \theta(-h) \xrightarrow{\Delta \to 0} 0,
$$
$$
\hat{\pi}(y_{k+1} = 1/\Delta, h) = \frac{\Delta}{\sqrt{2\Delta \hat{\Lambda}_{k+1}}} \left.\frac{\Theta'(x)}{\Theta(x)}\right|_{x = \frac{h}{\sqrt{2\Delta \hat{\Lambda}_{k+1}}}} \xrightarrow{\Delta \to 0} -\frac{h}{p} \theta(-h).
$$
(264)

Here we used Eq. (259) and the asymptotic expansions Eq. (229) and Eq. (232). We also used Eq. (261) and Eq. (154) which imply $\hat{\Lambda}_{k+1} = \tilde{\Lambda}_{k+1}/\Delta = \Lambda_{k+1}/\Delta = (1 - q_k^p)/\Delta \xrightarrow{\Delta \to 0} p$.

On the other hand the recursion formula given by Eq. (189) becomes

$$
P(y_j, h) = e^{-y_j \hat{f}(y_{j+1}, h)} \gamma_{\Lambda_j} \otimes_h \frac{P(y_{j-1}, h)}{e^{-y_j f(y_j, h)}}, \qquad j = 1, 2, \ldots, k+1,
$$
(265)

and the boundary condition given by Eq. (190) becomes

$$
P(y_0, h) = \frac{1}{\sqrt{2\pi\Lambda_0}} e^{-\frac{(\delta - h)^2}{2\Lambda_0}}, \qquad \Lambda_0 = q(0)^p.
$$
(266)

Finally Eq. (204) and Eq. (205) become,

$$
\frac{q_0}{\gamma_0^2} = \alpha \hat{\kappa}_0^0,
$$
$$
\frac{1}{\gamma_i} - \frac{1}{\gamma_0} = \alpha \left( \sum_{j=0}^{i-1} (y_j - y_{j+1}) \hat{\kappa}_j^0 + y_i \hat{\kappa}_i^0 \right) \qquad i = 1, 2, \ldots, k,
$$
(267)

$$
\hat{\kappa}_i^0 \equiv q_i^{p-1} \int dh P(y_i, h) \hat{\pi}^2(y_{i+1}, h) \qquad i = 0, 1, 2, \ldots, k.
$$
(268)

### 10.2.3 Algorithm to solve the continuous RSB equations close to jamming

The saddle point equations for a generic finite $k$-RSB ansatz with some fixed values of $0 < m_1 < m_2 < \ldots < m_k < 1$ can be solved numerically as explained in sec. 8.4.3. We can modify it using the rescaled quantities.

   0. Make some guess for the initial values of $q_i$ ($i = 0, 1, 2, \ldots, k$).

   1. Compute $\Delta$ as $\Delta = 1 - q_k$.

2. Given $\Delta$ we have $y_i = m_i/\Delta$ $(i = 1, 2, \ldots, k)$. Note that $y_0 = 0$ and $y_{k+1} = 1/\Delta$. Then compute $\gamma_i$ for $i = 0, 1, \ldots, k-1$ using Eq. (258). Note that $\gamma_k = 1$.

3. Compute $\hat{j}(y_i, h)$ and $\hat{\pi}(y_i, h)$ recursively for $i = k, \ldots, 1$ using Eq. (262) with the boundary condition given by Eq. (264).

4. Compute functions $P(m_i, h)$ recursively for $i = 1, \ldots, k+1$ using Eq. (265) with the boundary condition given by Eq. (266).

5. Compute $\hat{\kappa}_i^0$ for $i = 0, 1, \ldots, k$ using Eq. (268), $\gamma_i$ for $i = 0, 1, 2, \ldots, k$ using using Eq. (267) and finally $q_i$ for $i = 0, 1, 2, \ldots, k$ using Eq. (257). Finally compute $\Lambda_i$ and $\hat{\Lambda}_i$ for $i = 1, 2, \ldots, k+1$ using Eq. (261).

6. Return to 1.

The above steps 1-6 must be repeated until the solution converges.

### 10.2.4 Algorithm to look for the jamming point

We can also look for the $k$-RSB solution for a given, fixed $\Delta$. This can be seen as the following. In the step 5 of the procedure explained above we obtain $\gamma_k$ using Eq. (267) but $\gamma_k = 1$. Thus Eq. (267) for $i = k$ can be considered as an equation to determine $\alpha = \alpha(\Delta)$. In particular, the jamming point $\alpha_j(\delta)$ via $k$-RSB ansatz can be determined by choosing $\Delta = 0$.

## 10.3 Jamming criticality

Let us discuss properties of the system approaching the jamming in the case of continuous RSB, i. .e. $k \to \infty$. As mentioned in sec. 8.1, we expect the $q(x)$ function of the continuous RSB solution has a continuous part for some range $x < x_1$ and a plateau $q(x) = q(x_1) = q_{\mathrm{EA}}$ for $x_1 < x < 1$. Jamming means $q_{\mathrm{EA}} = q(x_1) \to 0$ in the continuous RSB. For a convenience we define

$$\Delta(x) \equiv 1 - q(x). \tag{269}$$

Then jamming implies $\Delta_1 = 1 - q(x_1) \to 0$. We discuss below properties of the system encoded in the continuous RSB solution in the vicinity of the core $x \to x_1$ which encodes physical properties of the system in the deepest part of the energy landscape.

In the following we will find results very similar to those found in the hardspheres in $d \to \infty$ [16] where it was shown that continuous RSB solution gives a qualitatively different result from finite $k$-RSB ansatz concerning the scaling behavior approaching jamming.

### 10.3.1 Scaling ansatz at the core $x \to x_1$ in the jamming limit $\Delta_1 \to 1$.

Following [16] and [35] we consider the following scaling ansatz at the core $x \to x_1$ in the jamming limit $\Delta_1 \to 0$,

$$\Delta(x)/\Delta_1 \simeq (x/x_1)^{-\kappa}, \tag{270}$$

with an exponent $\kappa > 0$.

From Eq. (192) and Eq. (199) we have,

$$\dot{P}(x, h) = \frac{\dot{\lambda}(x)}{2} \left[ P''(x, h) - 2x(P(x, h)\pi(x, h))' \right], \tag{271}$$

$$\dot{\pi}(x, h) = -\frac{\dot{\lambda}(x)}{2} \left[ \pi''(x, h) + 2x\pi(x, h)\pi'(x, h) \right]. \tag{272}$$

Based on the asymptotic behavior of the function $\pi(x,h)$ given in Eq. (246) we expect

$$P(x,h) \simeq \frac{1}{\sqrt{2\pi\lambda(x)}} e^{-(\delta-h)^2/2\lambda(x)}, \qquad h \to +\infty \tag{273}$$

and

$$\dot{P}(x,h) = \frac{\dot{\lambda}(x)}{2}\left[P''(x,h) + 2\frac{x}{\tilde{\Lambda}(x)}(P'(x,h)h + P(x,h))\right], \qquad h \to -\infty. \tag{274}$$

For $x \to x_1$ and $\Delta_1 \to 0$ we can assume

$$\dot{\lambda}(x) \simeq -p\dot{\Delta}(x), \qquad \tilde{\Lambda}(x) \simeq pG(x), \qquad G(x) \simeq \frac{\kappa}{\kappa-1}x\Delta(x), \tag{275}$$

which follow from Eq. (179), Eq. (247), Eq. (269) and Eq. (270). Then assuming $P(x,h) \simeq A(x)e^{B(x)h - C(x)h^2/2}$ for $x \to x_1$ one finds, $A \propto \Delta^{-(1-1/\kappa)}$, $B \propto \Delta^{-(1-1/\kappa)}$ and $C \propto \Delta^{-2(1-1/\kappa)}$. This implies the following scaling form for $x \to x_1$,

$$P(x,h) \sim \Delta^{-\frac{\kappa-1}{\kappa}}P_0(h\Delta^{-\frac{\kappa-1}{\kappa}}), \qquad h \to -\infty, \tag{276}$$

with some scaling function $P_0(x)$.

To sum up we can expect the following three regimes [35] [16] for $x \to x_1$:

(0) $h \to -\infty$: Eq. (276) and Eq. (246) read

$$P(x,h) \sim \Delta^{-c/\kappa}P_0(h\Delta^{-c/\kappa}), \qquad \pi(x,h) \quad \sim -\frac{h}{\tilde{\Lambda}(x)}, \tag{277}$$

with

$$c = \kappa - 1. \tag{278}$$

(1) $h \sim 0$ (intermediate regime)

$$P(x,h) \sim \Delta^{-a/\kappa}P_1(h\Delta^{-b/\kappa}), \qquad \pi(x,h) \sim \frac{\Delta^{b/\kappa}}{\tilde{\Lambda}(x)}\pi_1(h\Delta^{-b/\kappa}). \tag{279}$$

(2) $h \to \infty$: Eq. (273) ($\lambda(x) \to 1$ for $x \to x_1$ and $q(x_1) \to 1$) and Eq. (246) implies

$$P(x,h) \sim P_2(h), \qquad \pi(x,h) \sim 0. \tag{280}$$

In the above equations $P_0(x), P_1(x), \pi_1(x)$ and $P_2(x)$ are some smooth functions and $a$, $b$, $c$, $\kappa$ are some exponents. In the following we assume that these exponents are all positive.

Now we can make the following observations:

1. Matching between (0) and (1): assuming

$$P_0(u) \propto u^\theta, \qquad u \to 0, \tag{281}$$
$$P_1(u) \propto (-u)^\theta, \qquad u \to -\infty, \tag{282}$$

the following relation is needed,

$$\Delta^{-c/\kappa}(h\Delta^{-c/\kappa})^\theta \sim \Delta^{-a/\kappa}(h\Delta^{-b/\kappa})^\theta, \tag{283}$$

which implies

$$\theta = \frac{c-a}{b-c}. \tag{284}$$

We also find

$$\pi_1(u) \sim -u, \qquad u \to -\infty \tag{285}$$

must hold.

2. Matching between (1) and (2): assuming

$$P_1(z) \quad \propto \quad z^{-\alpha} \qquad z \to \infty \tag{286}$$

$$P_2(z) \quad \propto \quad z^{-\alpha} \qquad z \to 0 \tag{287}$$

we find the following relation is needed to eliminate the dependence on $\Delta$,

$$\alpha = \frac{a}{b} \tag{288}$$

3. Analysis on the intermediate regime $h \sim 0$: Plugging Eq. (279) in Eq. (271) and using Eq. (275) we find, the contribution from the 1st term on the r.h.s. scales are $(\Delta^{-b/\kappa})^2$ while those from the 2nd term on the r.h.s and the term on the l.h.s scales like $\Delta^{-1}$. Thus in order to have a non-trivial solution we need,

$$\frac{b}{\kappa} = \frac{1}{2}. \tag{289}$$

by which we can eliminate $b$. Now we are left with two exponents $a$ and $c = \kappa - 1$. Furthermore plugging Eq. (279) in Eq. (271) and Eq. (272) we find the following two ordinary differential equations,

$$\frac{a}{\kappa} P_1(z) + \frac{z}{2} P_1'(z) \quad = \quad \frac{p}{2} P_1''(z) - \frac{c}{\kappa} (P_1(z) \pi_1(z))' \tag{290}$$

$$\left( \frac{1}{2} - \frac{c}{\kappa} \right) \pi_1(z) - \frac{1}{2} z \pi_1'(z) \quad = \quad \frac{p}{2} \pi_1''(z) + \frac{c}{\kappa} \pi_1(z) \pi_1'(z) \tag{291}$$

which are subjected to the boundary condition

$$P_1(z) = \begin{cases} (-z)^\theta & z \to -\infty \\ z^{-\alpha} & z \to \infty \end{cases} \qquad \pi_1(z) = \begin{cases} -z & z \to -\infty \\ 0 & z \to \infty \end{cases} \tag{292}$$

One can check that the differential equations given by Eq. (291) with the boundary condition given by Eq. (292) is consistent with the scaling relations for $\theta$ and $\alpha$ given by Eq. (284) and Eq. (288).

Here we notice that the apparent dependence on $p$ in Eq. (291) can be formally eliminated by the following replacement,

$$\frac{z}{\sqrt{p}} \to z \qquad \frac{P_1(z)}{\sqrt{p}} \to P_1(z) \qquad \frac{\pi_1(z)}{\sqrt{p}} \to \pi_1(z) \tag{293}$$

This means that if we find a solution for the $p = 1$ case, the solutions for other values of $p$ can be obtained as well using Eq. (293) in the reversed manner. Importantly such a solution satisfies the same desired asymptotic behaviors given by Eq. (292). This implies the universality does not change with $p$.

However as pointed out in [16] the above equations do not completely solve the problem. We are left with the exponent $a$ undetermined while other quantities $P_1(z)$, $\pi_1(z)$ and the exponent $c$ can be obtained in a form parametrized by $a$. (All other exponents are fixed given $a$ and $c$.) Then the final task to fix the value of the exponent $a$ which can be done using the exact identity given by Eq. (217). The latter reads in the limit $x \to x_1$ and $q(x_1) = q(x_1) \to 1$

$$0 = (p - 1) \int dh\, T_1(h) + \int dh\, T_2(h) \tag{294}$$

where we defined,

$$
\begin{aligned}
T_1(h) &\equiv [(p-2)G^2(x_1)-2x_1 G(x_1)]P(x_1,h)\pi^2(x_1,h) \\
&+ 3pG^2(x_1)P(x_1,h)(\pi'(x_1,h))^2 && (295) \\
T_2(h) &\equiv p(-2x_1 G(x_1))P(x_1,h)(\pi'(x_1,h))^2 \\
&+ p^2 G^2(x_1)P(x_1,h)\big[(\pi''(x_1,h))^2 - 2x_1(\pi'(x_1,h))^3\big]. && (296)
\end{aligned}
$$

We notice that the contribution of $(p-1)\int dh T_1(h)$ into Eq. (294) vanishes for the $p=1$ case accidentally but not for $p>1$. Thus we must carefully examine whether $\int dh T_1(h)$ remain relevant in the jamming limit $\Delta_1 = \Delta(x_1) \to 0$ or not.

We examine contributions of the integrals $\int dh T_1(h)$ and $\int dh T_2(h)$ from the regimes (1) $h \to -\infty$ and (2) $h \sim 0$. In the regime (3) $h \to \infty$ $\pi(x,h) \sim 0$ as Eq. (280) so we do not need to consider the regime (3). Using Eq. (277), Eq. (278), and Eq. (275) we find for the regime (0) $h \to -\infty$,

$$
\begin{aligned}
\int_{\text{regime}(0)} dh T_1(h) &\sim -\frac{2}{p^2}x_1\frac{c}{\kappa}\Delta_1^{-(1-c)/\kappa}\int_0^\infty dt P_0(-t)t^2 \\
\int_{\text{regime}(0)} dh T_2(h) &\sim 0 && (297)
\end{aligned}
$$

where we took leading terms for the jamming limit $\Delta_1 \to 0$. Similarly using Eq. (279),Eq. (289) and Eq. (275) we find for the regime (1) $h \sim 0$,

$$
\begin{aligned}
\int_{\text{regime}(1)} dh T_1(h) &\sim \Delta^{1/2-a/\kappa}\int_{-\infty}^\infty dz P_1(z)\left[\frac{3}{p}(\pi_1'(z))^2 - \frac{2}{p^2}x_1\frac{c}{\kappa}\pi_1^2(z)\right] \\
\int_{\text{regime}(1)} dh T_2(h) &\sim \\
&\sim \Delta^{-(1+a)/\kappa}\int_{-\infty}^\infty dz P_1(z)\left[(\pi''(z))^2 - 2\frac{c}{\kappa}\frac{1}{p}\left\{(\pi_1'(z))^3 + (\pi'(z))^2\right\}\right] && (298)
\end{aligned}
$$

Collecting the above results we find the most relevant contribution in the jamming limit $\Delta_1 \to 0$ is given by $\int_{\text{regime}(1)} dh T_2(h)$ as long as the exponents $a, c$ are positive. It means that we must satisfy,

$$
\int_{-\infty}^\infty dz P_1(z)\left[(\pi''(z))^2 - 2\frac{c}{\kappa}\frac{1}{p}\left\{(\pi_1'(z))^3 + (\pi'(z))^2\right\}\right] = 0 \qquad (299)
$$

Again we find the apparent $p$ dependence can be formally eliminated by the replacement given by Eq. (293).

Based on the above analysis we can conclude that the critical exponents and the scaling functions $P_1(z)$, and $\pi_1(z)$ do not dependent on $p$, i. .e. super-universal. The exponents are $a = 0.29213..$, $b = 0.70787...$, $c = 0.41574...$, $\alpha = 0.41269..$, $\theta = 0.42311...$ and $\kappa = 1.41574...$ [16].

### 10.3.2 Divergence of the pressure

The pressure can be expressed as Eq. (201) which reads,

$$
\Pi = \int dh P(x,h)\pi(x,h) \qquad (300)
$$

where $x$ can be chosen arbitrary. Using the scaling ansatz given by Eq. (277) and Eq. (275) at the core $x \to x_1$ and jamming $\Delta_1 \to 0$ we find contribution from largely negative region of $h$ becomes

$$\int_{-\infty}^{0} dh \left(-\frac{1}{p}\right) \frac{\kappa - 1}{\kappa} \frac{h}{\Delta_1} \Delta^{-c/\kappa} P_0(h\Delta_1^{-c/\kappa}) \sim c_{\mathrm{nt}} \Delta^{-1/\kappa} \qquad c_{\mathrm{nt}} = \frac{1}{p} \frac{\kappa - 1}{\kappa} \int_0^{\infty} dt P_0(-t) t. \tag{301}$$

Similarly we can analyze contribution from the region $h \sim 0$ using Eq. (279), and Eq. (275)

$$\int dh \frac{1}{p} \frac{h}{\Delta_1} \Delta_1^{-a/\kappa} P_1(h\Delta^{-b/\kappa}) \propto \Delta^{-a/\kappa}. \tag{302}$$

If $a < 1$, which is the case ($a = 0.29213...$), the latter gives a only sub-dominant contribution. To sum up we find, the 'cage size' $\Delta_1$ vanishing in the jamming limit $\Pi \to \infty$ as,

$$\Delta_1 \propto \Pi^{-\kappa} \tag{303}$$

### 10.3.3 Distribution of gap

For the hardcore model the distribution of the gap $g(r)$ within the $k$-RSB ansatz given by Eq. (194) reads,

$$g(r) = \theta(r) \int \mathcal{D}z \frac{P(m_k, r - \sqrt{1 - q_k^p} z)}{\Theta\left(\frac{r - \sqrt{1 - q_k^p} z}{\sqrt{2(1 - q_k^p)}}\right)} \tag{304}$$

1. For *fixed* finite $r$, sending $q_k \to 1$, we find,

$$g(r) = \theta(r) P(m_k, r) \tag{305}$$

where we used $\lim_{X \to \infty} \Theta(X) = 1$. This is a generalization of the RS ($k = 0$) result given by Eq. (240).

In the $k \to \infty$ limit, the scaling behavior of $P(x, h)$ close to the core $x \to x_1$ as described by Eq. (277) and Eq. (279) in the region vanishing in the jamming limit $\Delta_1 \to 0$ implies development of a delta peak $\delta(r)$. On the other hand, we have the scaling behavior $P(x, h) \sim h^{-\alpha}$ for fixed $h \sim 0^+$ as given by Eq. (287) with $\alpha = a/b$ given by Eq. (288). These observations implies,

$$g(r) \sim \delta(r) + c_{\mathrm{nt}} \theta(r) r^{-\alpha}, \tag{306}$$

where $c_{\mathrm{nt}}$ is some numerical factor.

2. In the vanishing region around $r = 0$ parametrized as $r = (1 - q_k^p)\lambda$ we find, Assuming $q_k \sim 1$ we find for $r > 0$,

$$g(r) \sim \frac{1}{1 - q_k^p} \int_0^{\infty} dy P(m_k, -y) y e^{-\lambda y} \qquad \lambda = \frac{r}{1 - q_k^p} \tag{307}$$

This is a generalization of the RS ($k = 0$) result given by Eq. (240).

Now in the $k \to \infty$ limit we have the scaling behavior $P(x, h) \sim \Delta^{-c/\kappa} P_0(h\Delta^{-c/\kappa})$ for $h < 0$ in Eq. (277). Using this for $x \to x_1$ we find,

$$g(r) \sim \frac{1}{p} \frac{1}{\Delta_1^{1/\kappa}} \int_0^{\infty} dt P_0(-t) t e^{-\frac{t}{p} \frac{r}{\Delta_1^{1/\kappa}}} \tag{308}$$

where we used $c = \kappa - 1$ and $1 - q_k^p \simeq p\Delta_1$ for $\Delta_1 \to 0$.

Using the above result we can evaluate the fraction of interactions or contacts which is closed. For any small but finite $\epsilon$ we have,

$$\int_0^\epsilon dr g(r) = \int_0^\infty dt P_0(-t) \int_0^{\epsilon t/(p\Delta_1^{1/\kappa})} ds e^{-s} \underset{\Delta_1 \to 0}{\to} \int_0^\infty dt P_0(-t) \tag{309}$$

Thus in the jamming limit, the fraction of closed contact given by Eq. (19) can be expressed as,

$$f_{\text{closed}} = \lim_{\epsilon \to 0} \int_0^\epsilon dr g(r) = \int_0^\infty dt P_0(-t) \tag{310}$$

Note that here the lower limit of the integration is set to 0 because of the hardcore constraint.

### 10.3.4 Isostaticity

Let us consider whether isostaticity discussed in sec. 2.3 holds in the jamming limit in the present model. The condition of isostaticity given by Eq. (21) becomes in the $M \to \infty$ limit with $\alpha = c/M$ fixed at jamming $\Delta_1 \to 0$ becomes,

$$1 = \frac{\alpha}{p} \int_0^\infty dt P_0(-t) \tag{311}$$

where we used Eq. (310).

Actually using the exact identity given by Eq. (216) which holds for the continuous RSB together with the scaling behavior given by Eq. (277) in the $h < 0$ region and the relation $\tilde{\Lambda}(x) \simeq pG(x)$ given by the 2nd equation of Eq. (275) which hold close to the core $x \to x_1$ at jamming $\Delta_1 \to 0$ we find,

$$1 = \frac{\alpha(p-1)}{p^2} \int_{-\infty}^0 dh \Delta_1^{-c/\kappa} P_0(h\Delta_1^{-c/\kappa}) h^2 + \frac{\alpha}{p} \int_{-\infty}^0 dh \Delta_1^{-c/\kappa} P_0(h\Delta_1^{-c/\kappa}) \underset{\Delta_1 \to 0}{\to} \frac{\alpha}{p} \int_0^\infty dt P_0(-t) \tag{312}$$

Thus we see that the isostaticity holds at jamming. Note that the term which is proportional to $p-1$ apparently violates the isostaticity but it scales as $\Delta_1^{2c/\kappa}$ and becomes irrelevant in the jamming limit $\Delta_1 \to 0$ as long as $c/\kappa > 0$.

## 10.4 Detailed analysis on the $p = 2$ case

Let us take here the $p = 2$ case and study the model more in detail to work out the phase diagram and behavior of physical quantities. The following analysis is valid for the disorder-free model in the range $\delta > 0$ because of the flatness of the potential as we noted at the beginning of sec 10. We have found that the system exhibit continuous transition to anti-ferromagnetic phase for $\delta < 0$ at $T_c$ given by Eq. (57). We have to suppress the anti-ferromagnetic phase using the disordered model in order to realize the glassy phases for $\delta < 0$.

### 10.4.1 RS solution

For the $p = 2$ case, we find from Eq. (235) that the paramagnetic solution $q = 0$ becomes unstable at the critical point $\alpha_c(\delta)$. Then we are naturally led to examine the possibility of the $q \neq 0$ solution. Within the RS ansatz, it must solve $\mathcal{G}(q) = 0$ (see Eq. (135)), where the

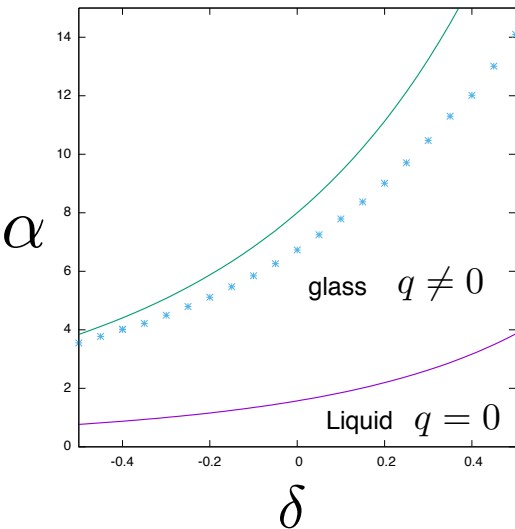

Figure 6: The phase diagram of the $p = 2$ body hardcore model within the replica symmetric (RS) ansatz: $q > 0$ RS solution emerges continuously at the lower curve which represents $\alpha = \alpha_c(\delta)$ given by Eq. (235). The value of the order parameter saturates $q \to 1$ approaching the upper solid line $\alpha = \alpha_j(\delta)$ given by Eq. (237), which is the jamming line within the RS ansatz. The lower curve coincides with the AT line above which the RS solution becomes unstable. The dotted line is the jamming line obtained by the continuous RSB solution which is discussed later.

function $\mathcal{G}(q)$ for the hardcore model is given by Eq. (230). Expanding $\mathcal{G}(q)$ up to order $O(q^2)$ we find,

$$0 = \mathcal{G}(q) = 1 - \frac{\alpha}{\alpha_c(\delta)} \left[ 1 - 2q + (2 - 2x_0 - r_0)q^2 \right] + O(q^4) \tag{313}$$

where

$$x_0 \equiv \frac{\delta}{\sqrt{2}} \qquad r_0 \equiv r(x_0) = \frac{\Theta'(x_0)}{\Theta(x_0)} \tag{314}$$

The above equation can be solved for $q$ to find,

$$q = \frac{1}{2}\epsilon - \frac{1}{4}\left(1 + x_0 + \frac{r_0}{2}\right)\epsilon^2 + O(\epsilon^3) \qquad \epsilon \equiv \frac{\alpha}{\alpha_c(\delta)} - 1 \tag{315}$$

Thus we find that $q \neq 0$ solution emerges at the critical point $\alpha = \alpha_c(\delta)$, where the $q = 0$ solution becomes unstable, and the EA order parameter $q$ grows continuously increasing $\alpha$.

In Fig. 6 we show the phase diagram for the $p = 2$ hardcore model within the RS ansatz. The glass transition line $\alpha = \alpha_c(\delta)$ is given by Eq. (235). The jamming line $\alpha = \alpha_j(\delta)$ is given by Eq. (237).

In Fig. 7, we display an example of a set of solutions of the RS saddle point equation given by Eq. (135) with Eq. (230) for a $\alpha = 4$ with varying $\delta$. As shown in the panel a), the glass order parameter $q$ emerges continuously at the critical point $\delta_c \sim 0.51$ (determined by $\alpha_c(\delta_c) = 4$, see Fig. 6) and increases by decreasing $\delta$ and saturates to $q = 1$ approaching the jamming point $\delta_j \sim -0.47065$ (determined by $\alpha_j(\delta) = 4$, see Fig. 6) There we also show the behavior of the pressure given by Eq. (238) which diverges approaching the jamming and evolution of $g(r)$ given by Eq. (239) which develops a diverging contact peak at $r = 0$ approaching the jamming.

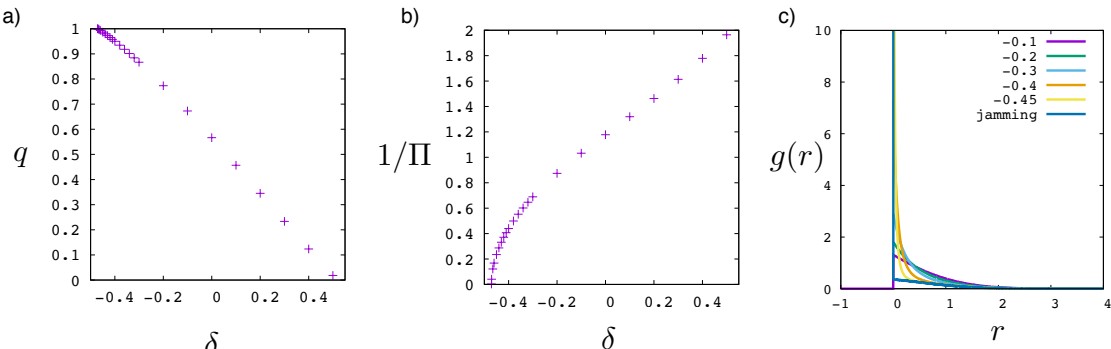

Figure 7: Glass phase of the hardcore model within the RS ansatz (here we choose $\alpha = 4$): a) behavior of the order parameter $q$, b) inverse of the pressure $\Pi$, c) the distribution of the gap $g(r)$. The line labeled as *jamming* is for $\delta_j \sim -0.47065$.

Finally let us examine the stability of this solution. From the result reported in appendix B.1.3 we find the replicon eigenvalue $\lambda_R$ of the RS solution as,

$$\lambda_{\mathrm{R}} = \frac{2}{(1-q)^2}\mathcal{R}(q) \tag{316}$$

$$\mathcal{R}(q) = 1 - \frac{\alpha}{2(1+q)^2}\int \mathcal{D}z_0 \left( r^2(x)(1-q^2) + 4q^2x^2r^2(x) + xr^3(x) + \frac{r^4(x)}{4} \right)_{x=\frac{\delta-\sqrt{q^p}z_0}{\sqrt{2(1-q^p)}}} \tag{317}$$

$$= 1 - \frac{\alpha}{\alpha_c(\delta)}\left[ 1 - 2q + \frac{1}{2}(5r_0^2 + 16r_0x_0 + 12x_0^2 + 2)q^2 + O(q^3) \right] \tag{318}$$

In the last equation we made an expansion in series of $q$ which can be obtained by using $r'(x) = -2xr(x) - r^2(x)$ which follows from Eq. (231). Comparing the function $\mathcal{G}(q)$ in Eq. (313) and $\mathcal{R}(q)$ in Eq. (318) we notice that they are identical up to $O(q)$ but different in the $O(q^2)$ terms. Using Eq. (315) in Eq. (318) we find up to $O(\epsilon^2)$,

$$\lambda_{\mathrm{R}} = \frac{2}{(1-q)^2}A(x_0)\epsilon^2 \qquad A(x) = \frac{1}{4} - \frac{x}{2} - \frac{r(x)}{4} - \frac{5}{8}r^2(x) - 2xr(x) - \frac{3}{2}x^2 \tag{319}$$

It turns out that $A(x)$ is a monotonically decreasing function of $x$. Using the asymptotic behavior of the function $r(x)$ given in Eq. (232) one can find $\lim_{x\to-\infty} A(x) = -1/4$. Thus we find that the replicon eigenvalue is definitely negative meaning that the RS solution is unstable for $\alpha > \alpha_c(\delta)$. We also checked numerically, solving $\mathcal{G}(q) = 0$ for $q$ and evaluating $\lambda_R$ (Eq. (317)) that this is indeed the case in the whole regime of $\alpha > \alpha_c(\delta)$. Thus the replica symmetry must be broken for $\alpha > \alpha_c(\delta)$.

Remarkably the situation is very similar to the Sherrington-Kirkpatrick (SK) model for the spin glasses [47, 56, 59]. To summarize we find the liquid solution described by the $q = 0$ RS solution which becomes unstable approaching the critical point $\alpha_c(\delta)$ where all eigenvalues of the Hessian matrix vanish. It immediately means divergence of the so called spin-glass susceptibility and negative divergence of non-linear compressibility $d^2p/d\delta^2$ much as the spinglass transition of the SK model [52, 59]. The line $\alpha = \alpha_c(\delta)$ is the equivalent of the d'Almeida-Thouless (AT) line [56]. Beyond the transition point, going into the glass phase, we have to consider breaking of the replica symmetry. [50, 66, 67].

### 10.4.2 RSB solution

Finally let us study the glass phase of the $p = 2$ hardcore model using the RSB ansatz. Here we use the softcore potential given by Eq. (14) and extend the analysis to finite temperatures.

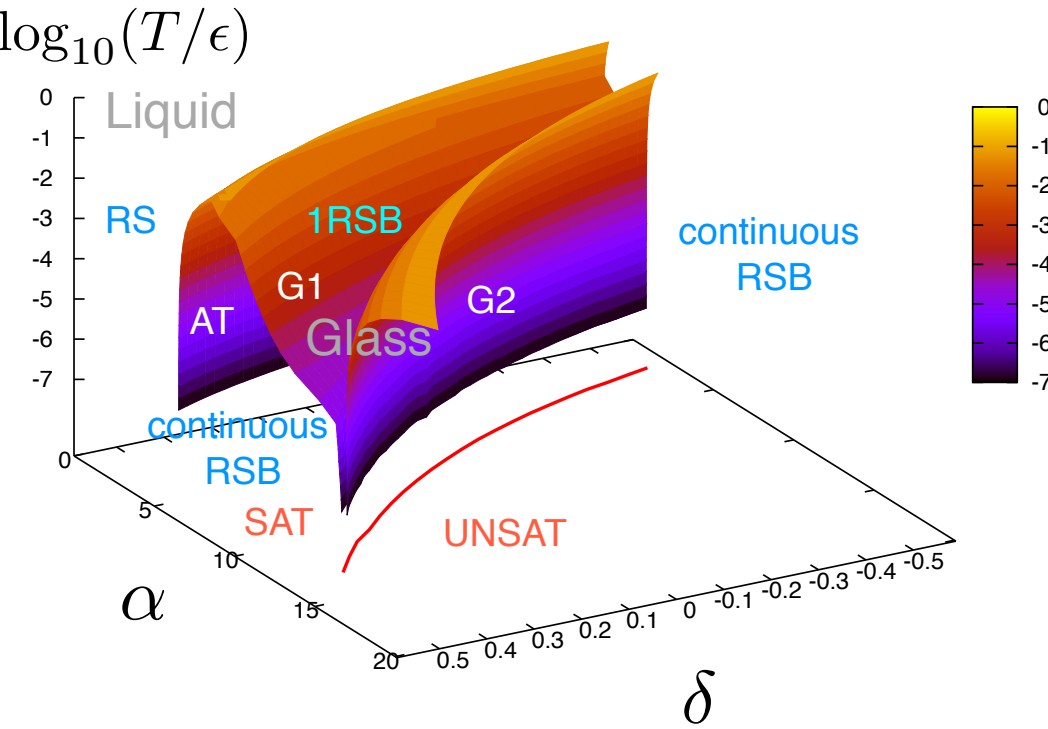

Figure 8: Phase diagram of the soft/hardcore model ($p = 2$). On the plane AT separating the liquid (RS) and glass (RSB) the d'Almeida-Thouless (AT) instability occurs. The 1RSB solution becomes unstable below the two planes G1 and G2 on which the Gardner transition occurs. The G1 plane separates from the AT plane at finite temperatures. The red line on the bottom represents the jamming line $\alpha = \alpha_j(\delta)$ at $T = 0$.

Using Eq. (142) and evaluating the integral numerically using the softcore potential we can easily find the plane $\alpha = \alpha_c(\delta, T)$ where the AT instability $\lambda_R = 0$ occurs. The result is shown in Fig. 8 where the AT plane is indicated as 'AT'. The zero temperature limit of it agrees with the AT line of the hardcore model shown in Fig. 6 as it should be. The AT plane separates the liquid phase (paramagnet) with $q = 0$ on its left hand side and glass the glass phase on the right hand side.

Next let us examined the 1RSB ansatz on the glass side of the AT plane. We solved the 1RSB equations numerically following the scheme explained in sec. 8.4.2 to obtain $q(x_1)$ assuming $q_0 = 0$. Note that $q_0 = 0$ always solves the saddle point equation for $p > 1$ as we mentioned in sec. 8.4. We found $q(x_1)$ emerges continuously starting from the AT plane, as expected. Then we evaluated the complexity $\Sigma^*(m_1)$ numerically (see sec. 8.2.2) and determined $m_1$ where the complexity vanishes. We examined the stability of the 1RSB solution by evaluating the replicon eigenvalue $\lambda_R$ of the 1RSB solution given by Eq. (218). As shown in Fig. 8. we have two planes indicated as 'G1' and 'G2' where the replicon eigenvalue vanishes suggesting the Gardner's transition [54]. We found 'G1' plane merges with the AT plane at higher temperatures as can be seen in the figure. The 1RSB solution is stable above these Gardner planes but become unstable below them.

Below the Gardner planes and on the right on the glass side of the AT plane we naturally expect continuous RSB. Indeed we obtain the continuous RSB solution (approximated by $k = 200$ RSB) as shown in Fig. 9 where we show some examples of the $q(x)$ functions obtained

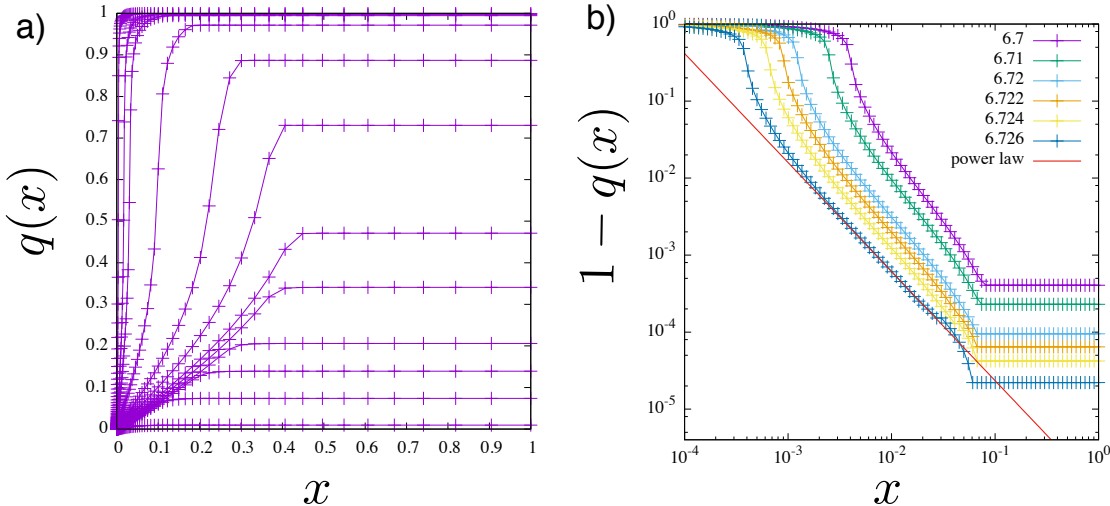

Figure 9: The $q(x)$ function of the hardcore model with $p = 2$, $\delta = 0$ for which $\alpha_c = 1.5708..$ and $\alpha_j = 6.732...$ a) $\alpha = 1.6, 1.8, 2.0, 2.2, 2.4, 2.6, 2.8, 3.0, 4.0, 5.0, 6.0, 6.5, 6.6, 6.7$ from the bottom to the top. b) The straight line represents the power law fit $ax^{-\kappa}$ with $\kappa = 1.4157$, the same exponent as that for the hardspheres [16].

at $T = 0$ (hardcore limit). We obtained the result using the scheme explained in sec. 8.4.3 together with the inputs for the hardcore case shown in sec. 10.2.1. As can be seen in Fig. 9 a), we found the continuous RSB solution with nonzero $q(x)$ function emerges continuously starting from the AT line $\alpha = \alpha_c(\delta)$ as expected.

Using the scheme explained in sec. 10.2.4 we obtained the jamming line $\alpha = \alpha_j(\delta)$ of the hardcore model and the result is displayed in Fig. 8 at the bottom. It is also shown in Fig. 6 where we can see that the RS ansatz overestimates the jamming line. Quite interestingly we find in Fig. 8 that the two Gardner planes 'G1' and 'G2' merges onto the the jamming line in the zero temperature limit. The geometry of the phase diagram is very different from that of the hardspheres [68] where there is only one Gardner line. We analyzed the criticality of jamming $q(x_1) \to 1$ of the hardcore model ($\epsilon \to \infty$), i. e. $\alpha \to \alpha_J^-(\delta)$ at $T = 0$. As shown in Fig. 9 b), we find power law behavior $1 - q(x) \propto x^{-\kappa}$ with the expected exponent $\kappa = 1.4157...$ This confirms the scaling argument presented in sec. 10.3.1 establishing that the jamming of the present model belong to the same universality class as that of the hardspheres [16].

The liquid phase $q = 0$ at $T = 0$ can be regarded as an easy SAT region where the space of the solutions to satisfy the hard constraints ($\epsilon \to \infty$), i. e. the manifold of the ground states are continuously connected. The glass phase at $T = 0$ with $0 < q(x_1) < 1$ can be regarded as hard SAT phase where the manifold of the ground states splits into clusters. The major difference with respect to the case of usual discrete coloring [32] is that the transition is continuous and the clustering is hierarchical reflecting the continuous RSB. The region $\alpha > \alpha_j(\delta)$ is the UNSAT region where the hard constraint cannot be satisfied.

# 11 Conclusions

In the present paper we developed a family of exactly solvable large $M$-component vectorial Ising/continuous spin systems with $p$-body interactions which exhibit glass transitions by the self-generated randomness. We also established a connection between the disorder-free model

and a completely disordered spin glasses model by constructing a model which interpolates the two limits. We showed that the supercooled paramagnetic states and glassy states are locally stable against crystallization under certain conditions, namely either 1) $p > 2$ or 2) the interaction potential $V(x)$ has a flatness. In those cases the quenched disorder is unnecessary to enable the glassy phases. Otherwise the quenched disorder is needed to enable glass transitions suppressing the crystalline phases. We developed a replica formalism to solve the problems exactly in the $M \to \infty$ limit.

We applied the scheme to explicitly analyze the continuous spin models with the two types of non-linear potentials: the quadratic and soft/hardcore potential. In both cases we found continuous RSB so that the free-energy landscape is marginal as indicated by vanishing of the replicon eigenvalue $\lambda_R = 0$. Interestingly this happens even with the $p = 2$ continuous spin model in contrast to the case of the linear potential. However there is an important difference between the two models that the criticality approaching jamming exists in the hardcore model but not in the quadratic model. This is evident in Fig. 4 where one can see that the $q(x)$ function of the quadratic model does not develop any power law behaviors approaching jammming $q(x_1) \to 1$ ($T \to 0$) in sharp contrast to that of the hardcore model shown in Fig. 9. Critical jamming implies *mechanical* marginality which is reflected, for instance, as avalanche like responses to perturbations [69–73]. Possible relation between the landscape marginality and mechanical marginality is an interesting open question [36, 69, 70, 72]. In the hardcore model, using the continuous RSB solution we found that the isostaticity holds at jamming and the universality of it turned out to be the same as hardspheres for all values of $p$ establishing the superniversality. This observation extend the result on the perceptron [35, 37] which corresponds to $p = 1$.

Although we limited ourselves to the case $M \to \infty$ in the present paper, systematic $1/M$ expansions are possible. Such an approach has been conducted in the case of $p = 2$ continuous spin model with the linear potential, where RSB does not take place [74, 75]. This would be an alternative, analytically tractable approach to analyze systems on tree-like lattices of finite connectivity (Bethe lattice) where mean-field approach should remain valid. So far such systems remained hard to be analyzed by the the replica approach [76] so that the cavity approach is usually preferred which however is limited to 1RSB at the moment [77]. An advantage of the $1/M$ expansion approach is that one can analyze the system almost as easily as the globally coupled systems so that one can construct continuous RSB explicitly as we have done in the $M \to \infty$ limit, which may become necessary deep in the glassy phases [54]. It will be interesting to study, for example, how the nature of jamming change if $M$ becomes finite. To what extent such mean-field results would remain useful for finite dimensional systems where the lattices are no more like trees, remains of course as an outstanding open problem.

We expect our results provide a useful basis to formulate theoretical approaches to study glass transitions of rotational degree of freedoms. For example it is natural to study glass transitions of particulate systems with rotational degrees of freedom such as patchy colloids and ellipsoids (Fig. 1 c)) extending the present work. Another interesting problem is to study the apparently disorder-free spinglass transitions realized in some frustrated magnets [29, 30] (Fig. 1 b)). Continuous constrained satisfaction problems such as the continuous coloring problem (Fig. 1 a)) and related statistical inference problems can also be studied in similar frameworks.

# Acknowledgements

The author thanks Silvio Franz, Atsushi Ikeda, Florent Krzakala, Yuliang Jin, Kota Mitsumoto, Pierfrancesco Urbani, Shuta Yokoi and Francesco Zamponi for useful discussions.

**Funding information** This work was supported by KAKENHI (No. 25103005 "Fluctuation & Structure" and No. 50335337) from MEXT, Japan.

# A Density functional approach

In this appendix we discuss an alternative derivation of the free-energy functional given by Eq. (85) using a density functional approach closely following the study on the hardspheres [14–16].

## A.1 Spin liquid

Let us introduce 'spin' density defined as

$$N\rho(\mathbf{S}) = \sum_{i=1}^{N} \delta(\mathbf{S} - \mathbf{S}_i), \tag{320}$$

where $\delta(\mathbf{S})$ is the delta function in the spin-space which satisfies $\int_S d\mathbf{S}\delta(\mathbf{S}) = 1$. Let us also introduce the Mayer function,

$$f(\mathbf{S}_1, \mathbf{S}_2, \ldots, \mathbf{S}_p) = e^{-\beta V(\mathbf{S}_1, \mathbf{S}_2, \ldots, \mathbf{S}_p)} - 1 = -1 + \exp\left[-\beta V\left(\delta - \frac{1}{\sqrt{M}}\sum_{\mu=1}^{M}\prod_{l=1}^{p}(S_l)^{\mu}\right)\right]. \tag{321}$$

A convenient strategy is to write the free-energy as

$$e^{-\beta F} = \int \mathcal{D}[\rho(\mathbf{S})]e^{-\beta\mathcal{F}[\rho(\mathbf{S})]}, \tag{322}$$

where we introduced a functional $\mathcal{F}[\rho]$,

$$e^{-\beta\mathcal{F}[\rho(\mathbf{S})]} \equiv \int_S \prod_{i=1}^{N} d\mathbf{S}_i \prod_{<i_1, i_2, \ldots, i_p>} e^{-\beta V(S_{i_1}, S_{i_2}, \ldots, S_{i_p})} \delta\left[\rho(\mathbf{S}) - N^{-1}\sum_{i=1}^{N}\delta(\mathbf{S} - \mathbf{S}_i)\right] \tag{323}$$

with $\int \mathcal{D}[\rho(\mathbf{S})]$ being a functional integration over $\rho(\mathbf{S}) > 0$ and $\delta[\ldots]$ is a functional delta function.

To obtain the functional $\mathcal{F}[\rho]$ one can follow the standard step of the liquid theory [45]: one defines first a free-energy $F[\phi(\mathbf{S})]$ of the system with modified Hamiltonian $H = \sum_{<i_1, i_2, \ldots, i_p>} V(\mathbf{S}_{i_1}, \mathbf{S}_{i_1}, \ldots, \mathbf{S}_{i_p}) + \int_S d\mathbf{S}\rho(\mathbf{S})\phi(\mathbf{S})$ then perform a Legendre transformation to obtain a free-energy as functional of the spin density $\mathcal{F}[\rho(\mathbf{S})] = F[\phi(\mathbf{S})] - \int_S d\mathbf{S}\rho(\mathbf{S})\phi(\mathbf{S})$. As the result one finds,

$$-\beta\frac{\mathcal{F}[\rho(\mathbf{S})]}{N} = -\int_S d\mathbf{S}\rho(\mathbf{S})\ln\rho(\mathbf{S}) + \frac{c}{p}\int_S d\mathbf{S}_1 d\mathbf{S}_2\cdots d\mathbf{S}_p\rho(\mathbf{S}_1)\rho(\mathbf{S}_2)\cdots\rho(\mathbf{S}_p)f(\mathbf{S}_1, \mathbf{S}_2, \ldots, \mathbf{S}_p). \tag{324}$$

The free-energy $F$ is obtained by minimizing the variational free-energy functional $\mathcal{F}[\rho(\mathbf{S})]$. The 1st term on the r.h.s of Eq. (324) represents the entropic (paramagnetic) part of the free-energy. The 2nd term is the 1st virial correction due to interactions. The reason for the absence of the higher order terms, all of which are represented as 1 particle irreducible (1PI) diagrams such as a triangle, a square, e.t.c. [45], is the tree-like geometry of the lattices that we consider.

## A.2 Replicated spin liquid

In principle all stable and metastable states of the system, including liquid (paramagnetic) state $\rho_{\text{liq}}(\mathbf{S})$, crystalline state $\rho_{\text{crystal}}(\mathbf{S})$ and glassy states $\rho_\alpha(\mathbf{S})$ ($\alpha = 1, 2, \ldots,$), would be found as local minima of the free-energy functional given by Eq. (324). In the present paper we focus on the properties of glassy states which emerge from supercooled paramagnetic state.

A useful way to analyze the properties of glassy states is the replica approach. We consider replicated spin liquid of $n$ replicas labeled as $a = 1, 2, \ldots, n$ obeying the Hamiltonian,

$$H_n = \sum_{a=1}^{n} \sum_{<i_1, i_2, \ldots, i_p>} V(\mathbf{S}_{i_1}^a, \mathbf{S}_{i_1}^a, \ldots, \mathbf{S}_{i_p}^a) \tag{325}$$

For convenience we introduce a short hand notation

$$\overline{\mathbf{S}} = (\mathbf{S}^1, \mathbf{S}^2, \ldots, \mathbf{S}^n) \tag{326}$$

where $\mathbf{S}$s themselves are $M$ component spin vectors. Introducing replicated spin density

$$N\rho(\overline{\mathbf{S}}) = \sum_{i=1}^{N} \prod_{a=1}^{n} \delta(\mathbf{S}^a - \mathbf{S}_i^a) \tag{327}$$

which is normalized such that $\int d\overline{\mathbf{S}}\rho(\overline{\mathbf{S}}) = 1$, we find,

$$-\beta F = \partial_n Z_n \big|_{n=0} \qquad Z_n = \int \mathcal{D}[\rho(\overline{\mathbf{S}})] e^{-\beta \mathcal{F}_n[\rho(\overline{\mathbf{S}})]} \tag{328}$$

with the variational replicated free-energy functional defined as,

$$-\beta \frac{\mathcal{F}_n[\rho(\overline{\mathbf{S}})]}{N} = -\int_S d\overline{\mathbf{S}}\rho(\overline{\mathbf{S}})\ln\rho(\overline{\mathbf{S}}) + \frac{c}{p}\int_S d\overline{\mathbf{S}}_1 d\overline{\mathbf{S}}_2 \cdots d\overline{\mathbf{S}}_p \rho(\overline{\mathbf{S}}_1)\rho(\overline{\mathbf{S}}_2)\cdots\rho(\overline{\mathbf{S}}_p)f_n(\overline{\mathbf{S}}_1, \overline{\mathbf{S}}_2, \ldots, \overline{\mathbf{S}}_p). \tag{329}$$

where $d\overline{\mathbf{S}} = \prod_{a=1}^{n} d\mathbf{S}^a$ and we introduced a replicated Mayer function,

$$f_n(\overline{\mathbf{S}}_1, \overline{\mathbf{S}}_2, \ldots, \overline{\mathbf{S}}_p) = -1 + e^{-\beta \sum_{a=1}^n V(\mathbf{S}_1^a, \mathbf{S}_2^a, \ldots, \mathbf{S}_p^a)} = -1 + \prod_{a=1}^{n} \exp\left[-\beta V\left(\delta - \frac{1}{\sqrt{M}}\sum_{\mu=1}^{M}\prod_{l=1}^{p}(S^a)_l^\mu\right)\right]. \tag{330}$$

## A.3 Glass order parameter functional

We look for glassy metastable states which keep the statistical rotational invariance of the liquid (paramagnetic) state. To this end it is natural to consider the overlap matrix given by Eq. (69) as the order parameter,

$$\hat{Q}_{ab} = Q_{ab} \qquad Q_{ab} = \lim_{N\to\infty} \frac{1}{N}\sum_{i=1}^{N}\frac{1}{M}\sum_{\mu=1}^{M}(S^a)_i^\mu(S^b)_i^\mu \tag{331}$$

for $a, b = 1, 2, \ldots, n$. Note that $Q_{aa} = 1$ due to the normalization of the spins $|\mathbf{S}_i^a|^2 = M$.

### A.3.1 Variational free-energy

Based on the above discussion we expect that $\rho(\overline{\mathbf{S}})$ of the the glassy states, which keeps the statistical rotational invariance of the liquid, is parametrized solely by the overlap matrix $\hat{Q}$,

$$\rho(\overline{\mathbf{S}}) = \rho(\hat{Q}). \tag{332}$$

Since the system is regular and every vertex is exactly equivalent to each other in our system, it is natural to expect the order parameter does not fluctuate in space.

Similarly we anticipate that the replicated Mayer function can be parametrized as,

$$f_n(\overline{\mathbf{S}}_1, \overline{\mathbf{S}}_2, \ldots, \overline{\mathbf{S}}_p) = f_n(\hat{Q}_1, \hat{Q}_2, \ldots, \hat{Q}_p). \tag{333}$$

so that the variational free-energy functional given by Eq. (329) as a whole can be cast into the following rotationally invariant form,

$$
\begin{aligned}
-\beta \frac{\mathcal{F}_n[\rho(\hat{Q})]}{N} &= -\int d\hat{Q} J(\hat{Q})\rho(\hat{Q})\ln\rho(\hat{Q}) + \frac{c}{p}\int\prod_{l=1}^{p}\{d\hat{Q}_l J(\hat{Q}_l)\rho(\hat{Q}_l)\}f_n(\hat{Q}_1, \hat{Q}_2, \ldots, \hat{Q}_p) \\
&\quad + \lambda\left(\int d\hat{Q} J(\hat{Q})\rho(\hat{Q}) - 1\right)
\end{aligned}
\tag{334}
$$

where $d\hat{Q} = \prod_{a<b} dQ_{ab}$. Here $J(\hat{Q})$ is the Jacobian (see below) and the parameter $\lambda$ in the last term of Eq. (334) is a Lagrange multiplier to enforce the normalization of the spin density. Note that in the 2nd integral on the r. h. s. of Eq. (334) we assumed a simply factorized Jacobian $\prod_{l=1}^{p} J(\hat{Q})$ disregarding possible cross-correlations of spins at different sites $l = 1, 2, \ldots, p$. We comment on the validity of this assumption later.

The Jacobian $J$ is defined as

$$J(\hat{Q}) \equiv \int d\overline{\mathbf{S}}\prod_{a\leq b}\delta\left(Q_{ab} - \frac{1}{M}\sum_{\mu=1}^{M}(S^a)^\mu (S^b)^\mu\right) \tag{335}$$

Here we have replaced the constrained integral $\int_S d\overline{\mathbf{S}}$ by an unconstrained integral $\int d\overline{\mathbf{S}} \equiv \prod_{\mu=1}^{M}\prod_{a=1}^{n}\int_{-\infty}^{\infty} d(S^a)^\mu$. This is made possible by setting

$$Q_{aa} = 1 \tag{336}$$

for all $a$ in Eq. (335) so that the normalization condition of the spins $|\mathbf{S}^2| = M$ is enforced. Then one can evaluate the Jacobian to find (see Eq.(17) and (78) of [14]),

$$J(\hat{Q}) = C_{n+1,M} e^{\frac{1}{2}(M-(n+1))\ln\det\hat{Q}} \tag{337}$$

with $C_{n,M}$ being a numerical prefactor which behaves for $M \gg 1$ as,

$$\ln C_{n,M} = \frac{M}{2}(n-1)\ln(2\pi e) - \frac{M}{2}(n-1)\ln M \qquad M \gg 1. \tag{338}$$

Minimization of the variational free-energy given by Eq. (334) with respect to $\rho(\hat{Q})$,

$$0 = \frac{\delta}{\delta\rho(\hat{Q})}\beta\frac{\mathcal{F}_n[\rho(\hat{Q})]}{N} \tag{339}$$

yields,

$$\ln\rho(\hat{Q}) = \lambda - 1 + c\int\prod_{l=1}^{p-1}\{d\hat{Q}_l J(\hat{Q}_l)\rho(\hat{Q}_l)\}f_n(\hat{Q}, \hat{Q}_1, \ldots, \hat{Q}_{l-1}). \tag{340}$$

In addition, normalization of the spin density implies,

$$1 = \int d\hat{Q} J(\hat{Q})\rho(\hat{Q}) = \int d\hat{Q}\exp\left(\ln J(\hat{Q}) + \ln\rho(\hat{Q})\right) \tag{341}$$

with

$$\ln J(\hat{Q}) + \ln\rho(\hat{Q}) = \ln C_{n,M} + \frac{1}{2}(M-n)\ln\det\hat{Q} + \lambda - 1 + c\int\prod_{l=1}^{p-1}\{d\hat{Q}_l J(\hat{Q}_l)\rho(\hat{Q}_l)\}f_n(\hat{Q}, \hat{Q}_1, \ldots, \hat{Q}_{l-1}) \tag{342}$$

where we used Eq. (337) and Eq. (340).

### A.3.2 $M \to \infty$ limit

In the present paper we limit ourselves with the $M \to \infty$ limit which greatly simplifies the analysis. The first advantage of the $M \to \infty$ limit is that the integrals over $\hat{Q}$ can be done by saddle point method. For example in Eq. (341) the saddle point value $\hat{Q}^*$ is determined by,

$$0 = \frac{\delta}{\delta \hat{Q}} \left( \ln J(\hat{Q}) + \ln \rho(\hat{Q}) \right) \Bigg|_{\hat{Q}=\hat{Q}^*} \tag{343}$$

Importantly the integrals over $\hat{Q}$ in the variational free-energy functional given by Eq. (334) can also be evaluated by the saddle point method in the $M \to \infty$ limit and the saddle point should be exactly the same as the one given by Eq. (343). This is because in the free-energy functional given by Eq. (334), only the factor $J(\hat{Q})\rho(\hat{Q})$ is exponentially large in $M$. [14]

Here let us comment on the validity of our assumption used in Eq. (334) that fluctuations of spins at different sites $l = 1, 2, \ldots, p$ are uncorrelated which allowed us to assume a simply factorized Jacobian $\prod_{l=1}^{p} J(\hat{Q})$ in the 2nd integral of Eq. (334). Actually more generally we should write the Jacobian as,

$$
\begin{aligned}
K(\{\hat{Q}_l\}, \{\hat{P}_{ll'}\}) \;\equiv\; & \int \prod_{l=1}^{p} d\overline{\mathbf{s}}_l \prod_{a \leq b} \prod_l \delta \left( (Q_l)_{ab} - \frac{1}{M} \sum_{\mu=1}^{M} ((S_l)^a)^\mu ((S_l)^b)^\mu \right) \\
& \times \prod_{l<l'} \delta \left( (P_{ll'})_{ab} - \frac{1}{M} \sum_{\mu=1}^{M} ((S_l)^a)^\mu ((S_{l'})^b)^\mu \right)
\end{aligned} \tag{344}
$$

where $P_{l,l'}$ represents cross-correlation of the fluctuation of spins are different sites $l$ and $l'$. Then similarly to Eq. (341) we may consider the normalization of the spin density,

$$1 = \int \prod_l d\hat{Q}_l \prod_{l<l'} d\hat{P}_{ll'} \prod_{l=1}^{p} \rho(\{\hat{Q}_l\}) K(\{\hat{Q}_l\}, \{\hat{P}_{ll'}\}). \tag{345}$$

which implies, similarly to Eq. (343),

$$
\begin{aligned}
0 &= \frac{\delta}{\delta \hat{Q}_l} \left( \ln K(\{\hat{Q}_l\}, \{\hat{P}_{ll'}\}) + \sum_{l=1}^{p} \ln \rho(\hat{Q}_l) \right) \Bigg|_{\hat{Q}=\hat{Q}^*} \qquad l = 1, 2, \ldots, p \\
0 &= \frac{\delta}{\delta \hat{P}_{l,l'}} \left( \ln K(\{\hat{Q}_l\}, \{\hat{P}_{ll'}\}) + \sum_{l=1}^{p} \ln \rho(\hat{Q}_l) \right) \Bigg|_{\hat{Q}=\hat{Q}^*} \qquad l < l'
\end{aligned} \tag{346}
$$

The explicit form of $K(\{\hat{Q}_l\}, \{\hat{P}_{ll'}\})$ can be worked out similarly to Eq. (337) (see Eq.(40) and (78) of [14]) and one finds the 2nd equation of Eq. (346) yields $\hat{P}_{l,l'} = 0$ meaning that the cross-correlation between spin fluctuations at different sites vanish (see Eq.(62) and (63) of [14]). Thus we can use the factorized form for the Jacobian.

Now inspecting Eq. (342), it is evident that a sensible choice for the scaling of the connectivity to obtain nontrivial result in the $M \to \infty$ limit is

$$c = \alpha M \tag{347}$$

parametrized by $\alpha > 0$. Then using Eq. (342) and Eq. (347) the saddle point equation given by Eq. (343) becomes

$$0 = \frac{\delta}{\delta \hat{Q}} \left( \frac{1}{2} \ln \det \hat{Q} + \alpha f_n(\hat{Q}, \hat{Q}^*, \ldots, \hat{Q}^*) \right) \Bigg|_{\hat{Q}=\hat{Q}^*} = \frac{\delta}{\delta \hat{Q}} \left( \frac{1}{2} \ln \det \hat{Q} + \frac{\alpha}{p} f_n(\hat{Q}, \hat{Q}, \ldots, \hat{Q}) \right) \Bigg|_{\hat{Q}=\hat{Q}^*}. \tag{348}$$

The value of the Lagrange multiplier $\lambda$ is fixed by Eq. (341), which requires vanishing of $O(M)$ terms in $\ln \rho(\hat{Q}^*) + \ln J(\hat{Q}^*)$,

$$0 = \lambda - 1 + M\left[\frac{1}{2}n\ln(2\pi e) - \frac{1}{2}n\ln M + \frac{1}{2}\ln \det\hat{Q}^* + \alpha f_n(\hat{Q}^*, \hat{Q}^*, \ldots, \hat{Q}^*)\right] \tag{349}$$

where we used Eq. (338), Eq. (342) and dropped sub-leading terms. Using this result together with Eq. (340), we find the saddle point value of the variational free-energy given by Eq. (334) in the $M \to \infty$ limit as,

$$-\beta \frac{\mathcal{F}_n[\rho(\hat{Q}^*)]}{N} = -\ln \rho(\hat{Q}^*) + M\frac{\alpha}{p}f_n(\hat{Q}^*, \hat{Q}^*, \ldots, \hat{Q}^*) = 1 - \lambda - M\frac{\alpha}{p}(p-1)f_n(\hat{Q}^*, \hat{Q}^*, \ldots, \hat{Q}^*)$$

$$= M\left[\frac{1}{2}n\ln\left(\frac{2\pi e}{M}\right) + \frac{1}{2}\ln \det\hat{Q}^* + \frac{\alpha}{p}f_n(\hat{Q}^*, \hat{Q}^*, \ldots, \hat{Q}^*)\right]. \tag{350}$$

Note that if we regard $\hat{Q}^*$ in the above expression as a variational parameter, we find a variational equation which is exactly the same as Eq. (348).

Next let us examine the interaction part of the free-energy to extract the explicit form of the replicated Mayer function. The interaction part of the free-energy of the replicated system reads as (see Eq. (329), Eq. (335)),

$$\frac{c}{p}\int_S d\bar{\mathbf{S}}_1 d\bar{\mathbf{S}}_2 \cdots d\bar{\mathbf{S}}_p \rho(\bar{\mathbf{S}}_1)\rho(\bar{\mathbf{S}}_2)\cdots\rho(\bar{\mathbf{S}}_p)f_n(\bar{\mathbf{S}}_1, \bar{\mathbf{S}}_2, \ldots, \bar{\mathbf{S}}_p)$$

$$= \frac{c}{p}\left[-1 + \int\prod_{l=1}^{p}\{d\hat{Q}_l J(\hat{Q}_l)\rho(\hat{Q}_l)\}\left\langle\prod_{a=1}^{n}\exp\left(-\beta V\left(\delta - \frac{1}{\sqrt{M}}\sum_{\mu=1}^{M}\prod_{l=1}^{p}(S^a)_l^\mu\right)\right)\right\rangle_{\hat{Q}}\right]$$

$$= \frac{c}{p}\left[-1 + \int\prod_{l=1}^{p}\{d\hat{Q}_l J(\hat{Q}_l)\rho(\hat{Q}_l)\}\int\prod_{a=1}^{n}\left\{\frac{d\kappa_a}{2\pi}e^{i\kappa_a\delta}\mathcal{Z}_{\kappa_a}\right\}\left\langle\exp\left(\sum_{a=1}^{n}\frac{-i\kappa_a}{\sqrt{M}}\sum_{\mu=1}^{M}\prod_{l=1}^{p}(S^a)_l^\mu\right)\right\rangle_{\hat{Q}}\right] \tag{351}$$

where we introduced a Fourier transform,

$$Z_\kappa \equiv \int dx\, e^{-i\kappa x}e^{-\beta V(x)} \tag{352}$$

and a short hand notation,

$$\langle\cdots\rangle_{\hat{Q}} \equiv \int\prod_{l=1}^{p}\left\{d\bar{\mathbf{S}}_l\frac{1}{J(\hat{Q}_l)}\prod_{a\leq b}\delta\left((Q_l)_{ab} - \frac{1}{M}\sum_{\mu=1}^{M}(S_l^a)^\mu(S_l^b)^\mu\right)\right\}\cdots \tag{353}$$

In the last equation $Q_{aa} = 1$ (Eq. (336)) to enforce the normalization of the spins $|(\mathbf{S}^a)^2| = M$ and $\int d\bar{\mathbf{S}}$ is an unconstrained integral.

For $M \gg 1$ we can evaluate the last factor in Eq. (351) by performing $1/\sqrt{M}$ expansion,

$$\ln\left\langle\exp\left(\sum_{a=1}^{n}\frac{-i\kappa_a}{\sqrt{M}}\sum_{\mu=1}^{M}\prod_{l=1}^{p}(S^a)_l^\mu\right)\right\rangle_{\hat{Q}}$$

$$= \ln\left[1 + \sum_{a=1}^{n}\frac{-i\kappa_a}{\sqrt{M}}\sum_{\mu=1}^{M}\prod_{l=1}^{p}\langle(S_l^a)^\mu\rangle_{\hat{Q}} + \frac{1}{2}\sum_{a=1}^{n}\sum_{b=1}^{n}(-i\kappa_a)(-i\kappa_b)\frac{1}{M}\prod_{l=1}^{p}\sum_{\mu,\nu=1}^{M}\langle(S_l^a)^\mu(S_l^a)^\nu\rangle_{\hat{Q}} + \cdots\right]$$

$$\xrightarrow[M\to\infty]{}\frac{1}{2}\sum_{a,b=1}^{n}(-i\kappa_a)(-i\kappa_b)\prod_{l=1}^{p}(Q_l)_{ab} = \frac{1}{2}\sum_{a,b=1}^{n}(-i\kappa_a)(-i\kappa_b)\prod_{l=1}^{p}(Q_l)_{ab} \tag{354}$$

Here we used the fact that in the $M \to \infty$ limit, different components of the spins $S^\mu$ become independent from each other. This can be checked by introducing integral representation of the $\delta$ function in Eq. (353) which can be evaluated by the saddle point method in the $M \to \infty$ limit.

To sum up we find the replicated Mayer function in the $M \to \infty$ limit as,

$$
\begin{aligned}
&f_n(\hat{Q}_1, \hat{Q}_2, \ldots, \hat{Q}_p) \\
=\ & -1 + \int \prod_{a=1}^{n} \left\{ \frac{d\kappa_a}{2\pi} e^{i\kappa_a \delta} \right\} \exp\left( \frac{1}{2} \sum_{a,b=1}^{n} \prod_{l=1}^{p} (Q_l)_{ab} (-i\kappa_a)(-i\kappa_b) \right) \prod_{a=1}^{n} \int dh_a e^{-i\kappa_a h_a} e^{-\beta V(h_a)} \\
=\ & -1 + \int \prod_{a=1}^{n} dh_a \left\{ \exp\left( \frac{1}{2} \sum_{a,b=1}^{n} \prod_{l=1}^{p} (Q_l)_{ab} \frac{\partial^2}{\partial h_a \partial h_b} \right) \prod_{a=1}^{n} \delta(\delta - h_a) \right\} \prod_a e^{-\beta V(h_a)} \\
=\ & -1 + \exp\left( \frac{1}{2} \sum_{a,b=1}^{n} \prod_{l=1}^{p} (Q_l)_{ab} \frac{\partial^2}{\partial h_a \partial h_b} \right) \prod_{a=1}^{n} e^{-\beta V(\delta + h_a)} \Bigg|_{\{h_a = 0\}}
\end{aligned}
\tag{355}
$$

The last equation is obtained by repeating integrations by parts.

Collecting the above results we find the thermodynamic free-energy given by Eq. (328)as,

$$
-\beta \frac{F}{NM} = -\beta f[\hat{Q}^*]
\tag{356}
$$

with the variational free-energy (more precisely free-entropy),

$$
\begin{aligned}
-\beta f[\hat{Q}] &= \partial_n s_n[\hat{Q}] \Big|_{n=0} \\
s_n[\hat{Q}] &\equiv \frac{1}{2} \ln \det \hat{Q} - \frac{\alpha}{p} \mathcal{F}_{\text{int}}[\hat{Q}] \\
-\mathcal{F}_{\text{int}}[\hat{Q}] &\equiv \exp\left( \frac{1}{2} \sum_{a,b=1}^{n} (Q_{ab})^p \frac{\partial^2}{\partial h_a \partial h_b} \right) \prod_{a=1}^{n} e^{-\beta V(\delta + h_a)} \Bigg|_{\{h_a = 0\}}
\end{aligned}
\tag{357}
$$

where we dropped off irrelevant constants. The saddle point $\hat{Q}^*$ is determined by,

$$
\frac{\delta s[\hat{Q}]}{\delta Q_{ab}} \Bigg|_{\hat{Q} = \hat{Q}^*} = 0
\tag{358}
$$

for all $a \neq b$.

### A.3.3 Gaussian ansatz

Finally let us note that one can check that the above result can be reproduced by assuming an Gaussian ansatz for the replicated spin density,

$$
\rho_{\text{Gaussian}}(\mathbf{S}) = \frac{e^{-\frac{1}{2} \sum_{a,b=1}^{n} (Q^{-1})_{ab} \sum_{\mu=1}^{M} (S^a)^\mu (S^b)^\mu}}{\sqrt{2\pi (\det \hat{Q})^M}}.
\tag{359}
$$

The situation is essentially the same as that of hardspheres in the large dimensional limit [14–16].

# B Eigenvalues of the stability matrix

Here we analyze the Hessian matrix $M_{a\neq b,c\neq d}$ of the free-energy around the saddle points. It is a matrix of size $n(n-1) \times n(n-1)$ defined as,

$$M_{a\neq b,c\neq d} \equiv -\frac{\partial^2 s[\hat{Q}]}{\partial Q_{a<b}\partial Q_{c<d}} \tag{360}$$

where $s_n[\hat{Q}]$ is the free-entropy defined in Eq. (357) which reads,

$$s_n[\hat{Q}] \equiv \frac{1}{2}\ln\det\hat{Q} - \frac{\alpha}{p}\mathcal{F}_{\text{int}}[\hat{Q}] \tag{361}$$

$$-\mathcal{F}_{\text{int}}[\hat{Q}] \equiv \exp\left(\frac{1}{2}\sum_{a,b=1}^{n}(Q_{ab})^p \frac{\partial^2}{\partial h_a \partial h_b}\right)\prod_{a=1}^{n} e^{-\beta V(\delta+h_a)}\Bigg|_{\{h_a=0\}}. \tag{362}$$

The Hessian matrix can be naturally written as sum of the contribution from the entropic part and interaction part of the free-energy,

$$M_{a\neq b,c\neq d} = M^{\text{ent.}}_{a\neq b,c\neq d} + M^{\text{int.}}_{a\neq b,c\neq d}, \tag{363}$$

with

$$M^{\text{ent.}}_{a\neq b,c\neq d} = -\frac{\partial^2}{\partial Q_{a<b}\partial Q_{c<d}}\frac{1}{2}\ln\det\hat{Q} = Q^{-1}_{ac}Q^{-1}_{bd} + Q^{-1}_{ad}Q^{-1}_{bc} \tag{364}$$

$$\begin{aligned}
M^{\text{int.}}_{a\neq b,c\neq d} &= \frac{\alpha}{p}\frac{\partial^2}{\partial Q_{a<b}\partial Q_{c<d}}\mathcal{F}_{\text{int}}[\hat{Q}] \\
&= \frac{\alpha}{p}\Bigg[ p(p-1)Q^{p-2}_{a<b}(\delta_{ac}\delta_{bd} + \delta_{ad}\delta_{bc})\frac{\partial^2}{\partial h_a \partial h_b} \\
&\quad + p^2 Q^{p-1}_{a<b}Q^{p-1}_{c<d}\frac{\partial^4}{\partial h_a \partial h_b \partial h_c \partial h_d}\Bigg]\mathcal{F}_{\text{int}}[\hat{Q},\{h_a\}]\Bigg|_{\{h_a=0\}}, 
\end{aligned} \tag{365}$$

with

$$-\mathcal{F}_{\text{int}}[\hat{Q},\{h_a\}] \equiv \exp\left(\frac{1}{2}\sum_{e,f=1}^{n} Q^p_{ef}\frac{\partial^2}{\partial h_e \partial h_f}\right)\prod_{a=1}^{n} e^{-\beta V(\delta+h_a)}. \tag{366}$$

## B.1 RS ansatz

Here we analyze the eigenvalues of the Hessian matrix for the case of the replica symmetric (RS) solution characterized by the order parameter matrix of the form given by Eq. (126), which reads as,

$$\hat{Q}^{\text{RS}} = (1-q)\delta_{ab} + q. \tag{367}$$

The replica symmetry implies the following matrix structure,

$$M_{a\neq b,c\neq d} = M_1 \frac{\delta_{ac}\delta_{bd} + \delta_{ad}\delta_{bc}}{2} + M_2 \frac{\delta_{ac} + \delta_{ad} + \delta_{bc} + \delta_{bd}}{4} + M_3 \tag{368}$$

from which the eigenvalues of the Hessian matrix are obtained as [15,56],

$$\lambda_R = M_1, \tag{369}$$

$$\lambda_L = n(n-1)M_3 + (n-1)M_2 + M_1 \xrightarrow[n\to 0]{} M_1 - M_2, \tag{370}$$

$$\lambda_A = \frac{1}{2}(n-2)M_2 + M_1 \xrightarrow[n\to 0]{} M_1 - M_2. \tag{371}$$

The factors $M_i$'s can be decomposed into the entropic and interaction parts, $M_i = M_i^{\text{ent}} + M_i^{\text{int}}$ like Eq. (363).

### B.1.1 Contribution form the entropic part

First let us examine the entropic part. The replica symmetric matrix given by Eq. (367) can be easily inverted to find,

$$(\hat{Q}^{\text{RS}})^{-1}_{ab} = \hat{q}\delta_{ab} + \tilde{q} \tag{372}$$

with

$$\hat{q} = \frac{1}{1-q}, \tag{373}$$

$$\tilde{q} = -\frac{q}{1+(n-2)q-(n-1)q^2} \xrightarrow{n\to 0} -\frac{q}{(1-q)^2}. \tag{374}$$

Using this in Eq. (364), we obtain the entropic contributions as,

$$M_1^{\text{ent.}} = \lim_{n\to 0} 2(\hat{q})^2 = \frac{2}{(1-q)^2}, \tag{375}$$

$$M_2^{\text{ent.}} = \lim_{n\to 0} 4\hat{q}\tilde{q} = -4\frac{q}{(1-q)^3}, \tag{376}$$

$$M_3^{\text{ent.}} = \lim_{n\to 0} 2(\tilde{q})^2 = 2\frac{q^2}{(1-q)^4}. \tag{377}$$

### B.1.2 Contribution form the interaction part

Next let us examine the interaction part Eq. (365). Within the RS ansatz we find,

$$\lim_{n\to 0} M_{a\neq b, c\neq d}^{\text{int.}} = \lim_{n\to 0} \exp\left(\frac{q^p}{2}\sum_{e,f=1}^{n}\frac{\partial^2}{\partial h_e \partial h_f}\right)\frac{\alpha}{p}\left[p(p-1)q^{p-2}(\delta_{ac}\delta_{bd}+\delta_{ad}\delta_{bc})\frac{\partial^2}{\partial h_a \partial h_b}\right.$$

$$\left.+p^2 q^{2(p-1)}\frac{\partial^4}{\partial h_a \partial h_b \partial h_c \partial h_d}\right]\prod_{a=1}^{n}g(\delta+h_a)\Bigg|_{\{h_a=0\}} \tag{378}$$

where we used

$$-\mathcal{F}_{\text{int}}[\hat{Q}^{\text{RS}},\{h_a\}] = \exp\left(\frac{1}{2}\sum_{e,f=1}^{n}((1-q^p)\delta_{ab}+q^p)\frac{\partial^2}{\partial h_e \partial h_f}\right)\prod_{a=1}^{n}e^{-\beta V(\delta+h_a)}$$

$$= \exp\left(\frac{q^p}{2}\sum_{e,f=1}^{n}\frac{\partial^2}{\partial h_e \partial h_f}\right)\prod_{a=1}^{n}g(\delta+h_a) \tag{379}$$

In the last equation we introduced a shorthanded notation of the quantity defined in Eq. (156),

$$g(h) \equiv g(m_{k+1},h) = \gamma_{1-q_k^p}\otimes e^{-\beta V(h)}. \tag{380}$$

For a convenience let us also introduce a related shorthanded notation (See Eq. (160))

$$f(h) \equiv f(m_{k+1},h) = -\frac{1}{m_{k+1}}\log g(m_{k+1},h) = -\log g(h), \tag{381}$$

where $m_{k+1} = 1$. Note that $k = 0$ for the RS case.

By taking derivatives we find,

$$\lim_{n\to 0}\exp\left(\frac{q^p}{2}\sum_{e,f=1}^{n}\frac{\partial^2}{\partial h_e\partial h_f}\right)\frac{\partial^2}{\partial h_a\partial h_b}\prod_{a=1}^{n}g(\delta+h_a)\Bigg|_{h_a=0}$$

$$=\exp\left(\frac{q^p}{2}\frac{\partial^2}{\partial h^2}\right)\left(\frac{g'(h)}{g(h)}\right)^2\Bigg|_{h=\delta}=\gamma_{q^p}\otimes\left(\frac{g'(\delta)}{g(\delta)}\right)^2 \qquad (382)$$

in the last equation we used Eq. (130). Similarly we obtain,

$$\lim_{n\to 0}\exp\left(\frac{q^p}{2}\sum_{e,f=1}^{n}\frac{\partial^2}{\partial h_e\partial h_f}\right)\frac{\partial^4}{\partial h_a\partial h_b\partial h_c\partial h_d}\prod_{a=1}^{n}g(\delta+h_a)\Bigg|_{\{h_a=0\}}$$

$$=\gamma_{q^p}\otimes\Bigg\{(\delta_{ac}\delta_{bd}+\delta_{ad}\delta_{bc})\left(\frac{g''(\delta)}{g(\delta)}\right)^2$$

$$+[\delta_{ac}+\delta_{bc}+\delta_{ad}+\delta_{bd}-2(\delta_{ad}\delta_{bc}+\delta_{ac}\delta_{bd})]\left[\frac{g''(\delta)}{g(\delta)}\left(\frac{g'(\delta)}{g(\delta)}\right)^2\right]$$

$$+[1-(\delta_{ac}+\delta_{bc}+\delta_{ad}+\delta_{bd})+(\delta_{ad}\delta_{bc}+\delta_{ac}\delta_{bd})]\left(\frac{g'(\delta)}{g(\delta)}\right)^4\Bigg\}. \qquad (383)$$

From the above result we find the contributions by the interaction part as,

$$-M_1^{\text{int.}}=\frac{2\alpha}{p}\Bigg[p(p-1)q^{p-2}\gamma_{q^p}\otimes\left(\frac{g'(\delta)}{g(\delta)}\right)^2$$

$$+(pq^{p-1})^2\gamma_{q^p}\otimes\left\{\left(\frac{g''(\delta)}{g(\delta)}\right)^2-2\frac{g''(\delta)}{g(\delta)}\left(\frac{g'(\delta)}{g(\delta)}\right)^2+\left(\frac{g'(\delta)}{g(\delta)}\right)^4\right\}\Bigg]$$

$$=\frac{2\alpha}{p}\left[p(p-1)q^{p-2}\gamma_{q^p}\otimes(f'(\delta))^2+(pq^{p-1})^2\gamma_{q^p}\otimes(f''(\delta))^2\right],$$

$$-M_2^{\text{int.}}=\frac{4\alpha}{p}(pq^{p-1})^2\gamma_{q^p}\otimes\left\{\frac{g''(\delta)}{g(\delta)}\left(\frac{g'(\delta)}{g(\delta)}\right)^2-\left(\frac{g'(\delta)}{g(\delta)}\right)^4\right\}$$

$$=\frac{4\alpha}{p}(pq^{p-1})^2\gamma_{q^p}\otimes(-f''(\delta)(f'(\delta))^2),$$

$$-M_3^{\text{int.}}=\frac{\alpha}{p}(pq^{p-1}))^2\gamma_{q^p}\otimes\left(\frac{g'(\delta)}{g(\delta)}\right)^4=\frac{\alpha}{p}(pq^{p-1}))^2\gamma_{q^p}\otimes(f'(\delta))^4. \qquad (384)$$

### B.1.3 Replicon eigenvalue

Summing up the above results we find the replicon eigenvalue which is responsible for the RSB instability of the RS ansatz as,

$$\lambda_{\text{R}}=\frac{2}{(1-q_0)^2}-2\frac{\alpha}{p}\int dh\,\frac{e^{-\frac{h^2}{2q^p}}}{\sqrt{2\pi q^p}}\left[p(p-1)q^{p-2}(f'(\delta-h))^2+(pq^{p-1})^2(f''(\delta-h))^2\right].$$

$$(385)$$

## B.2 $k$-RSB ansatz

Next let us analyze the case of $k$-step replica symmetry breaking solution with the ansatz given by Eq. (143). Within the $k$-RSB ansatz, $n$ replicas are divided into $n/m_1$ groups of size $m_1$ and each of the latter is divided into $m_1/m_2$ groups of size $m_2$, and so on. Finally we find $n/m_k$ groups of size $m_k$. Within each of the groups of size $m_k$, the replica symmetry remains. As we did in the 1-RSB case, here we only analyze stability of the replica symmetry within such a most inner-core group. Thus we just consider the Hessian matrix $M_{a \neq b, c \neq d}$ given by Eq. (360) assuming that all indexes $a,b,c,d$ are in the same most-inner core replica group of size $m_k$, which we denote as $\mathcal{C}$ in the following.

### B.2.1 Contributions from the interaction term

Let us first examine the contributions from the interaction term. Within the $k$-RSB ansatz, the interaction part of the Hessian matrix $M^{\text{int}}_{a \neq b, c \neq d}$ given by Eq. (363) for $a,b,c,d$ in the same most-inner core replica group $\mathcal{C}$ becomes, using Eq. (153) and Eq. (159),

$$
\begin{aligned}
- \lim_{n \to 0} M^{\text{int.},\mathcal{C}}_{a \neq b, c \neq d} &= \lim_{n \to 0} \prod_{l=0}^{k} \exp \left( \frac{\Lambda_l}{2} \sum_{e,f=1}^{n} I^{m_l}_{ef} \frac{\partial^2}{\partial h_e \partial h_f} \right) \left( \prod_{a \notin \mathcal{C}} g(\delta + h_a) \right) \\
&\quad \frac{\alpha}{p} \left[ p(p-1) q_k^{p-2} (\delta_{ac}\delta_{bd} + \delta_{ad}\delta_{bc}) \frac{\partial^2}{\partial h_a \partial h_b} + p^2 q_k^{2(p-1)} \frac{\partial^4}{\partial h_a \partial h_b \partial h_c \partial h_d} \right] \prod_{a \in \mathcal{C}} g(\delta + h_a) \bigg|_{\{h_a = 0\}} \\
&= \lim_{m_0 \to 0} \gamma_{\Lambda_0} \otimes \left\{ g^{m_0/m_1 - 1}(m_1, \delta) \gamma_{\Lambda_1} \otimes \left\{ g^{m_1/m_2 - 1}(m_2, \delta) \gamma_{\Lambda_2} \otimes \left\{ \cdots \right. \right. \right. \\
&\quad \left. \left. \left. \cdots g^{m_{k-1}/m_k - 1}(m_k, h) \gamma_{\Lambda_k} \otimes \left\{ g^{m_k}(m_{k+1}, h) \left[ S_1(h) \frac{\delta_{ac}\delta_{bd} + \delta_{ad}\delta_{bc}}{2} \right. \right. \right. \right. \right. \\
&\quad \left. \left. \left. \left. \left. + S_2(h) \frac{\delta_{ac} + \delta_{ad} + \delta_{bc} + \delta_{bd}}{4} + S_3(h) \right] \right\} \right\} \right|_{h=\delta} \\
&= \int dh P(m_k, h) \left[ S_1(h) \frac{\delta_{ac}\delta_{bd} + \delta_{ad}\delta_{bc}}{2} + S_2(h) \frac{\delta_{ac} + \delta_{ad} + \delta_{bc} + \delta_{bd}}{4} + S_3(h) \right].
\end{aligned}
\tag{386}
$$

In the last equation we used Eq. (407) derived in appendix D and Eq. (188). In the last equation we introduced,

$$
S_1(h) = \frac{2\alpha}{p} \left[ p(p-1) q^{p-2} (f'(h))^2 + (pq^{p-1})^2 (f''(h))^2 \right], \tag{387}
$$

$$
S_2(h) = \frac{4\alpha}{p} (pq^{p-1})^2 (-f''(h)(f'(h))^2), \tag{388}
$$

$$
S_3(h) = \frac{\alpha}{p} (pq^{p-1})^2 (f'(h))^4. \tag{389}
$$

Thus we find the contributions from the interaction term as,

$$
-M^{\text{int.}}_1 = \int dh P(m_k, h) S_1(h), \tag{390}
$$

$$
-M^{\text{int.}}_2 = \int dh P(m_k, h) S_2(h), \tag{391}
$$

$$
-M^{\text{int.}}_3 = \int dh P(m_k, h) S_3(h). \tag{392}
$$

The above formula reduces to the RS one given by Eq. (378) for $k = 0$ case as it should.

### B.2.2 Contributions from the entropic term

Next let us examine the entropic contribution. To this end it is useful to note first that the entropic contribution to the $k$-RSB free-energy can also be expressed in a recursive manner exploiting the hierarchical structure of the order parameter,

$$\frac{1}{2}\ln\det\hat{Q} = -\ln I(\hat{Q}) \tag{393}$$

$$I(\hat{Q}) \equiv \int \prod_{a=1}^{n} d\phi_a e^{-\frac{1}{2}\sum_{a,b=1}^{n}\phi_a Q_{ab}\phi_b + \sum_a h_a \phi_a}\Bigg|_{h_a=0}$$

$$= \prod_{l=0}^{k} e^{-\frac{\Lambda_l}{2}\sum_{ab}I_{ab}^{m_l}\frac{\partial^2}{\partial h_a \partial h_b}} \prod_{a=1}^{n} g_e(m_{k+1}, h_a), \tag{394}$$

where (see Eq. (143))

$$\Lambda_0 = q_0, \qquad \Lambda_i = q_i - q_{i-1}, \tag{395}$$

and

$$g_e(m_{k+1}, h) \equiv \int \frac{d\phi}{\sqrt{2\pi}} e^{-\frac{1}{2}(1-q_k)\phi^2 + h\phi} = \frac{e^{\frac{h^2}{2(1-q_k)}}}{\sqrt{1-q_k}}. \tag{396}$$

Comparing the above expressions with Eq. (153) we find the entropic term is expressed very similarly as the interaction term. We just need to put $p = 1$ in Eq. (155) (see also Eq. (154)) and replace $g(m_{k+1}, h)$ by $g_e(m_{k+1}, h)$ defined above. Again we can define a family of functions $g_e(m_l, h)$ for $l = 0, 1, \ldots, k$ through Eq. (157) with the boundary condition given by Eq. (396). Then we can write $\ln I(\hat{Q}) = g_e(m_0, 0)$. In addition we can introduce $f_e(m_i, h) \equiv -(1/m_i)\ln g_e(m_i, h)$ (Eq. (160)) and $P_{i,j}^e(y, h) \equiv \delta f_e(m_i, y)/\delta f_e(m_j, h)$ (Eq. (182)) and $P^e(m_j, h) \equiv P_{0,j+1}^e(0, h)$ (Eq. (188)).

We have to note however that sign in front of $\Lambda_l$ in Eq. (394) is *negative*. Thus we have to understand the operator $\otimes$ which appears in equations like Eq. (157) not in the Gaussian convolution form given by Eq. (132) but in the original differential form given by Eq. (130).

Using the above results we can write the entropic part of the sub-matrix of the Hessian matrix associated with a most inner core group $\mathcal{C}$ as,

$$\begin{aligned}
\lim_{n\to 0} M_{a\neq b, c\neq d}^{\text{ent.},\mathcal{C}} &= -\lim_{n\to 0}\frac{\partial^2}{\partial Q_{a\neq b}\partial Q_{c\neq d}}\frac{1}{2}\ln\det\hat{Q} \\
&= \lim_{n\to 0}Q_{a\neq b}^{-1}Q_{c\neq d}^{-1} + \lim_{n\to 0}\frac{\partial^2}{\partial Q_{a\neq b}\partial Q_{c\neq d}}I(\hat{Q}). \tag{397}
\end{aligned}$$

Note that the 1st term on the r.h.s of the last equation contributes only to $M_3^{\text{ent.}}$. For the replicon mode we need $M_1^{\text{ent.}}$ which is obtained as,

$$M_1^{\text{ent.}} = \frac{2}{(1-q_k)^2}. \tag{398}$$

This can be obtained using Eq. (390) and Eq. (387) with the following modifications: $p \to 1$, $f''(h) \to f_e''(m_{k+1}, h) = -1/(1-q_k)$ which can be obtained from Eq. (396), and $-\alpha/p \to 1$.

### B.2.3 Replicon eigenvalue

Summing up the above results we find the replicon eigenvalue which is responsible for the RSB instability of a most-inner core replica group in the $k$-RSB ansatz as,

$$\lambda_R = \frac{2}{(1-q_k)^2} - 2\frac{\alpha}{p}\int dh P(m_k, h)\left[p(p-1)q^{p-2}(f'(m_{k+1}, h))^2 + (pq^{p-1})^2(f''(m_{k+1}, h))^2\right]. \tag{399}$$

## C  Derivation of Eq. (203)

Here we show the derivation of Eq. (203). Let us begin with the case $1 \le i = j \le k$. Using the recursion formula given by Eq. (161) we find,

$$\partial_{\lambda_i} f(m_i, y) = e^{m_i f(m_i, y)} \int \mathcal{D} z_i e^{-m_i f(m_{i+1}, \Xi_i)} \partial_{\lambda_i} f(m_{i+1}, \Xi_i), \tag{400}$$

where $\Xi_i = y - \sqrt{\lambda_i - \lambda_{i-1}} z_i$ and with $\Xi_{i+1} = \Xi_i - \sqrt{\lambda_{i+1} - \lambda_i} z_{i+1}$ we find,

$$\partial_{\lambda_i} f(m_{i+1}, y) = e^{m_{i+1} f(m_{i+1}, y)} \int \mathcal{D} z_{i+1} e^{-m_{i+1} f(m_{i+2}, \Xi_{i+1})} \partial_{\lambda_i} f(m_{i+2}, \Xi_{i+1}). \tag{401}$$

Then by noting that $\Xi_{i+1} = y - \sqrt{\lambda_i - \lambda_{i-1}} z_i - \sqrt{\lambda_{i+1} - \lambda_i} z_{i+1}$ we find,

$$\partial_{\lambda_i} f(m_{i+2}, \Xi_{i+1}) = f'(m_{i+2}, \Xi_{i+1}) \left( \frac{1}{2} \frac{z_{i+1}}{\sqrt{\lambda_{i+1} - \lambda_i}} - \frac{1}{2} \frac{z_i}{\sqrt{\lambda_i - \lambda_{i-1}}} \right), \tag{402}$$

where the dash represents the partial derivative with respect to the 2nd argument $\partial_h f(x, h) = f'(x, h)$.

Collecting the above results, we find for $i = 0, 2, \ldots, k$,

$$\begin{aligned} \partial_{\lambda_i} f(m_i, y) &= \frac{1}{2}(m_{i+1} - m_i) e^{m_i f(m_i, y)} \int \mathcal{D} z_i e^{-m_i f(m_{i+1}, \Xi_i)} \left( f'(m_{i+1}, \Xi_i) \right)^2 \\ &= \frac{1}{2}(m_{i+1} - m_i) \int dh P_{i,i+1}(y, h) \left( f'(m_{i+1}, h) \right)^2. \end{aligned} \tag{403}$$

To derive the 1st equation we performed integrations by parts. In 2nd equation we used the identity given by Eq. (184). One can naturally generalize the above analysis and find for $1 \le i \le j \le k$,

$$\partial_{\lambda_j} f(m_i, y) = \int dh \frac{\delta f(m_i, y)}{\delta f(m_j, h)} \partial_{\lambda_j} f(m_j, h) = \frac{1}{2}(m_j - m_{j+1}) \int dh P_{ij+1}(y, h) \left( f'(m_{j+1}, h) \right)^2, \tag{404}$$

which is the desired result given by Eq. (203). In the last equation we used Eq. (182), Eq. (403) and the identity given by Eq. (183).

## D  Expansion of $P_{o,j}(h, y)$

Using the recursion formula given by Eq. (158) (see also the expansion displayed in Eq. (159)) we find,

$$\begin{aligned} \frac{\delta g(m_0, h)}{\delta g(m_j, y)} &= \frac{m_0}{m_1} \gamma_{\Lambda_0} \otimes \left\{ g^{m_0/m_1 - 1}(m_1, h) \frac{\delta g(m_1, h)}{\delta g(m_j, y)} \right\} \Bigg|_{h=\delta} \\ &= \frac{m_0}{m_1} \frac{m_1}{m_2} \gamma_{\Lambda_0} \otimes \left\{ g^{m_0/m_1 - 1}(m_1, h) \gamma_{\Lambda_1} \otimes \left\{ g^{m_1/m_2 - 1}(m_2, h) \frac{\delta g(m_2, h)}{\delta g(m_j, y)} \right\} \right\} \Bigg|_{h=\delta} \\ &= \cdots \\ &= \frac{m_0}{m_1} \frac{m_1}{m_2} \cdots \frac{m_{j-1}}{m_j} \gamma_{\Lambda_0} \otimes \left\{ g^{m_0/m_1 - 1}(m_1, h) \gamma_{\Lambda_1} \otimes \left\{ g^{m_1/m_2 - 1}(m_2, h) \gamma_{\Lambda_2} \otimes \cdots \right. \right. \\ &\qquad \left. \left. \cdots \gamma_{\Lambda_{j-1}} \otimes \left\{ g^{m_{j-1}/m_j - 1}(m_j, h) \delta(h - y) \right\} \right\} \right\}. \end{aligned} \tag{405}$$

On the other hand using Eq. (182) and Eq. (160) we find,

$$\frac{\delta g(m_0, h)}{\delta g(m_j, y)} = \frac{m_0}{m_j} \frac{g(m_0, \delta)}{g(m_j, y)} P_{0,j}(\delta, y).$$ (406)

Combining the above results we find,

$$
\begin{aligned}
P_{0,j}(h, y) = \ & \gamma_{\Lambda_0} \otimes \Big\{ g^{m_0/m_1 - 1}(m_1, h) \gamma_{\Lambda_1} \otimes \Big\{ g^{m_1/m_2 - 1}(m_2, h) \gamma_{\Lambda_2} \otimes \cdots \\
& \cdots \gamma_{\Lambda_{j-1}} \otimes \Big\{ g^{m_{j-1}/m_j}(m_j, h) \delta(h - y) \Big\} \Big\} \Big\}.
\end{aligned}
$$ (407)

In the last equation we have took the limit $m_0 = n \to 0$ so that $\lim_{m_0 \to 0} g(m_0, h) \to 1$.

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
