# Peer review of "Disorder-free spin glass transitions and jamming in exactly solvable mean-field models"

_SciPost Physics, doi:SciPost Phys. 4, 040 (2018)_

## Round 3 · Referee Report · Anonymous (Referee 1) · 2018-3-27

Strengths

1 - Structural glass with nice analytic properties and many interesting physical features.

2 - Most notably the model is exactly solvable.

3 - The problem exhibits a rich and interesting phase diagram.

4 - The analyses is carried out in two complementary ways.

Weaknesses

1 - The paper is assuming a strong familiarity with the methodology.

2 - There are several minor language mistakes.

Report

The manuscript studies an interesting structural glass problem that exhibits a variety of interesting features. Most notably it is exactly solvable, in the limit where the components of the spins goes to infinity, by virtue of the tree-like structure of the underlying graphical model. The author outlines nicely the different features and carries out an extensive analysis of the model. The different phases -- ferromagnetic phase, supercooled paramagnetic phase and glassy phase -- are analysed in two complementary ways. First by a mean-field calculation which can be carried out owing to the large connectivity of the variable degrees of the factor graph. And second by a density functional approach in the appendix. The consistency of the two approaches provides a nice cross-check of the results. The glassy structure is studied by a detailed replica analysis, including the full-RSB solution of the problem.

In my opinion the model exhibits many interesting features and the analysis is carried out with much care and in great detail.

Comments:

1 - The results in section III A are nice manifestation of the fact that the two limits in $M$ and $N$ can be taken separately in this model. I think this point could be stressed.

2 - The connection to the REM (section IV) can be understood as a consequence of the central limit theorem which applies here due to the tree-like structure of the underlying graphical model. This also explains why the derivation is reminiscent of that of the CLT.

3 - A discrete version of circular colouring has been studied for some particular graphical models in J. Stat. Mech. (2016) 083303.

4 - For readability things such as '... Eq. ( )' should be avoided. A separation by comma can be performed: '..., Eq. ( )' .

5 - There are some minor english flaws. Examples include, but are not limited to, missing preposition and articles such as in "in $...\to \infty$ limit" which should rather read something like "in the $... \to \infty$ limit".

Requested changes

1 - Fig. 2 b: It is said that both, (a) and (b), contain an example of a $p=2$-body interaction, although (b) shows a three-body interaction.

2 - II A: The assumptions on $\alpha$ and $p$, i.e. that they are constants of O(1), should be stated. Further I recommend to stress more clearly that the two dimensions, $N$ and $M$, scale independently as the consequences of this choice are very dramatic.

3 - II A: Equation (6) is a free entropy, not free energy.

4 - II C: It should be made clear that (18) in this form only holds at the line of jamming. Otherwise in the UNSAT phase the lower integration limit should not be zero.

5 - III A: I think it could be nice to outline why the symmetry is only present for $p=2$.

6 - III A: The second sum in (21) should run from $1$ to $M$

7 - III A 1: In equation (24) the Fourier-representation of the delta-distribution is introduced without mentioning the region of integration. I think that it is no good practice to introduce implicit notations like this, that are standard in the community, but can be very confusing to other readers.

8 - V A 2: The reference to (81) in the second paragraph should probably be to (67) and the following formula should be for the replicated partition function $Z^n$ instead of $Z$.

9 - A A: Notation change in (A2). $\prod_{l=1}^p (S_l)^{\mu}$ should be either $\prod_{l=1}^p (S_{l(\blacksquare)})^{\mu}$ or $\prod_{l=1}^p (S_{i_l})^{\mu}$. Similarly in (A11)

10 - A A: It is also assumed that the order parameter does not fluctuate with $i$. This translational invariance should be justified. It certainly should be mentioned.

---

## Round 3 · Referee Report · Anonymous (Referee 2) · 2018-4-24

Strengths

  1. Useful computations and techniques very clearly exposed
  2. Novel models introduced
  3. Wide phenomenology covered

Weaknesses

  1. Maybe too long, too many things together.

Report

The author introduces a family of spin systems with M components and studies its behaviour in a variety of physically relevant declinations, from various combinations of p-spin-like glassy systems (with both disorder-free induced frustration and quenched disorder) to soft/hardcore potentials. In particular, the glass and spin-glass transitions and the jamming transitions are exactly computed in the different models of the family, with a precise identification of their universality classes.The REM limit and the perceptron limit of this class of models are, as well, recovered.

The work is important, scientifically sound, computationally robust. It is a huge work, actually, extremely detailed, even pedagogical, and yet very precise and enlightening. Therefore, I would like to support the publication of the manuscript on SciPost after some pedantic observations of mine, herewith listed, are considered.

Mainly, I just raise two points (the rest goes in the list of changes)

1) Sec. II. A, I have a question about the scaling of X, Eq (4). In Eq. (3) \delta is of O(1) and, therefore, r, if I understood correctly.
In the definition of the disorder-free model X^μ =1, Eq (4). In the Ising case |S_k^μ| = 1. In the continuous model with spherical constraint it is |S_k^μ| ~ 1, at least if the connectivity is large enough.
With this choices Eq. (3) contains a sum over M terms of O(1), i.e., in the disorder-free case r ~ M/\sqrt{M} O(X^μ) ~ \sqrt{M}. I would ask the author to discuss the reason why this scaling Eq. (4) is adopted, rather than, e. g., X = a/\sqrt{M} + b \xi, with b = \sqrt{1-a^2}, a \in [0,1], being a = 0 the completely frustrated case and a = 1 the disorder-free case. On page 20, it is actually mentioned that - for the p \to \infty limit - a much weaker interaction energy scale O(1/\sqrt{M}) can be chosen for the disorder-free model. It is the choice that would keep T_c-T_K = O(1), rather than O(\sqrt{M}).

2) Sec. III.B, page 11. About the gauge transformation \tilde S_i \equiv \sigma_i S_i. How is the gauge transformation justified in the continuous spin case? The transformation is defined for Ising-like variables, including \sigma and \eta defined in Eq. (43). In the case of spherical value is this an approximation? If yes how is it justified? I would ask the author to discuss this point explicitly.

Requested changes

  1. Sec. II. A, when considering the Ising M-component spin, please consider the similarities with the so called M-p Ising spin model, see, e.g., PRE 60, 58 (1999) for finite dimensional lattice simulations, PRB 81, 064415 (2010) for simulations of power law interacting models and PRB 83, 104202 (2011) for mean-field theory.
  2. After Eq. (1), “where we will ind that exact analysis….”, please correct.
  3. Page 9, after Eq. (29), “integrations over and λ” in “integrations over λ”
  4. III.C “saddple” in “saddle” (also somewhere else in text) IV first paragraph “we we” in “we”.
  5. page 14, after Eq. (64), “becomes stack” in “becomes stuck” (?).
  6. page 16, after Eq. (77) “trance” in “trace”
  7. page 17, after Eq. (80), “the partition function Eq. (81)” in “the partition function Eq. (67)”.
  8. page 21, after Eq. (110) “and and that in terms” in “and in terms”.
  9. page 21, IV.C “Here assume limit” in “Here we limit” (?).
  10. page 23, VII intro, last line, “Although we may …. in the following but we must keep…” in “Although we may …. in the following, we must keep…”
  11. page 24, citation of Duplantier, Ref. [48], should be anticipated right after/before Eq. (127).
  12. Eq. (157) the “=“ sign is missing between second and third expression.
  13. Page 29, typo: the number of states should be e^{N\Sigma(f)} df and complexity \Sigma(f). Instead the symbol shortening Eq. (165) is used (only here, as far as I could notice).
  14. Eq. (179), second term: correct m_i in the numerator in m_{j-1} and m_{i+1} at the denominator in m_j. Third term: a closing parenthesis is missing in the exponent.
  15. VIII.C,D. An explicit reference should be given to the variational approach of Sommers and Dupont - J Phys C 17 (1984) - to solve Parisi antiparabolic equation. Between Eqs. (303) and (304) “faction” in “fraction”.
  16. page 47, first paragraph, “disorder-model” in “disorder-free model”.
  17. Figs. 7 and 9. Enlarge dots, lines and inner keys.

---

## Round 4 · Referee Report · Anonymous (Referee 1) · 2018-5-17

Strengths

1 - Structural glass with nice analytic properties and many interesting physical features.

2 - Most notably the model is exactly solvable.

3 - The problem exhibits a rich and interesting phase diagram.

4 - The analyses is carried out in two complementary ways.

Weaknesses

1 - The paper is assuming a strong familiarity with the methodology.

Report

The manuscript studies an interesting structural glass problem that exhibits a variety of interesting features. Most notably it is exactly solvable, in the limit where the components of the spins goes to infinity, by virtue of the tree-like structure of the underlying graphical model. The author outlines nicely the different features and carries out an extensive analysis of the model. The different phases -- ferromagnetic phase, supercooled paramagnetic phase and glassy phase -- are analysed in two complementary ways. First by a mean-field calculation which can be carried out owing to the large connectivity of the variable degrees of the factor graph. And second by a density functional approach in the appendix. The consistency of the two approaches provides a nice cross-check of the results. The glassy structure is studied by a detailed replica analysis, including the full-RSB solution of the problem.

In my opinion the model exhibits many interesting features and the analysis is carried out with much care and in great detail.

---

## Round 4 · Referee Report · Anonymous (Referee 2) · 2018-5-21

Strengths

  1. Useful computations and techniques very clearly exposed
  2. Novel models introduced
  3. Wide phenomenology covered

Weaknesses

None

Report

The manuscript appears to be substantially improved and I support its publication in the current version.

---

## Round 4 · Author Response

Warnings issued while processing user-supplied markup:

  • Inconsistency: plain/Markdown and reStructuredText syntaxes are mixed. Markdown will be used.
    Add "#coerce:reST" or "#coerce:plain" as the first line of your text to force reStructuredText or no markup.
    You may also contact the helpdesk if the formatting is incorrect and you are unable to edit your text.

I would like to thank the referees for their efforts and patience to carefully read the manuscript and provide us useful comments. The following is our responses.

(Note) In the following, (A).. is our response to the comments. In our responses we refer to the equation numbers Eq. (...), section numbers and page numbers of those in the revised version.

<< report 1 >>

[Comments:]

1 - The results in section III A are nice manifestation of the fact that the two limits in M and N can be taken separately in this model. I think this point could be stressed.

(A) I agree. I added a sentence to emphasize this point in the end of sec. 3.1 (p10). I also changed the sentence earlier, in the paragraph below Eq. (2) to emphasize the same point.

2 - The connection to the REM (section IV) can be understood as a consequence of the central limit theorem which applies here due to the tree-like structure of the underlying graphical model. This also explains why the derivation is reminiscent of that of the CLT.

(A) Yes I agree. I put a paragraph to point this out in the send of sec 4.

3 - A discrete version of circular coloring has been studied for some particular graphical models in J. Stat. Mech. (2016) 083303.

(A) Thank you for the reference. I put a citation to it in the introduction (p5, 1st paragraph).

4 - For readability things such as '... Eq. ( )' should be avoided. A separation by comma can be performed: '..., Eq. ( )’ .

(A) I changed these to " ... given by Eq.() ".

5 - There are some minor English flaws. Examples include, but are not limited to, missing preposition and articles such as in "in . . . $\to \infty$ limit" which should rather read something like "in the ...$\to \infty$ limit”.

(A) I fixed these in the revised version.

[Requested changes]

1 - Fig. 2 b: It is said that both, (a) and (b), contain an example of a p = 2 -body interaction, although (b) shows a three-body interaction.

(A) The error is fixed.

2 - II A: The assumptions on α and p, i.e. that they are constants of O(1), should be stated. Further I recommend to stress more clearly that the two dimensions, N and M, scale independently as the consequences of this choice are very dramatic.

(A) I agree. I changed the paragraph below Eq (2) to state these explicitly.

3 - II A: Equation (6) is a free entropy, not free energy. (A) Missing minus sign is supplied in Eq (7).

4 - II C: It should be made clear that (18) in this form only holds at the line of jamming. Otherwise in the UNSAT phase the lower integration limit should not be zero.

(A) Right. I changed the lower limit of the integral in Eq. (19) to $-\infty$.

5 - III A: I think it could be nice to outline why the symmetry is only present for p = 2 .

(A) I agree. I put a footnote in p.10 to explain this point.

6 - III A: The second sum in (21) should run from 1 to M

(A) The error is fixed.

7 - III A 1: In equation (24) the Fourier-representation of the delta-distribution is introduced without mentioning the region of integration. I think that it is no good practice to introduce implicit notations like this, that are standard in the community, but can be very confusing to other readers.

(A) I agree. I made explicit the regions of integrations in this and other similar and related equations.

8 - V A 2: The reference to (81) in the second paragraph should probably be to (67) and the following formula should be for the replicated partition function Zn instead of Z.

(A) Right. The error is fixed.

9 - A A: Notation change in (A2). $\prod_{l=1}^{p}(S_{l})^{\mu}$ should be either $\prod_{l=1}^{p}(S_{l(\blacksquare)})_{\mu}$ or $\prod_{l=1}^{p}(S_{i_{l}})_{\mu}$. Similarly in (A11)

(A) Actually it is OK here because the spin variables which appear here are just dummy variables to introduce the Mayer function.

10 - A A: It is also assumed that the order parameter does not fluctuate with i. This transnational invariance should be justified. It certainly should be mentioned.

(A) Since the system is regular and every vertex is exactly equivalent to each other in our system, it is natural to expect the order parameter does not fluctuate in space. I put this remark below Eq. (330). Similar remarks are put also in sec 3.1.2 (p12) below Eq (36), sec 5.1.2 (p21) below Eq (84) and sec 5.2.2 (p25) below Eq (108).

<< report 2 >>

[Comments]

Mainly, I just raise two points (the rest goes in the list of changes)

1) Sec. II. A, I have a question about the scaling of X, Eq (4). In Eq. (3) \delta is of O(1) and, therefore, r, if I understood correctly. In the definition of the disorder-free model X^μ =1, Eq (4). In the Ising case |S_k^μ| = 1. In the continuous model with spherical constraint it is |S_k^μ| ~ 1, at least if the connectivity is large enough. With this choices Eq. (3) contains a sum over M terms of O(1), i.e., in the disorder-free case r ~ M/\sqrt{M} O(X^μ) ~ \sqrt{M}. I would ask the author to discuss the reason why this scaling Eq. (4) is adopted, rather than, e. g., X = a/\sqrt{M} + b \xi, with b = \sqrt{1-a^2}, a \in [0,1], being a = 0 the completely frustrated case and a = 1 the disorder-free case. On page 20, it is actually mentioned that - for the p \to \infty limit - a much weaker interaction energy scale O(1/\sqrt{M}) can be chosen for the disorder-free model. It is the choice that would keep T_c-T_K = O(1), rather than O(\sqrt{M}).

(A) The reason for this parametrization is to make the energy scale of glass transition O(1) and keep it independent of the parameter \lambda, whose range is 0 < \lambda/\sqrt{M} < 1. By varying this \lambda, the glass transition temperature does not change but that of the crystalline one does. See discussions below (85) and the paragraph below (105). (With the one proposed by the referee, the glass energy scale decreases with increasing a.)

2) Sec. III.B, page 11. About the gauge transformation \tilde S_i \equiv \sigma_i S_i. How is the gauge transformation justified in the continuous spin case? The transformation is defined for Ising-like variables, including \sigma and \eta defined in Eq. (43). In the case of spherical value is this an approximation? If yes how is it justified? I would ask the author to discuss this point explicitly.

(A) Here we are only considering a class of gauge transformations defined by gauge variables which are Ising. This transformation can be applied to both continuous and Ising variables without problems. This does not change the nature of the spins. In the present paper we are limiting ourselves with crystalline states which can be related to the simple ferromagnetic one by such gauge transformation. To explain this point more carefully, I revised the text at the beginning of sec 3.2. In some system there are other types of gauge transformations. For instance, in the so called "gauge glass model", a certain continuous gauge transformation is possible. There the Ising one which we consider can be viewed as a subset of the continuous one. The systems we consider do not have such continuous gauge invariance.

[Requested changes]

  1. Sec. II. A, when considering the Ising M-component spin, please consider the similarities with the so called M-p Ising spin model, see, e.g., PRE 60, 58 (1999) for finite dimensional lattice simulations, PRB 81, 064415 (2010) for simulations of power law interacting models and PRB 83, 104202 (2011) for mean-field theory.

(A) Thank you very much for this remark. I fully agree. I created a new paragraph in the end of sec 2.1 to comment on the similarity to the M-p Ising spin models.

  1. After Eq. (1), “where we will ind that exact analysis....”, please correct.

(A) The error is fixed.

  1. Page 9, after Eq. (29), “integrations over and λ” in “integrations over λ”

(A) The error is fixed.

  1. III.C “saddle” in “saddle” (also somewhere else in text) IV first paragraph “we we” in “we”.

(A) The errors are fixed.

  1. page 14, after Eq. (64), “becomes stack” in “becomes stuck” (?).

(A) The errors is fixed.

  1. page 16, after Eq. (77) “trance” in “trace”

(A) The errors is fixed.

  1. page 17, after Eq. (80), “the partition function Eq. (81)” in “the partition function Eq. (67)”.

(A) The errors is fixed.

  1. page 21, after Eq. (110) “and and that in terms” in “and in terms”.

(A) The errors is fixed.

  1. page 21, IV.C “Here assume limit” in “Here we limit” (?).

(A) Right. The errors is fixed.

  1. page 23, VII intro, last line, “Although we may .... in the following but we must keep...” in “Although we may .... in the following, we must keep...”

(A) The errors is fixed.

  1. page 24, citation of Duplantier, Ref. [48], should be anticipated right after/before Eq. (127).

(A) Done.

  1. Eq. (157) the “=“ sign is missing between second and third expression.

(A) "=" is inserted.

  1. Page 29, typo: the number of states should be e^{N\Sigma(f)} df and complexity \Sigma(f). Instead the symbol shortening Eq. (165) is used (only here, as far as I could notice).

(A) The error is fixed.

  1. Eq. (179), second term: correct m_i in the numerator in m_{j-1} and m_{i+1} at the denominator in m_j. Third term: a closing parenthesis is missing in the exponent.

(A) The errors are fixed.

  1. VIII.C,D. An explicit reference should be given to the variational approach of Sommers and Dupont - J Phys C 17 (1984) - to solve Parisi antiparabolic equation.

(A) I put citation to the reference at the beginning of sec 8.

Between Eqs. (303) and (304) “faction” in “fraction”.

(A) The error is fixed.

  1. page 47, first paragraph, “disorder-model” in “disorder-free model”.

(A) The errors are fixed.

  1. Figs. 7 and 9. Enlarge dots, lines and inner keys.

(A) The figures are updated following the suggestions.

---

## Round 4 · List of Changes

List of changes

  • A comment on a discrete version of circular coloring with a citation to the reference suggested in report 1 (requested change 3) is added in p5, 1st paragraph.
  • Paragraph below Eq. (2) is updated following report 1 (comment 1 and requested change 2).
  • A paragraph (last paragraph in p7) is added to comment on the M-p spinglass models as suggested in report 2 (requested change 1).
  • The sentence at the end of 3.1 (p10) is added following the suggestion in report 1 (comment 1).
  • A footnote on the symmetry of the model is added in p10 following the suggestion in report 1 (requested change 5).
  • Comments on the spacial uniformity of the order parameters are added regarding the suggestion in report 1 (requested change 10) : In appendix A, below Eq. (330), sec 3.1.2 (p12) below Eq. (36), sec 5.1.2 (p21) below Eq. (84) and sec 5.2.2 (p25) below Eq. (108).
  • Comments on the gauge transformations in sec 3.2 are slightly revised.
  • A remark on the similarity of the derivation of the REM (random energy model) and the standard proof of the central limit theorem is mentioned in the end of sec. 4 as suggested in report 1 (comment 2).
  • A new paragraph below Eq. (117) (p27) is added to make connections to related equations Eq. (46) and (48).
  • sec 8.1 Parisi's ansatz is revised adding new paragraphs at the end, for pedagogical reasons.
  • Some texts regarding local stability against crystallization are slightly revised: 2nd and 3rd paragraphs in sec 5.3, the end of sec 9, beginning of sec 10.
  • Other minor changes: corrections of minor errors, mistypes, wording pointed out in the reports. Some formulas written in some sentences are displayed as equations.

---

## Editorial Decision

published